# Paneth cell SIRT1 deficiency increases intestinal stress resistance by modulating the gut microbiota

Liz M Garcia-Peterson [ID][1,2,10], Alicia S Wellman [ID][1,10], Xiaojiang Xu [ID][3,9,10], Ming Ji[1,10], Caroline Duval[1], Igor Shats[1], Xiaoyue Wu [ID][1], Thomas A Randall[3], Hamed Bostan [ID][3], David Cunefare[4,5], Charan K Ganta[4,6], Maria Sifre [ID][1], Xin Xu [ID][7], Richard S Blumberg[8], Jian-Liang Li [ID][3] & Xiaoling Li [ID][1✉]

## Abstract

**Paneth cells, intestine-originated innate immune-like cells, are important for maintenance of the intestinal stem cell niche, gut microbiota, and gastrointestinal barrier. Dysfunctional Paneth cells under pathological conditions are a site of origin for intestinal inflammation. However, mechanisms underlying stress-induced Paneth cell dysregulation remain unclear. Here, we report that SIRT1, the most conserved mammalian NAD⁺-dependent protein deacetylase and a well-known genetic repressor of inflammation, cell-autonomously suppresses Paneth cell function and sensitizes the gut epithelium to environmental stress. Specifically, deletion of Paneth cell SIRT1 in mice elevates Wnt signaling and ATF4/ endoplasmic reticulum stress pathway in Paneth cells. These molecular alterations are coupled with increased Paneth cell abundance and enhanced anti-microbial peptide production in young mice, improved protection against intestinal immune cell expansion in aged mice, and increased resistance to chemically induced colitis. Using microbiota-depleted mice with or without fecal transplantation, we further demonstrate that Paneth cell SIRT1 deficiency ameliorates colitis by interacting with the gut microbiota. Collectively, our findings uncover an unanticipated function of Paneth cell SIRT1 in conferring stress sensitivity in the gut epithelium.**

**Keywords** SIRT1; Wnt/β-catenin; ER Stress; Anti-microbial Peptides; Gut Microbiota
**Subject Categories** Immunology; Microbiology, Virology & Host Pathogen Interaction; Signal Transduction

## Introduction

The intestinal epithelium, a monolayer of columnar epithelial cells organized into crypts and villi, is essential for nutrient metabolism and waste elimination (Reed and Wickham, 2009). It contains absorptive enterocytes and secretory cells such as Paneth, goblet, and enteroendocrine cells, all of which are replenished continuously from intestinal stem cells (ISCs) residing at the bottom of the crypts (Noah and Shroyer, 2013). The intestinal epithelium is also critical in immune surveillance and homeostasis (Mason et al, 2008; Reed and Wickham, 2009). It comprises the largest compartment (70–80%) of the immune system in response to the constant ingestion of foreign antigens and environmental agents (Mowat and Agace, 2014). Additionally, the nutrient-rich intestinal mucosal surface is the largest surface of our body that harbors billions of gut microbiota, which interact extensively with the host and contribute substantially to its evolutionary fitness. Therefore, intestinal epithelial homeostasis is maintained by a complex interplay between epithelial cells, immune cells, and gut microorganisms. Disruption of this interaction contributes to inflammatory and immune diseases associated with gut microbiome dysbiosis.

One genetic factor that modulates the interactions between gut epithelium, immune cells, and microbiota is sirtuin 1 (SIRT1), the most conserved mammalian NAD⁺-dependent protein deacetylase (Houtkooper et al, 2012). As a predominant nuclear sirtuin, SIRT1 is a critical nuclear metabolic sensor that enhances stress resistance, regulates metabolism, inhibits apoptosis, and suppresses inflammation in response to environmental signals. Importantly, SIRT1 has been identified as a key repressor of inflammation in multiple tissues/cells over the past decades (Yang et al, 2022b). For instance, we and others have shown that deficiency of intestinal epithelial SIRT1 leads to intestinal inflammation, disruption of the gut microbial composition, and altered susceptibility to environmentally induced colitis and tumorigenesis (Kazgan et al, 2014; Lo Sasso et al, 2014; Ren et al, 2017; Wellman et al, 2017). Particularly, deletion of SIRT1 in the whole intestinal epithelium (SIRT1 iKO) results in increased abundance and activity of Paneth cells (Lo

[1]Molecular and Cellular Biology Laboratory, National Institute of Environmental Health Sciences, Research Triangle Park, NC 27709, USA. [2]Postdoctoral Research Associate Training (PRAT) Program, National Institute of General Medical Sciences, Bethesda, MD 20892, USA. [3]Integrative Bioinformatics, National Institute of Environmental Health Sciences, Research Triangle Park, NC 27709, USA. [4]Comparative and Molecular Pathogenesis Branch, National Institute of Environmental Health Sciences, Research Triangle Park, NC 27709, USA. [5]Experimental Pathology Laboratories, Morrisville, NC 27560, USA. [6]Inotiv, Morrisville, NC 27560, USA. [7]Epigenetics and Stem Cell Biology Laboratory, National Institute of Environmental Health Sciences, Research Triangle Park, NC 27709, USA. [8]Department of Medicine, Brigham and Women's Hospital, Boston, MA 02115, USA. [9]Present address: Department of Pathology and Laboratory Medicine, Tulane University School of Medicine, New Orleans, LA 70112, USA. [10]These authors contributed equally: Liz M Garcia-Peterson, Alicia S Wellman, Xiaojiang Xu, Ming Ji. ✉E-mail: lix3@niehs.nih.gov

 

Sasso et al, 2014; Wellman et al, 2017), which are intestine-originated innate immune cells crucial for maintenance of the intestinal stem cell niche (Sato et al, 2011). As the main source of antimicrobial peptides in the intestine, Paneth cells also play a key role in establishing and shaping the gut microbiota and maintaining the integrity of the gastrointestinal barrier (Alenghat et al, 2013; Bevins and Salzman, 2011). Under pathological conditions, Paneth cells have been reported to be a site of origin for intestinal inflammation (Adolph et al, 2013). However, it remains unknown whether SIRT1 suppresses the abundance and activity of Paneth cells in a cell-autonomous manner by inhibiting the expression of functional genes within the Paneth cells or in a non-cell-autonomous manner by affecting their differentiation and maturation from progenitor cells or by regulating environmental factors that influence Paneth cell proliferation and activity. The functional impacts of SIRT1 deficiency in Paneth cells on intestinal epithelial homeostasis are still unclear.

Here, using a newly generated SIRT1 KO mouse model in which SIRT1 was specifically deleted in Paneth cells, we report that SIRT1 in Paneth cells functions to suppress their anti-microbial activity. This suppression affects the gut microbiota and makes the gut epithelium more susceptible to aging- or chemical-induced stress and inflammation.

# Results

## Deletion of SIRT1 in Paneth cells increases Paneth cell abundance and attenuates age-induced increase in mucosal immune cells

To investigate whether and how SIRT1 cell-autonomously regulates Paneth cells, we generated a Paneth-cell specific SIRT1 KO mouse model (SIRT1 PKO) by breeding mice containing a SIRT1 Flox allele (Cheng et al, 2003) with a Paneth-cell-specific Defa6-Cre line (Adolph et al, 2013) (Fig. EV1A,B). Young SIRT1 PKO mice ($Sirt1^{f/f}$;$Defa6$-$Cre^+$) were phenotypically normal under the standard feeding condition compared to age- and gender-matched Flox littermates ($Sirt1^{f/f}$;$Defa6$-$Cre^-$), and displayed no signs of defects in morphology and proliferation of intestinal epithelial cells (Fig. EV1C,D). They also had normal expression levels of almost all tested pro-inflammatory genes (Fig. EV1E) and normal abundance of all analyzed immune cells (Fig. EV1F) in the ileum. The mRNA levels of all tested pro-inflammatory genes were also normal in the colon (Fig. EV1E) despite increased abundances of $CD8^+$ T cells and B cells (Fig. EV1F).

We have previously shown that deletion of SIRT1 in the whole intestinal epithelium activates Paneth cells in the small intestine, particularly in aged mice (Wellman et al, 2017). To analyze whether specific deletion of SIRT1 in Paneth cells also age-dependently affects Paneth cell abundance/activation, we performed a single-cell RNA-seq (scRNA-seq) analysis of all live cells isolated from the small intestines of young (2–3 months) and aged (12–14 months) Flox and SIRT1 PKO female mice. Based on the individual cell transcriptome and the expression patterns of known marker genes, the small intestinal cells could be categorized into 17 functional groups, including 9 epithelial cell groups and 8 immune cell groups (Figs. 1A and EV2A). Interestingly, both young and aged SIRT1 PKO mice had an increased abundance of Paneth cells compared

with age-matched Flox mice (Figs. 1B and EV2B). This increase was coupled with the elevation of intermediate ISCs (Lg5 low), transit amplifying cells, and enterocytes, particularly in aged mice (Figs. 1B and EV2B), suggesting that SIRT1 PKO mice might have an increased intestinal epithelial renewal. Further FACS analysis of $CD24^{high}SSC^{high}$ Paneth cells in the small intestines of young Flox and SIRT1 PKO female mice on the Lgr5-EGFP background confirmed the significant increase of the Paneth cell fraction in SIRT1 PKO mice without any notable alteration of their Lgr5-EGFP positive ISCs (Figs. 1C and EV2C,D). Moreover, the small intestinal organoids cultured from freshly isolated crypts from young SIRT1 PKO mice on the R26-TdTomato background ($Sirt1^{f/f}$;$R26$-$TdTomato^+$;$Defa6$-$Cre^+$) contained a significantly higher proportion of $TdTomato^+$ Paneth cells compared to those from matched SIRT1 PHet control mice ($Sirt1^{f/+}$;$R26$-$TdTomato^+$;$Defa6$-$Cre^+$, Fig. 1D). Therefore, specific deletion of SIRT1 in Paneth cells increases their abundance in young and aged mice.

From the scRNA-seq analysis, we also observed that age induced a reduction of the epithelial cell fraction along with a massive expansion of immune cells in the small intestine of Flox female mice. However, this epithelial cell reduction/immune cell expansion was blunted in aged SIRT1 PKO female mice (Figs. 1B and EV2B). Further FACS analysis confirmed that aged SIRT1 PKO female mice had a significantly reduced fraction of $CD45^+$ immune cells in the small intestine than Flox controls (Fig. 1E, Small Intestine). They also displayed a trend of reduction of $CD45^+$ immune cells in the blood (Fig. 1E, Blood). Histological analysis of 2-year-old female mice further showed that the fraction of SIRT1 PKO mice that had mucosal lymphoid follicular hyperplasia in the ileum was lower than that of Flox controls (Fig. 1F). Interestingly, age also induced an expansion of immune cells in the colon of Flox female mice, but SIRT1 PKO mice were not protected from this expansion (Fig. EV2E). Therefore, the increased Paneth cell abundance observed in aged SIRT1 PKO mice is linked to reduced abundance of small intestinal immune cells, which contrasts with our previous findings in aged SIRT1 iKO mice (Wellman et al, 2017).

## SIRT1 deficiency in Paneth cells induces ER stress response and increases antimicrobial peptide production in Paneth cells

To better understand how deletion of SIRT1 specifically in Paneth cells protected against age-induced increase of intestinal intrae-pithelial immune cells, we analyzed the transcriptomes of Paneth cells from the scRNA-seq dataset. Gene Set Enrichment analysis (GSEA) showed that SIRT1 deficient Paneth cells had increased gene sets involved in the unfolded protein response (UPR) associated with endoplasmic reticulum (ER) stress, but reduced gene sets involved in mitochondrial functions in both young and aged mice (Fig. 2A; Appendix Fig. S1A). Gene Ontology (GO) analysis of differentially expressed genes in Paneth cells further revealed the increased expression of genes involved in chemical and protein transport, host defense, and stress response in SIRT1 KO Paneth cells compared to Flox Paneth cell controls (Fig. 2B and Dataset EV1), suggesting that deletion of SIRT1 in Paneth cells increases their stress response despite the possible impairment of mitochondrial functions. Subsequent analysis of the intestinal epithelial cell (IEC) fraction in the scRNA-seq dataset showed that

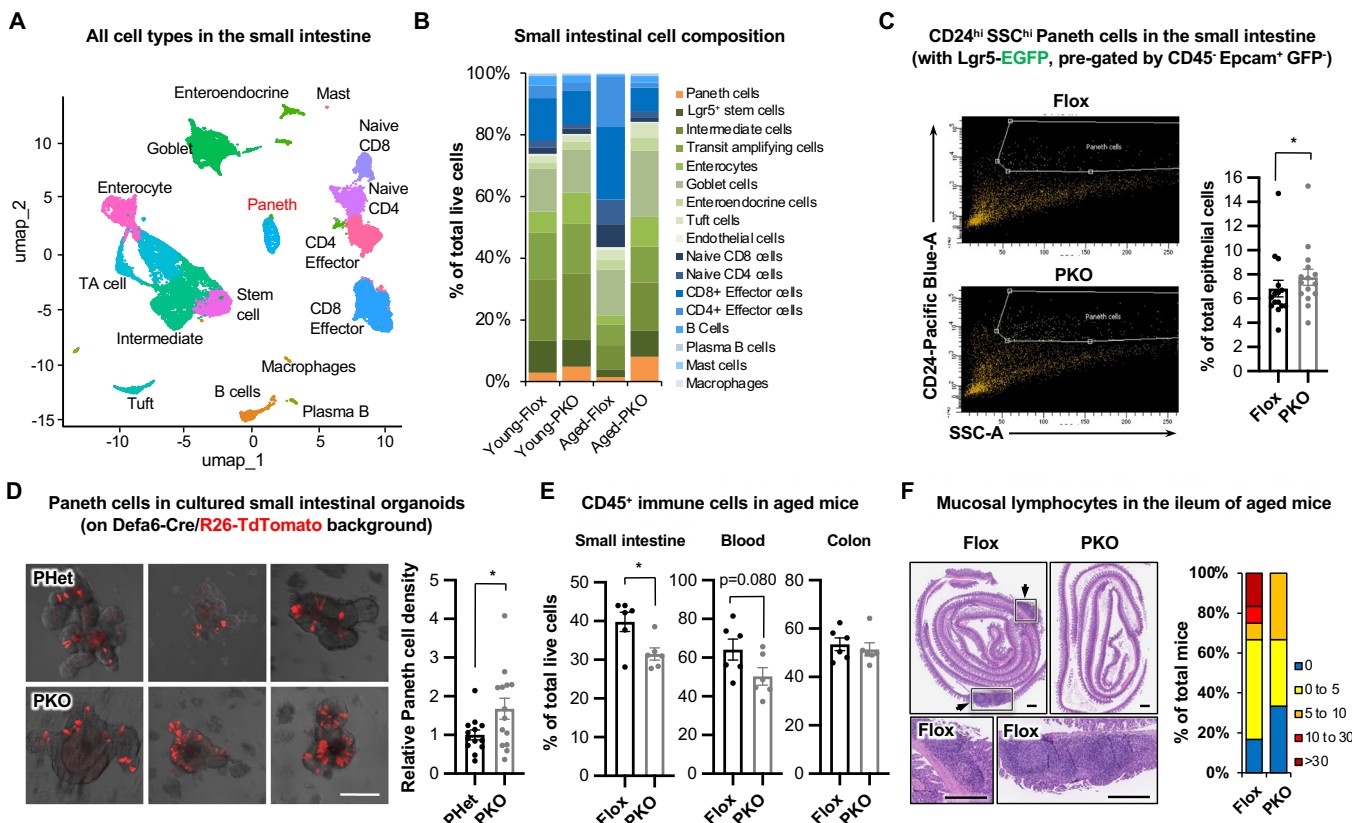

**Figure 1. Deletion of SIRT1 in Paneth cells increases Paneth cell abundance.**

(A) Annotation of total live cells from the small intestine of Flox and SIRT1 PKO mice. Total live cells isolated from the small intestine of young (2–3 months) and aged (12–14 months) mice were analyzed by scRNA-seq. (B) SIRT1 PKO mice have increased Paneth cell abundance in both young and aged mice. Total intestinal live cells were analyzed by scRNA-seq as in (A). (C) SIRT1 PKO mice have increased abundance of Paneth cells in the small intestine. The abundance of Paneth cells in Flox and PKO mice on the Lgr5-GFP background was analyzed by FACS ($n = 15$ pairs, paired t-test). (D) The small intestinal organoids from SIRT1 PKO mice have a higher density of Paneth cells compared to those from SIRT1 PHet mice. The freshly isolated small intestinal epithelial cells (mainly crypts) from SIRT1 PHet/TdTomato control and SIRT1 PKO/TdTomato mice were cultured in the Intesticult medium for 6 days. The area of organoids and Paneth cell number in organoids were quantified in Fiji ($n = 13$ PHet and 14 PKO organoids, Student's t-test). Bar, 200 μm. (E) Aged SIRT1 PKO mice (10–11 months) have reduced CD45+ immune cells in the small intestine. The fraction of CD45+ immune cells in total live cells isolated from Flox and PKO mice was analyzed by FACS as described in Methods ($n = 6$ Flox and 6 PKO, Student's t-test). (F) Two-year-old SIRT1 PKO mice have reduced mucosal immune cells (arrows) in the ileum ($n = 5$ out of 12 Flox mice and 1 out of 6 PKO mice display patches of mucosal lymphoid follicles; right panel, % of tissue area with mucosal immune cells was quantified by Fiji and categorized as indicated). Bars, 500 μm. Data information: in (C, D, and E), values are expressed as mean ± SEM; *$p < 0.05$; no marks, not significant. Source data are available online for this figure.

Paneth cells were clustered into two clusters, cluster 12 and 16 (Fig. 2C,D; Appendix Fig. S1B and Appendix Table S1). Both clusters of SIRT1 KO Paneth cells had significantly increased expression of many anti-microbial genes, particularly in aged mice (Fig. 2E), indicating that SIRT1 deficiency cell-autonomously activates Paneth cells. The increased expression of several anti-microbial genes could also be observed in the whole ileum tissue by qPCR (Fig. 2F).

## SIRT1 deficiency in Paneth cells enhances Wnt and ATF4 signaling

Among the two Paneth cell clusters, cells in cluster 16 were more differentiated than those in cluster 12 as determined by pseudo time analysis (Fig. 3A; Appendix Fig. S1C), with significantly higher expression of various defensin genes but reduced expression of multiple mitochondrial genes (Dataset EV2). The cluster 16

Paneth cells also had a significantly higher level of *Wnt3*, a key Wnt signaling molecule primarily derived from Paneth cells among the gut epithelial cells which is involved in supporting Lgr5+ ISCs (Farin et al, 2012) (Fig. 3B, Appendix Table S1, and Dataset EV2), suggesting that this cluster of Paneth cells are more important to maintain the ISC niche than the cluster 12 Paneth cells. Consistent with this observation, age-associated reduction of Lgr5+ ISCs in Flox controls was coupled with a marked depletion of the cluster 16 but not cluster 12 Paneth cells (Fig. 3C, Aged-Flox vs Young-Flox). Importantly, SIRT1 deficiency in the cluster 16 Paneth cells significantly elevated their expression of *Wnt3* in aged mice (Fig. 3B, Aged-PKO vs Aged-Flox), which was accompanied by a complete protection against age-induced depletion of the cluster 16 Paneth cells and Lgr5+ ISCs (Fig. 3C, Aged-PKO vs Young-PKO). These results suggest that SIRT1 in the cluster 16 Paneth cells increases their sensitivity to aging, which in turn reduces Paneth cell Wnt3 production, leading to age-dependent

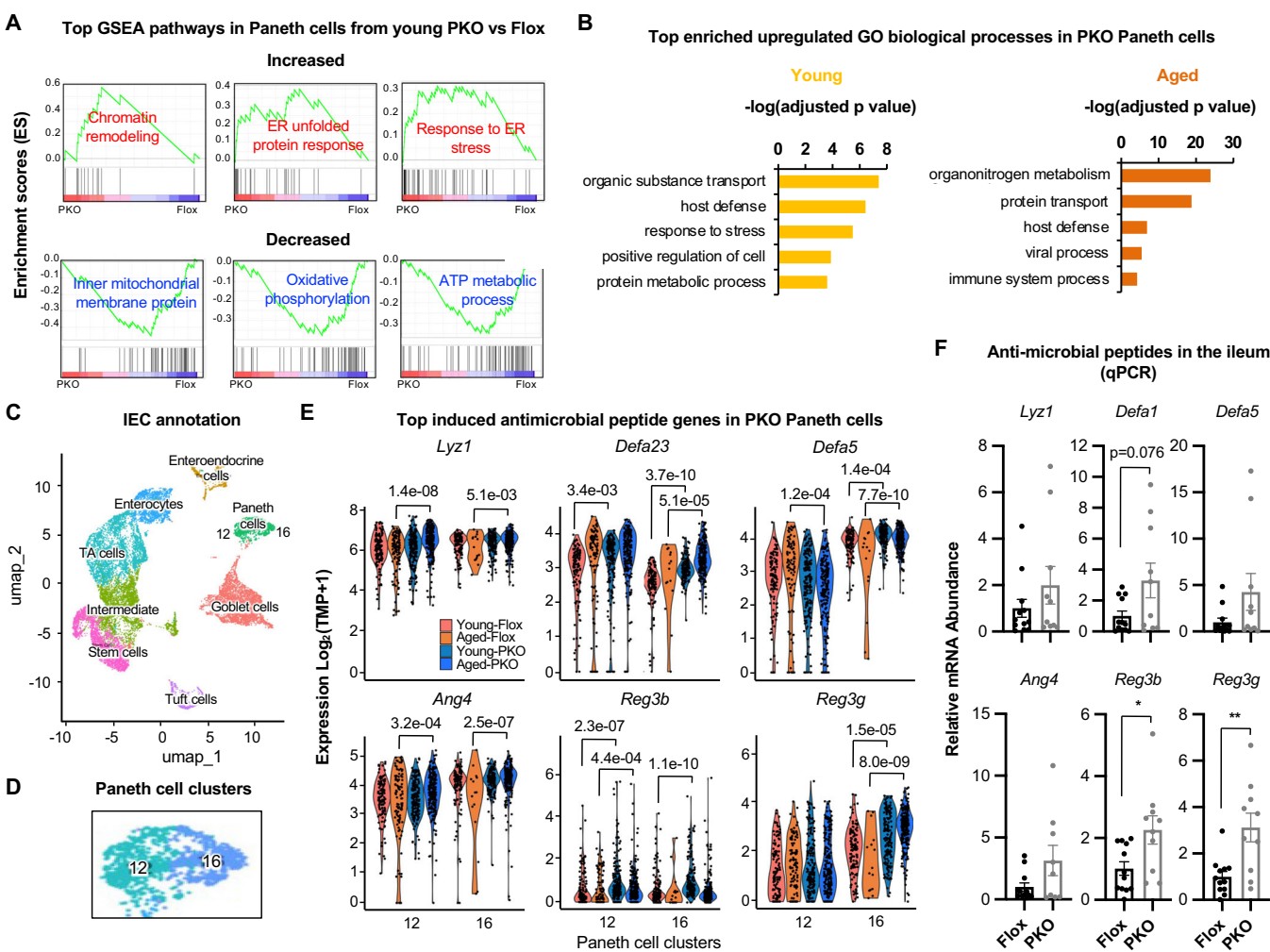

**Figure 2. Deletion of Paneth cell SIRT1 increases ER stress response in Paneth cells.**

(A) SIRT1 KO Paneth cells have increased expression of gene sets involved in ER stress response but reduced expression of gene sets mediating mitochondrial activities. Transcriptomes of Flox and SIRT1 KO Paneth cells from scRNA-seq dataset were analyzed. Top enriched Gene set enrichment (GSEA) pathways in SIRT1 KO vs Flox Paneth cells in young mice are shown. (B) SIRT1 KO Paneth cells have increased expression of genes involved in transport and host defense compared to Flox Paneth cells. Differentially Expressed Genes (DEGs) in SIRT1 KO vs Flox Paneth cells from scRNA-seq dataset were analyzed by GO biological process and top enriched pathways are shown (Permutation testing with the Benjamini–Hochberg adjusted $p$-values. Enrichment score represents $-\log_{10}$-transformed adjusted $p$-values. (C) Annotation of intestinal epithelial cells (IECs) in the small intestine of Flox and SIRT1 PKO mice. (D) Two clusters of Paneth cells. (E) SIRT1 KO Paneth cells have increased expression of anti-microbial peptide genes. The color-coded violin plots display the expression distribution of anti-microbial peptide genes involved in host defense in two clusters of Paneth cells from indicated scRNA-seq samples. Each dot represents a single cell. Gene expression changes between samples were compared and the significance of change was labeled (two-sided Wilcoxon test). (F) SIRT1 PKO mice have increased expression of several anti-microbial peptide genes in the ileum. The mRNA levels of indicated anti-microbial peptide genes in the whole ileum tissue were analyzed by qPCR ($n = 12$ Flox and 10 PKO, Student's t-test). Data information: in (F), values are expressed as mean ± SEM; $^*p < 0.05$, $^{**}p < 0.01$; no marks, not significant. Source data are available online for this figure.

depletion of Lgr5⁺ ISCs. Since Wnt3 can also cell-autonomously promote differentiation of Paneth cells (Farin et al, 2012), this action of SIRT1 could inhibit the Wnt signaling in both ISCs and Paneth cells. In line with this possibility, β-catenin, the downstream transcription factor of the Wnt signaling, was highly enriched in the nucleus in crypt cells, and aged SIRT1 PKO mice had notably more crypt cells with intense nuclear staining of β-catenin compared to age matched Flox mice (Fig. 3D). Further Artificial Intelligence (AI)-assisted quantification showed that ISCs and Paneth cells at the bottom of the crypts had high nuclear staining intensities of β-catenin compared to epithelial cells at the

upper portion of the crypts (Appendix Fig. S1D,E, most with scores higher than 9). Moreover, the ileal epithelium of aged PKO mice contained a significantly increased fraction of nuclei with an intensity score of 11 (Fig. 3E). The average nuclear β-catenin intensity in the whole ileal epithelial tissue also tended to be higher in aged PKO mice compared to that in age-matched Flox mice (Fig. 3F). Immunoblotting confirmed that aged PKO mice had increased β-catenin protein in the ileum (Fig. 3G). Furthermore, aging significantly reduced the average nuclear staining intensity of β-catenin in the ileal epithelium of Flox mice but failed to do so in SIRT1 PKO mice (Fig. 3F), supporting the notion that Paneth cell

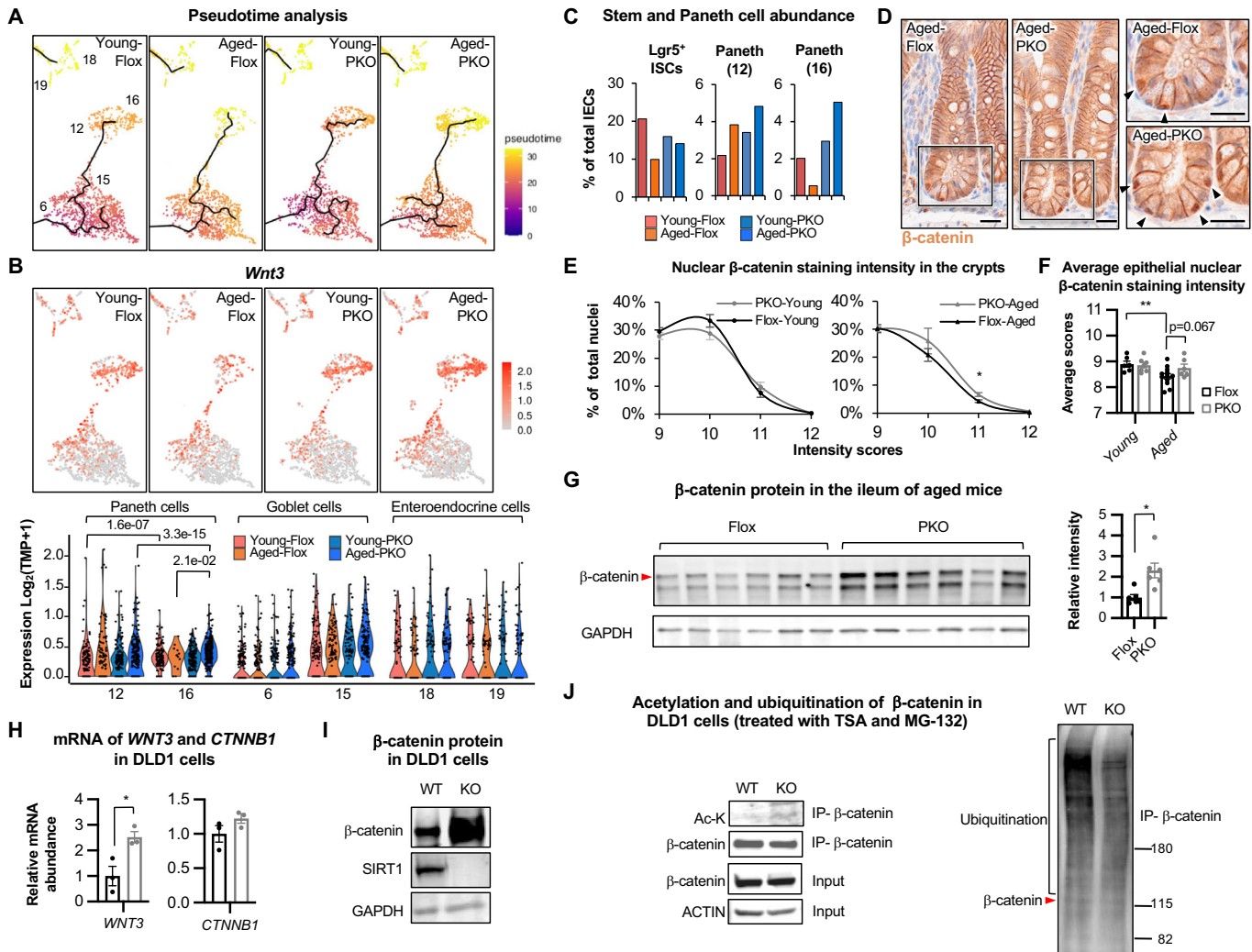

**Figure 3. Deletion of SIRT1 enhances Wnt signaling pathways in Paneth cells.**

(A) Pseudotime analysis of IECs in the small intestine of young and aged Flox and SIRT1 PKO mice. (B) The expression of *Wnt3* is increased in cluster 16 Paneth cells of aged SIRT1 KO mice. The expression of *Wnt3* is projected onto tSNE (upper) or shown in violin plots (bottom, two-sided Wilcoxon test). (C) Deletion of Paneth cell SIRT1 reduces age-induced depletion of cluster 16 Paneth cells and Lgr5+ ISCs. (D–F) Aged SIRT1 PKO mice have increased nuclear β-catenin accumulation in the crypts than Flox mice. The nuclear β-catenin intensity in total ileal epithelium was quantified by AI as described in Methods. Bars in (D), 20 μm. (E) The fraction of nuclei with high β-catenin intensity scores (>9, nuclei in the crypts; $n = 6$ Flox-Young, 12 Flox-Aged, 7 PKO-Young, and 6 PKO-Aged mice, Student's t-test). (F) The average nuclear β-catenin intensity scores in all epithelial tissue ($n = 6$ Flox-Young, 12 Flox-Aged, 7 PKO-Young, and 6 PKO-Aged mice, two-way ANOVA). (G) The protein level of β-catenin is increased in the ileum of SIRT1 PKO mice ($n = 6$ Flox and 6 PKO, Student's t-test). (H) SIRT1 KO colon epithelial cells have increase mRNA levels of *WNT3* but not *CTNNB1* genes. WT and SIRT1 KO DLD1 cells were cultured in regular RMPI medium ($n = 3$ biological replicates, Student's t-test). (I) SIRT1 KO colon epithelial cells have increased β-catenin protein. WT and SIRT1 KO DLD1 cells were cultured in regular RMPI medium. (J) β-catenin is hyper-acetylated and hypo-ubiquitinated in SIRT1 KO colon epithelial cells. WT and SIRT1 KO DLD1 cells cultured in regular RMPI medium were treated with 5 μM TSA and 10 μM MG-132 for 3 h before harvesting. Total cell lysates were immunoprecipitated with anti-β-catenin antibodies (IP-β-catenin), then immunoblotted with anti-β-catenin, acetyl-lysine (Ac-K), or anti-ubiquitin antibodies. Data information: in (E, F, and G), values are expressed as mean ± SEM; $*p < 0.05$, $**p < 0.01$; no marks, not significant. Source data are available online for this figure.

SIRT1 is involved in an age-associated decrease of the Wnt signaling.

SIRT1 and Wnt signaling have complicated multi-level interactions (Bartoli-Leonard et al, 2018; Holloway et al, 2010; Lu et al, 2017; Simic et al, 2013; Wu et al, 2017). Particularly, β-catenin is an acetylated protein known to be a deacetylation target of SIRT1 (Bartoli-Leonard et al, 2018; Simic et al, 2013). However, SIRT1-mediated deacetylation of β-catenin has been reported to have distinct outcomes on Wnt signaling in different experimental systems. For instance, during mesenchymal stem cell (MSC) differentiation, deacetylation of β-catenin by SIRT1 promotes its accumulation in the nucleus, leading to transcription of genes for MSC differentiation (Simic et al, 2013). Further, in primary mouse embryonic fibroblasts expressing a hyperactive mutant SIRT1, the binding of β-catenin to its target *Gata3* promoter is enhanced despite reduced protein levels (Lu et al, 2017). Yet in vascular calcification, SIRT1 deacetylates β-catenin to reduce its stability and activity, dampening Wnt signaling (Bartoli-Leonard et al, 2018). Our observations in SIRT1 KO Paneth cells suggested that SIRT1 normally inhibits Wnt/β-catenin signaling in intestinal epithelial

     

cells by reducing Wnt3 expression and the nuclear accumulation of β-catenin. To test whether SIRT1 can indeed cell-autonomously suppress Wnt/β-catenin signaling in intestinal epithelial cells, we utilized a previously generated SIRT1 KO DLD1 human colorectal cancer cell line (Ren et al, 2017). Deletion of SIRT1 in DLD1 cells significantly increased the expression of *WNT3* but not the gene encoding β-catenin, *CTNNB1* (Fig. 3H). However, the protein levels of β-catenin were dramatically elevated (Fig. 3I), indicating that SIRT1 post-transcriptionally suppresses β-catenin protein. To test whether SIRT1 reduces the stability of β-catenin through deacetylation, WT and SIRT1 KO DLD1 cells cultured in the regular RPMI medium were treated with 5 μM TSA to and 10 μM MG-132 for 3 h to inhibit other HDACs and proteasome-induced protein degradation, respectively. β-catenin in treated cells were then immunoprecipitated with anti-β-catenin antibodies (IP-β-catenin) and immunoblotted with anti-β-catenin or anti-acetyl-lysine (Ac-K) antibodies. Indeed, this treatment eliminated the β-catenin protein abundance difference between WT and SIRT1 KO cells (Fig. 3J, β-catenin). Moreover, in SIRT1 KO cells, the elevation of acetylated β-catenin protein (Fig. 3J, Ac-K) was coupled with the reduction of ubiquitination levels (Fig. 3J, Ubiquitination). Therefore, in the gut epithelial cells, SIRT1 cell-autonomously suppresses Wnt/β-catenin signaling by inhibiting Wnt3 expression and reducing acetylation and stability of β-catenin.

To further dissect the mechanisms underlying Paneth cell SIRT1 deficiency-induced activation of Paneth cells, we employed a modified Single-Cell Regulatory Network Inference and Clustering (SCENIC) analysis, pySCENIC (Aibar et al, 2017), to infer the activity of different transcription factors in each cell in our scRNA-seq dataset. Consistent with observations in Fig. 3, in Paneth cells from young mice, the top transcription factors that were significantly activated by deletion of SIRT1 included KAISO and TCF4 (Appendix Fig. S2A), both of which are involved in regulation of the Wnt signaling. Interestingly, cluster analysis of activity profiles from the top 100 transcription factors showing significant activity changes across different IECs revealed that Paneth cells and intermediate ISCs in aged PKO mice more closely resembled their counterparts in young mice than those in aged Flox mice (Appendix Fig. S2B), suggesting that deletion of Paneth cell SIRT1 specifically protects these two cell types against age-induced transcriptional changes. Furthermore, the pySCENIC analysis identified ATF4 as the top altered transcription factor in Paneth cells (Appendix Fig. S2A,B). Specifically, ATF4 had the highest transcriptional activity in Paneth cells compared to other IECs and was one of the most active transcription factors in Paneth cells compared to other transcription factors (Appendix Fig. S2A–C). More importantly, its activity was significantly induced in both young and aged SIRT1 KO Paneth cells (Fig. 4A; Appendix Fig. S2B,C). GO analysis of significantly altered ATF4 target genes in SIRT1 KO Paneth cells showed that they were enriched in pathways involved in unfolded protein and ER stress response (Fig. 4B; Appendix Table S2), strongly suggesting that the induced expression of ER stress response genes observed in these cells (Fig. 2A,B; Appendix Fig. S1A) is mediated by an increased ATF4 activity. In agreement with this possibility, deletion of SIRT1 in DLD1 cells increased the induction of ATF4 target genes, *CHOP* and *ASNS*, in response to various culture conditions known to induce ER stress, including brefeldin A treatment, glucose (Glu) depletion, serum depletion and/or glutamine (Gln) depletion

(Fig. 4C). This increased induction of ATF4 target genes in SIRT1 KO DLD1 cells was coupled with reduced cell death upon ER stress (Fig. 4D).

SIRT1 has been reported to suppress ATF4 via both transcriptional and the post-transcriptional mechanisms. In cells challenged with proteasome inhibitor-induced unfolded protein response and ER stress, SIRT1 reduces ATF4 protein without affecting its mRNA (Woo et al, 2013). Yet in cardiac myoblasts treated with palmitic acids, SIRT1 inhibits the mRNA of *ATF4* and its target *CHOP* (Yang et al, 2022a). In brown adipose tissue, SIRT1 overexpression alleviates ER stress-induced apoptosis by inhibiting Smad3 and ATF4 (Liu et al, 2017). We found that when cultured in either regular or glucose-free (Glu-free) RPMI media, SIRT1 KO DLD1 cells had elevated levels of ATF4 protein but not mRNA compared to WT DLD1 cells (Fig. 4C,E). These findings suggest that similar to β-catenin, SIRT1 also post-translationally reduces ATF4 protein, possibly through proteasome-mediated protein degradation in these cells. Consistently, WT DLD1 cells had comparable ATF4 protein levels to those of SIRT1 KO cells after MG-132 and TSA treatment (Fig. 4F, ATF4), along with the reduction of ubiquitination of ATF4 (Fig. 4F, Ubiquitination). However, the acetylation of ATF4 protein was not affected by SIRT1 KO (Fig. 4F, Ac-K). Thus, SIRT1 promotes degradation of ATF4, thereby suppressing ATF4 signaling in the gut epithelial cells.

The elevated ATF4 activity observed in SIRT1 KO cells may explain their increased expression of antimicrobial peptides. In SIRT1 KO DLD1 cells, mRNA levels of two tested defensin genes, *DEFA6* and *DEFB4A*, were higher when cultured in regular medium and became further induced by glucose depletion compared to WT DLD1 cells (Fig. 4G). In small intestinal organoids from Flox mice, glucose depletion led to increased expression of *Atf4* and its target gene *Chop*, while suppressing the expression of anti-microbial genes *Lyz1*, *Defa1*, and *Defa5* (Fig. 4H, Flox, Glu free vs regular medium). In small intestinal organoids from PKO mice, glucose depletion led to a higher induction of *Atf4* and *Chop*, and this increase was coupled with a reduced suppression of anti-microbial genes (Fig. 4H, PKO, Glu free vs regular medium), indicating enhanced resistance to ER stress. Taken together, these findings demonstrate that deletion of SIRT1 in Paneth cells enhances the Wnt/β-catenin signaling and cell-autonomously activates an ATF4/ER stress response, both of which could contribute to the observed protection to age-induced dysregulation of intestinal tissue homeostasis.

## Paneth cell SIRT1 regulates the gut microbiota

The increased expression of anti-microbial peptides in the small intestine (Fig. 2E,F) in SIRT1 PKO female mice suggested that these mice may have altered gut microbiota. Indeed, 16S rRNA gene amplicon sequencing analysis revealed a profound change of their small intestinal mucosal bacteria (Fig. 5A–C; Appendix Fig. S3A and Dataset EV3). Notably, the small intestinal bacteria from SIRT1 PKO females, but not males, had significantly reduced alpha and beta diversities (Fig. 5A,B), indicating that deletion of Paneth cell SIRT1 results in fewer, less balanced, and less complex mucosal bacteria in the small intestine. Specifically, SIRT1 PKO females displayed a substantial expansion of the Bacilli class in the small intestine (Fig. 5C). In particular, the fraction of the

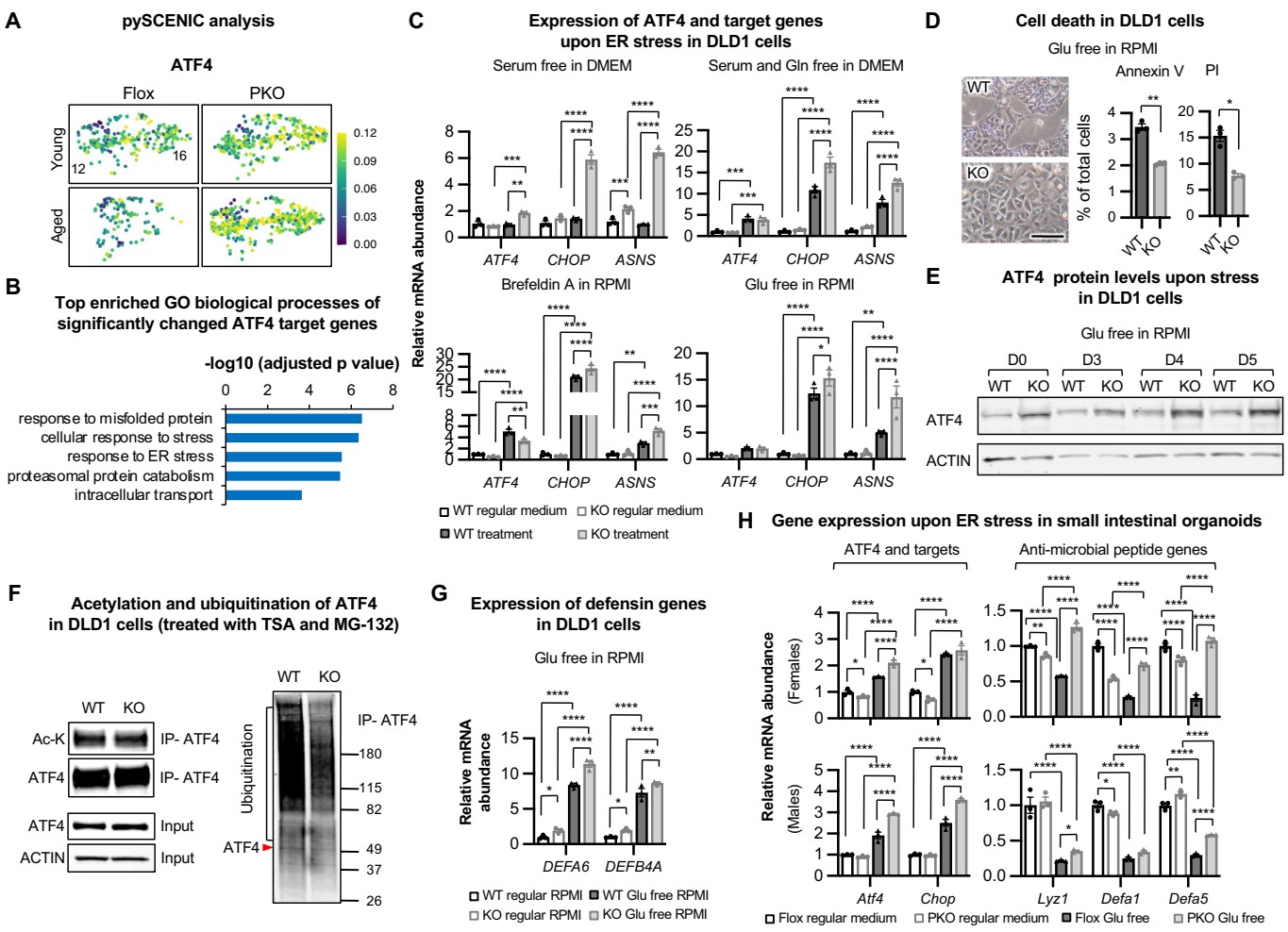

**Figure 4. Deletion of SIRT1 activates ATF4/ER stress response in Paneth cells.**

(A) The transcription activity of ATF4 is increased in the SIRT1 KO Paneth cells. All the combined promoters of the genes expressed in a cell were used to infer the activity of a transcription factor by pySCENIC, and the activity of ATF4 in Paneth cells was projected onto tSNE space with its activity color coded. (B) ER stress response pathways are enriched in ATF4 target genes that are significantly increased in SIRT1 KO Paneth cells (Permutation testing with the Benjamini–Hochberg adjusted p-values). Enrichment score represents $-\log_{10}$-transformed adjusted $p$-values. (C) SIRT1 KO colon epithelial cells have increased response to ER stresses. WT and SIRT1 KO DLD1 cells were treated with indicated stress conditions for 24 h. The expression of indicated genes was analyzed by qPCR ($n = 3$ biological replicates, two-way ANOVA). (D) SIRT1 KO colon epithelial cells have reduced cell death in response to ER stresses. WT and SIRT1 KO DLD1 cells were cultured in glucose free RPMI medium for 3 days ($n = 3$ biological replicates, Student's t-test). Bar, 100 μm. (E) SIRT1 KO DLD1 cells have increased induction of ATF4 protein during glucose free medium induced ER stress. (F) ATF4 is hypo-ubiquitinated in SIRT1 KO colon epithelial cells. WT and SIRT1 KO DLD1 cells cultured in regular RMPI medium were treated with 5 μM TSA and 10 μM MG-132 for 3 h before harvesting. Total cell lysates were immunoprecipitated with anti-ATF4 antibodies (IP-ATF4), then immunoblotted with anti-ATF4, acetyl-lysine (Ac-K), or anti-ubiquitin antibodies. (G) SIRT1 KO DLD1 cells have increased expression of defensin genes ($n = 3$ biological replicates, two-way ANOVA). (H) Small intestinal organoids from PKO mice have increased induction of ATF4 targets and reduced suppression of anti-microbial peptide genes in response to ER stress. Small intestinal organoids from Flox and PKO mice were cultured in regular medium or glucose free medium overnight. The expression of indicated genes was analyzed by qPCR ($n = 3$ biological replicates/group, two-way ANOVA). Data information: in (C, D, G, and H), values are expressed as mean ± SEM; *$p < 0.05$, **$p < 0.01$, ***$p < 0.001$, ****$p < 0.0001$; no marks, not significant. Source data are available online for this figure.

*Lactobacillus* genus elevated significantly from under 40% to over 60% of total bacterial counts, with *L. intestinalis* increased by more than two-fold (Fig. 5D; Appendix Fig. S3B). Further analyses of colonic mucosal and fecal bacteria showed that they were also impacted by deletion of Paneth cell SIRT1 in female mice (Fig. EV3; Dataset EV4 and EV5). For instance, feces from SIRT1 PKO female mice had a significantly reduced abundance of the Verrucomicrobiae class bacteria but a significant increase of the Clostridia class (Fig. EV3F and Dataset EV5). At the genus level, the fraction of *Akkermansia* was significantly reduced yet the

fraction of *Lactobacillus* was significantly increased (Fig. EV3G and Dataset EV5). It is worth noting that the most pronounced changes of microbiota were observed in the small intestine rather than in the colon or feces (Figs. 5 and EV3), supporting the notion that the gut microbiota alternations in SIRT1 PKO mice originate from the increased anti-microbial peptide production in the small intestine.

As the main source of antimicrobial peptides in the intestine to shape the gut microbiota and maintain the gastrointestinal barrier, Paneth cells can directly sense gut commensals through

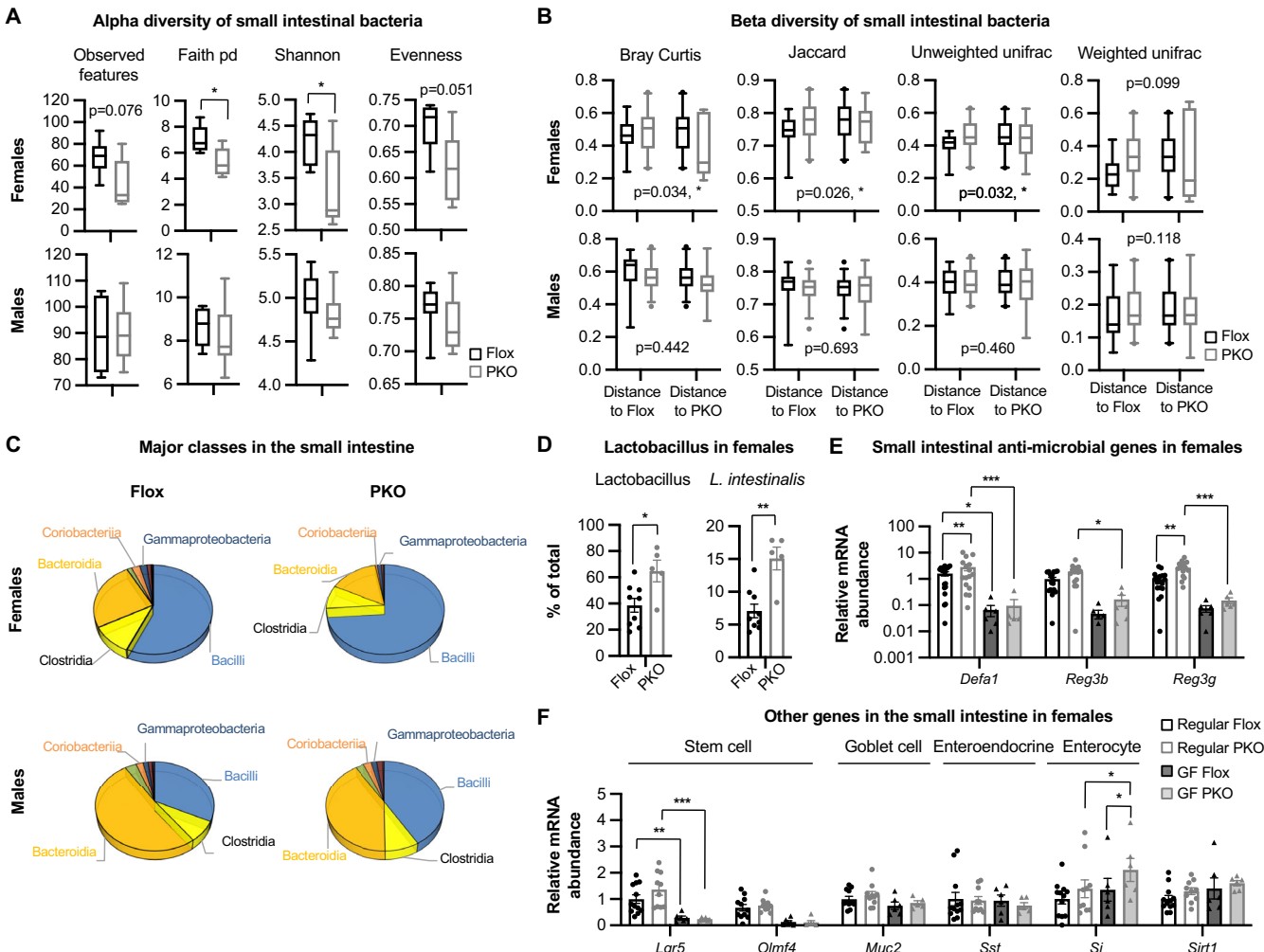

**Figure 5. Paneth cell SIRT1 modulates the interaction between Paneth cells and the small intestinal microbiota in female mice.**

(A–D) SIRT1 PKO female mice have altered small intestinal microbiota. Total small intestinal DNA from Flox and SIRT1 PKO mice were analyzed for mucosal adherent microbiota using 16S rRNA gene amplicon sequencing as described in Methods ($n = 10$ Flox and 8 PKO females and 10 Flox and 10 PKO males). (A) Alpha diversity of small intestinal bacteria (Student's t-test). (B) Beta diversity of small intestinal bacteria (Pair-wise permanova test). (C) SIRT1 PKO females have altered levels of several major classes of small intestinal microbiota. (D) SIRT1 PKO females have increased abundance of Lactobacillus in the small intestine ($n = 10$ Flox and 8 PKO females, Student's t-test). (E) Depletion of microbiota reduces the expression of anti-microbial peptides in the ileum in both Flox and PKO mice. The expression of indicated genes in the ileum of regular and germ-free (GF) Flox and PKO mice were analyzed by qPCR ($n = 18$ regular Flox, 16 regular PKO, 6 GF Flox, and 6 GF PKO mice, two-way ANOVA). (F) The effect of microbiota on the expression of other intestinal epithelial cell markers. The expression of indicated genes in the ileum of regular and germ-free (GF) Flox and PKO mice were analyzed by qPCR ($n = 12$ regular Flox, 10 regular PKO, 6 GF Flox, and 6 GF PKO mice, two-way ANOVA). Data information: in (A and B), box-and-whisker plot with the box representing the interquartile range (Q1 to Q3), a line inside indicating the median, and whiskers representing the 2.5–97.5 percentile; in (D, E, and F), values are expressed as mean ± SEM; *$p < 0.05$, **$p < 0.01$, ***$p < 0.001$; no marks, not significant. Source data are available online for this figure.

cell-autonomous MyD88-dependent toll-like receptor (TLR) activation, which in turn triggers the expression of antimicrobial genes (Ayabe et al, 2000; Vaishnava et al, 2008). Conversely, gut microbiota is also important for the development and maturation of the intestinal epithelium, particularly the Paneth cell differentiation from ISCs (Kim et al, 2022; Sommer and Backhed, 2013). Therefore, the observed changes of intestinal tissue homeostasis in SIRT1 PKO mice, including increase of Paneth cell abundance and resistance to age-induced depletion of ISCs and increase of intraepithelial immune cells, could be resulted from the reciprocal interaction between Paneth cells and the gut microbiota.

To test this possibility, we derived germ-free (GF) Flox and SIRT1 PKO mice, then compared their small intestinal epithelial homeostasis side-by-side with their conventional counterparts. Compared to age and gender matched conventional mice, both GF Flox and GF PKO mice had dramatically reduced mRNA levels of multiple anti-microbial peptide genes (Fig. 5E). Consistently, lysozyme+ Paneth cells in the small intestinal epithelium were not detectable in many crypts in GF mice (Appendix Fig. S3C). Importantly, the difference in anti-microbial peptides between conventional SIRT1 PKO mice and Flox control mice was completely abolished in the GF condition (Fig. 5E), indicating that Paneth cell SIRT1-mediated regulation of Paneth cells requires the

gut microbiota. Intriguingly, the absence of microbiota also massively reduced the expression of ISC marker genes, particularly *Lgr5*. In contrast, the expression of markers of several other intestinal epithelial cells, including goblet cells (*Muc2*), enteroendocrine cells (*Sst*), and enterocytes (*Si*), was not significantly affected by the status of microbiota in Flox mice (Fig. 5F). Together, our data indicate that Paneth cell SIRT1 modulates the gut microbiota.

## Paneth cell SIRT1 deficiency protects mice from chemically induced colitis through modulating the gut microbiota

Disruption of the balance of all major genera of gut commensal bacteria that are significantly affected in SIRT1 PKO mice, including *Lactobacillus*, *Muribaculaceae*, and *Akkermansia*, has been frequently associated with inflammatory bowel diseases (IBDs), metabolic diseases, intestinal carcinogenesis, and the therapeutic outcomes of these diseases (Borruel et al, 2002; Llopis et al, 2009; Longhi et al, 2020; Parker et al, 2020; Saus et al, 2019; Zagato et al, 2020; Zhong et al, 2014; Zhu et al, 2024). The interaction of these bacteria with gut epithelium and host immune system is complex and context-dependent. For instance, *Akkermansia* is generally anti-inflammatory, however, in the setting of IEC-associated ER stress, it can be proinflammatory (Matute et al, 2023). To directly test whether the alteration of gut microbiota observed in SIRT1 PKO mice could impact intestinal tissue homeostasis and inflammation, we first investigated whether Flox and SIRT1 PKO mice had differential responses to chemically induced intestinal inflammation by challenging them with 2.5% Dextran Sodium Sulfate (DSS) in drinking water to induce colitis. In a DSS-colitis protocol where young mice at the age of 3 months were treated for 5–7 consecutive days of DSS to induce acute colitis, SIRT1 PKO mice were indeed protected from DSS-induced body weight loss, rectal bleeding, and colonic crypt erosion compared to Flox mice (Fig. 6A–D). Histopathological evaluation further revealed that compared to DSS-treated Flox mice, DSS-treated SIRT1 PKO mice had significantly reduced inflammation, goblet cell loss, and granulation tissue, thereby overall pathological severity, in their colons (Fig. 6E). In a second DSS-colitis protocol in which 9-month-old mice were challenged with two cycles of 7-day DSS drinking water separated by a 14-day regular water break to allow tissue recovery between DSS treatments and to induce chronic inflammation (Fig. EV4A), SIRT1 PKO mice experienced significantly reduced rectal bleeding (Fig. EV4B) and colonic shortening (Fig. EV4C). They also displayed a trend of reduction in ulceration, goblet cell loss, and granulation tissue in the colon (Fig. EV4D,E). However, surprisingly, both their colon and ileum had increased immune cell infiltration compared to Flox control mice (Fig. EV4D,F). A pilot RNA-seq analysis in conjunction with subsequent qPCR validation confirmed that the colon of DSS-treated 9-month-old SIRT1 PKO mice had a significant increase in expression of many immune cell genes, along with several genes involved in ion transport and protein metabolism (Figs. 6F and EV4G; Dataset EV6). In the ileum of DSS-treated 9-month-old SIRT1 PKO mice, in addition to increased expression of multiple immune cell genes, a few genes involved in cell cycle and cell proliferation were significantly increased, together with elevated ISC and Paneth cell markers

(Figs. 6G and EV4H). Therefore, Paneth cell SIRT1 deficiency-induced protection against DSS-colitis was intriguingly associated with increased immune cell infiltration together with enhanced gut epithelial cell proliferation/recovery. These seemingly contradictory phenotypes suggest that SIRT1 PKO mice are not broadly immunosuppressive. Instead, they likely reprogramed the immune response toward regulation, resolution, and repair rather than unchecked inflammation in this chronic colitis model. Particularly, the elevation of *Cd19*, *Foxp3*, and *Il10* in the colon (Fig. 6F) suggests activation of regulatory B cells and Treg cells, which are known to suppress inflammation, protect against tissue injury, and promote tissue repair (Li and He, 2004; Wang et al, 2015; Yanaba et al, 2011). Furthermore, as a subset of intestinal macrophages is known to secret Wnt ligands to maintain the mesenchymal niche cells important for Paneth cell differentiation (Kim et al, 2022), it is possible that the increased gut immune cells in DSS-treated SIRT1 PKO mice induces the Wnt signaling in crypt cells (ISCs and Paneth cells), contributing to observed increase in tissue repair. In line with this possibility, AI-assisted image analysis showed that the nuclear accumulation of β-catenin was significantly enhanced in both ISCs and Paneth cells in the ileum of DSS-treated SIRT1 PKO mice (Figs. 6H,I and EV4I), and the expression of *c-Myc*, a Wnt/β-catenin target gene, was also significantly increased (Fig. 6G, *c-Myc*).

We next investigated whether Paneth cell SIRT1 deficiency-induced alteration of gut microbiota plays a role in observed protection against DSS-colitis by depleting gut microbiota in Flox and SIRT1 PKO mice with four weeks of treatment of an antibiotic cocktail containing drinking water (Abx). We then treated them with DSS. Compared to Flox and SIRT1 PKO mice in the conventional condition (Regular), Abx-treatment increased resistance to DSS-induced colitis in Flox mice, but not further in PKO mice (Fig. 7A–D, Abx vs Regular). Consistently, the expression levels of several proinflammatory cytokines were reduced in both colon and ileum of antibiotic-treated DSS-colitis Flox mice compared to conventional DSS-colitis Flox mice (Fig. 7E, Abx Flox vs Regular Flox), and there was no difference between PKO and Flox mice after the antibiotic treatment (Fig. 7E, Abx PKO vs Abx Flox). These observations suggest that the highly expressed anti-microbial peptides in PKO mice may eliminate certain colitogenic gut microbiota species, thereby mimicking Abx-induced protection against DSS-colitis. Therefore, Paneth cell SIRT1 deficiency induced protection against DSS-colitis through modulating the gut microbiota.

Finally, to directly assess the importance of the gut microbiota in mediating Paneth cell SIRT1 deficiency-induced protection against DSS-colitis, we performed a fecal transplantation (FT) experiment in which GF Flox mice were transplanted with fecal microbes from regular Flox mice (Flox FT Flox) or PKO mice (Flox FT PKO), and GF PKO mice were transplanted with fecal microbes from regular Flox mice (PKO FT Flox) or PKO mice (PKO FT PKO) (Fig. EV5A). Ten days after the transplantation, these mice were treated with 2.5% DSS to induce acute colitis. Interestingly, 16S rRNA amplicon sequencing analysis of the fecal bacteria of recipient mice revealed that fecal transplantation switched the abundance of an uncharacterized (uc) genus of *Muribaculaceae*, as well as *Lactobacillus* (Figs. 7F and EV5B). However, it did not switch the levels of *Akkermansia* and other bacteria between Flox and SIRT1 PKO mice (Fig. EV5B), possibly due to the

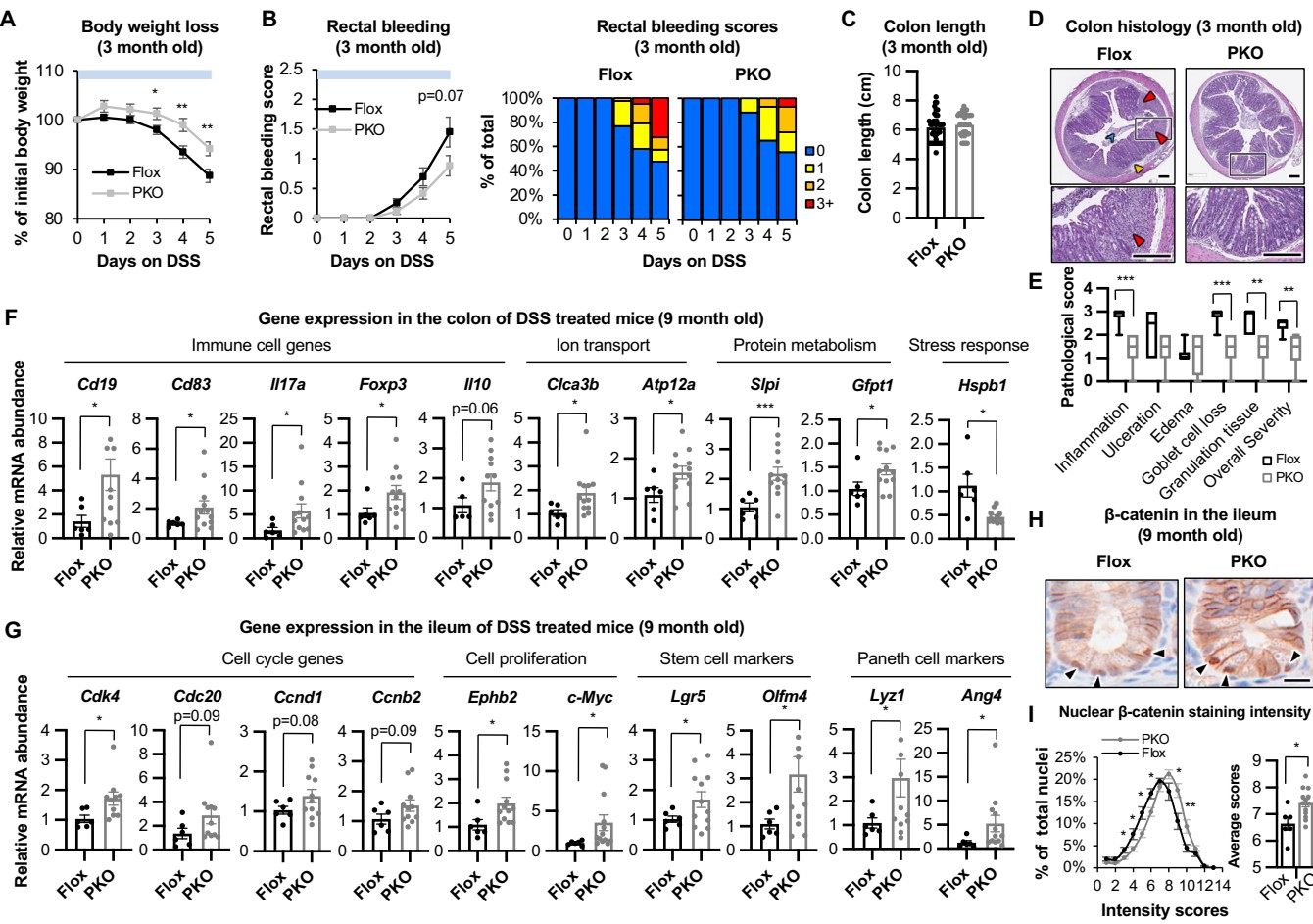

**Figure 6. SIRT1 PKO mice are protected from DSS-induced colitis.**

(A–C) Young SIRT1 PKO mice are more protected against DSS-induced colitis compared to Flox controls. Three-month-old Flox and SIRT1 PKO mice were treated with 2.5% DSS in drinking water for 5–7 days. Their body weight (A), rectal bleeding (B), and colon length (C) were analyzed ($n = 43$/group, Student's t-test). (D, E) Young SIRT1 PKO mice have reduced pathological severity in the colon after DSS treatment. Three-month-old Flox and SIRT1 PKO mice were treated as in (A). The histopathological severity was evaluated by a professional pathologist as described in Methods ($n = 6$ Flox and 8 PKO, Student's t-test). Red arrowheads, segmental mucosal ulceration and consequent replacement with granulation tissue; orange arrowhead, submucosal edema; blue arrowhead, sloughed off necrotic and cellular debris within the lumen. Bars, 200 µm. (F) Altered gene expression in the colon of mice after 2 cycles of DSS treatment. Nine-month-old Flox and SIRT1 PKO mice were treated with 2 cycles of DSS as described in Methods and the expression of indicated genes were analyzed by qPCR ($n = 6$ Flox and 12 PKO mice, Student's t-test). (G) The ileum of DSS-treated 9-month-old SIRT1 PKO mice have increased expression of genes involved in cell cycle, cell proliferation, as well as increased levels of stem cell and Paneth cell genes. The expression of indicated genes was analyzed by qPCR ($n = 6$ Flox and 12 PKO mice, Student's t-test). (H, I) The ileum of 9-month-old SIRT1 PKO mice treated with 2 cycles of DSS water have increased β-catenin signaling. (H) The expression of β-catenin was analyzed by IHC. Arrowheads, nuclear β-catenin in Paneth cells. Bar, 20 µm. (I) The nuclear β-catenin intensity in the epithelial tissue was quantified by AI as described in Methods ($n = 6$ Flox and 12 PKO mice, Student's t-test). Data information: in (A, B, C, F, G, and I), values are expressed as mean ± SEM; in (E), box-and-whisker plot with the box representing the interquartile range (Q1 to Q3), a line inside indicating the median, and whiskers representing the 2.5–97.5 percentile; *$p < 0.05$, **$p < 0.01$, ***$p < 0.001$; no marks, not significant. Source data are available online for this figure.

hypersensitivity of these bacteria to the anti-microbial peptides highly produced in SIRT1 PKO mice. Yet this partial switch was sufficient to exchange the sensitivity of mice to DSS colitis, as fecal microbiome from SIRT1 PKO mice protected Flox control mice from DSS-induced rectal bleeding and tissue damage, whereas fecal microbiome from Flox control mice rendered sensitivity to DSS in SIRT1 PKO mice (Figs. 7H,I and EV5C,D). Collectively, these findings suggest that Paneth cell SIRT1 mediates DSS-induced stress response in the gut through modulating gut *Muribaculaceae* and/or *Lactobacillus*.

## Low Paneth cell SIRT1 is associated with increased Paneth cell abundance in patients of Crohn's disease

To further understand whether Paneth cell SIRT1 regulates Paneth cell abundance and activity in the context of human IBD, we analyzed a published scRNA-seq dataset from a study that sequenced cells from the terminal ileum and colon of Crohn's disease (CD) patients (Data ref: (Kong et al, 2023). From this dataset, about 3.1% of epithelial cells from the terminal ileum were Paneth cells (Dataset EV7A). We were also able to detect a small

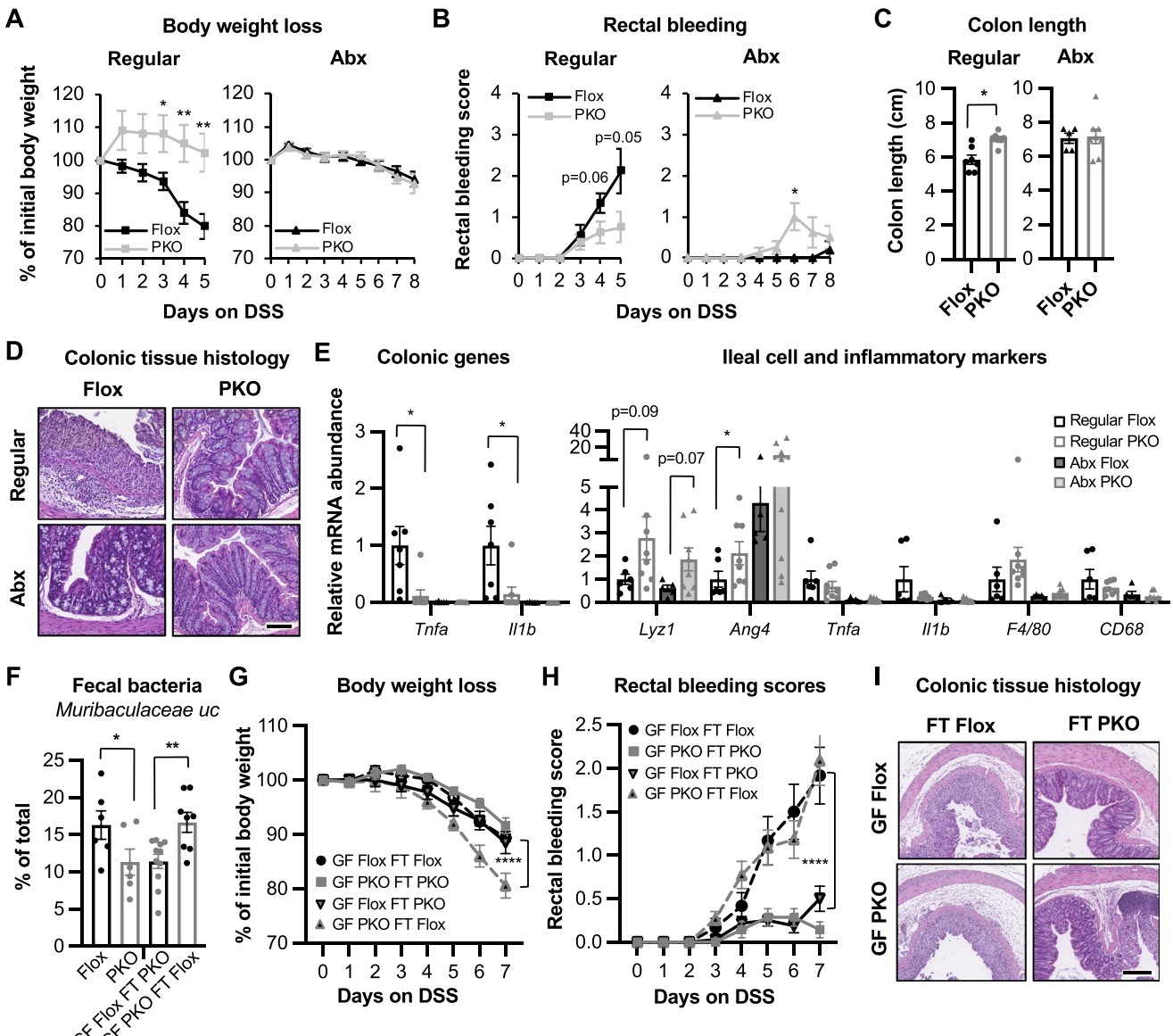

**Figure 7. Paneth cell SIRT1 regulates the sensitivity to DSS-colitis through gut microbiota.**

(A–D) Antibiotics (Abx)-mediated depletion of gut microbiota ameliorates DSS-induced colitis in Flox but not PKO mice. Flox and PKO mice were treated with an antibiotic cocktail (Abx) for 4 weeks then with 2.5% DSS for 8 days (Regular: $n = 9$ Flox and 8 PKO; Abx: $n = 5$ Flox and 8 PKO; Student's t-test). Bar in (D), 100 μm. (E) Gene expression in the colon and ileum of regular and Abx mice after the 7-day DSS-treatment. Flox and SIRT1 PKO mice were treated as in Fig. 5A (Regular, $n = 6$ Flox and 8 PKO mice) or 6A (Abx, $n = 5$ Flox and 8 PKO mice). Student's t-test. (F) Fecal transplantation switches the abundance of *Muribaculaceae uc*. Fecal transplantation was performed as described in Methods and fecal microbiota were analyzed by 16S rRNA amplicon sequencing ($n = 6, 6, 11$, and 8 mice; two-way ANOVA). (G–I) Mice transplanted with fecal microbes from PKO mice are protected from DSS-induced rectal bleeding and colonic tissue damage. GF Flox and PKO mice were transplanted with fecal microbes from either regular Flox mice or PKO mice, then treated with 2.5% DSS ($n = 6$ GF Flox FT Flox, 7 GF PKO FT PKO, 16 GF Flox FT PKO, and 11 GF PKO FT Flox; two-way ANOVA). Bar in (I), 200 μm. Data information: in (A, B, C, E, F, G, and H), values are expressed as mean ± SEM; *$p < 0.05$, **$p < 0.01$, ****$p < 0.0001$; no marks, not significant. Source data are available online for this figure.

number of Paneth cells (about 0.32%) in the colon, particularly in inflamed and non-inflamed colon samples from these CD patients (Dataset EV7A), presumably due to colorectal Paneth cell metaplasia (Tanaka et al, 2001).

The expression of SIRT1 has been previously reported to be reduced in the colonic biopsies from both UC and CD patients (Caruso et al, 2014; Deng et al, 2020; Wellman et al, 2017). Consistent with these reports, in the terminal ileum, the average

SIRT1 mRNA levels were significantly reduced in inflamed samples compared to healthy and non-inflamed samples when all ileal epithelial cells were considered together (Fig. 8A). When analyzed in individual cell types, Paneth cells, like many other different cell types in the terminal ileum, had significantly reduced average SIRT1 mRNA levels in inflamed samples (Fig. 8B). In the colon, although the average SIRT1 mRNA levels were surprisingly higher in both inflamed and noninflamed samples from CD patients

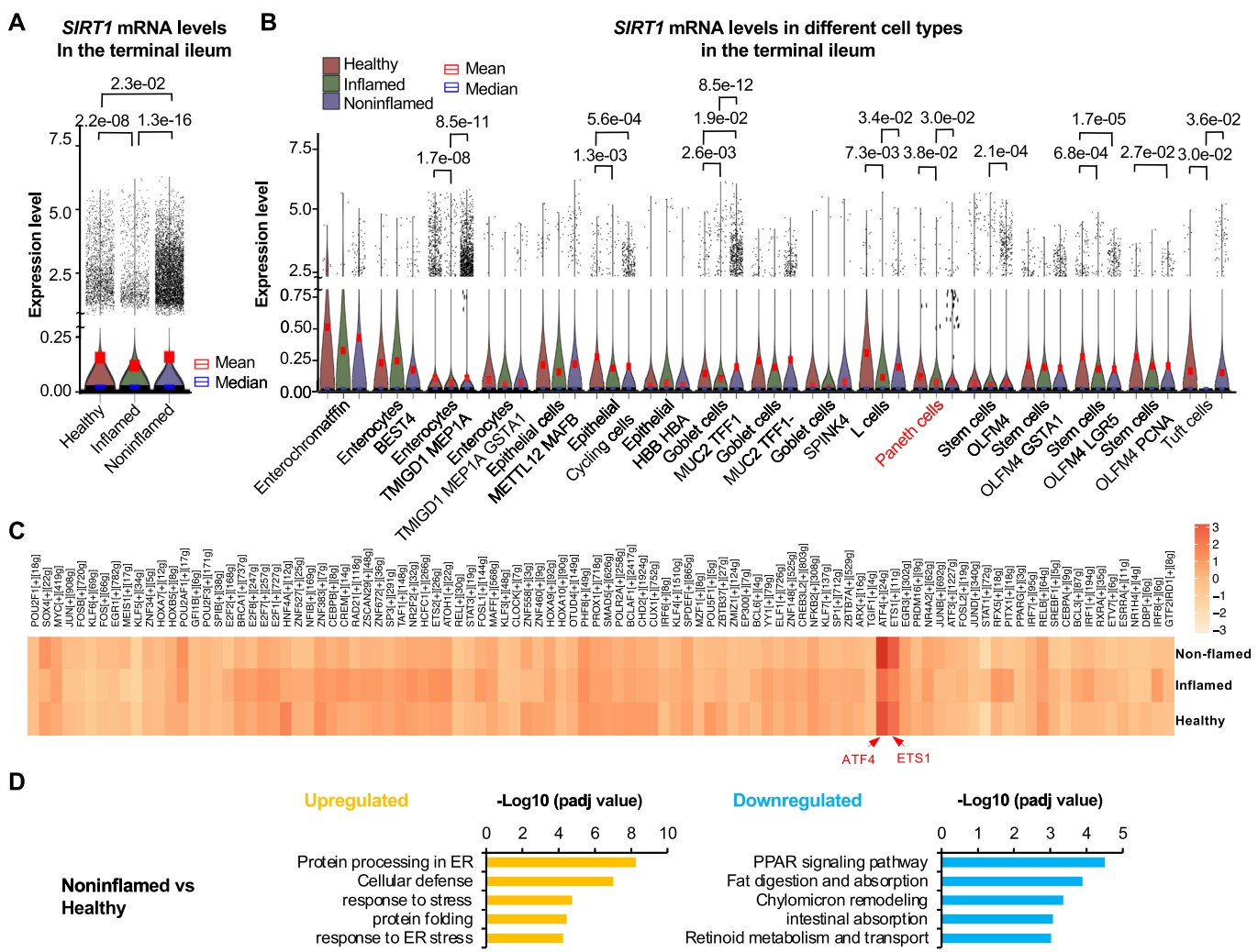

**Figure 8. Paneth cells from non-inflamed terminal ileum of human CD patients have reduced *SIRT1* and increased ATF4 activity.**

scRNA-seq datasets (Accession DUOS-000146 CD_Atlas_2021_GIDER; DUOS-000145 CD_Atlas_2021_PRISM) of intestinal epithelial cells from healthy donors or CD patients was analyzed. (A) The expression levels of *SIRT1* in the terminal ileum were compared between Healthy controls, inflamed, and non-inflamed CD patients (two-sided Wilcoxon test, and adjusted p-values were obtained using the Holm–Bonferroni correction). (B) The expression of *SIRT1* in different cell types in the terminal ileum (two-sided Wilcoxon test, and adjusted *p*-values were obtained using the Holm–Bonferroni correction). (C) pySCENIC analysis of transcription activity of TFs was performed using above scRNA-seq datasets and top 100 TFs altered in indicated Paneth cells from the terminal ileum. (D) The top enriched functional pathways in the Paneth cells isolated from Healthy and Noninflamed terminal ileum of CD patients. Paneth cell DEGs between Non-inflamed vs Healthy were analyzed by g:Prolifer and top enriched pathways are shown (Permutation testing with the Benjamini–Hochberg adjusted *p*-values). Enrichment score represents −log₁₀-transformed *p*-values after adjusted for FDR false discovery rate.

compared to healthy controls, the inflamed samples still showed significantly lower *SIRT1* mRNA levels than the noninflamed samples (Appendix Fig. S4A). Interestingly, unlike other cell types in the colon, Paneth cells were the only cells displaying a trend of reduced average *SIRT1* mRNA levels in CD samples compared to healthy control samples (Appendix Fig. S4B), albeit that the difference was not significant due to the very small number of Paneth cells in healthy control samples (*n* = 13 cells, Dataset EV7A). Therefore, in the colon of CD patients analyzed in this study, SIRT1 is differentially affected in Paneth cells compared to other cell types. Indeed, Paneth cells were the only cell type in which the mRNA levels of *SIRT1* were reduced in both terminal ileum and colon from CD patients regardless of their inflammatory status

(Fig. 8B; Appendix Fig. S4B, Paneth cells). More importantly, the reduction of *SIRT1* mRNA in Paneth cells was associated with increased Paneth cell fractions in both tissues in CD patients (Dataset EV7A, Paneth cells). This observation is consistent with our observation that deletion of Paneth cell SIRT1 in mice increases Paneth cell abundance (Fig. 1).

To gain better insight on whether and how Paneth cell SIRT1 may regulate Paneth cell activity in humans, we focused on the terminal ileum in this dataset. This region contained a significant proportion of Paneth cells in the healthy control samples, unlike the colon, where only 13 Paneth cells were detected in the healthy controls (Dataset EV7A). This allowed us to draw more reliable conclusions. As shown in Fig. 8C, pySCENIC analysis identified

ATF4 as the top transcription factor whose activity was reduced in ileal Paneth cells from inflamed samples compared to those from healthy and non-inflamed samples. Another gene that showed a similar pattern was ETS1, a proto-oncogene reported to be induced in UC but reduced in CD (Konno et al, 2004) and an upstream activator of Wnt/β-catenin (Gu et al, 2019). However, because Paneth cells from both inflamed and non-inflamed ileum have reduced expression of *SIRT1* (Fig. 8B), the role of Paneth cell SIRT1 in intestinal inflammation of CD patients was unclear, which we considered as a confounding variable. We therefore compared Paneth cells isolated from noninflamed vs healthy ileum. This comparison revealed that in the absence of intestinal inflammation, low Paneth cell SIRT1 expression was coupled with high ATF4 and ETS1 activities, high response to ER stress, and high levels of cellular defense genes (antimicrobial peptides) in Paneth cells (Dataset EV7B, Fig. 8C,D, Noninflamed vs Healthy).

# Discussion

As the cornerstones of intestinal health, Paneth cells have multiple important functions in preserving gut barrier integrity, including maintaining ISCs through niche factors and shaping the gut microbiome via antimicrobial peptides (Wallaeys et al, 2023). Disruption of one or more of these functions can lead to intestinal and systemic inflammatory and infectious diseases (Clevers and Bevins, 2013; Wallaeys et al, 2023). In the present study, we show that SIRT1, one of the well-known genetic repressors of inflammation, cell-autonomously suppresses Paneth cell activation in the presence of gut commensal bacteria. This action of Paneth cell SIRT1, in turn, modulates the interaction between the intestinal epithelium, immune cells, and gut microbiota, sensitizing the gut epithelium to aging- or chemically-induced stress and inflammation. At the molecular level, we provide evidence that SIRT1 deficiency-induced stress resistance is associated with cell-autonomous enhancement of Wnt/β-catenin signaling and increased ATF4/ER stress response in vivo and in vitro. These findings demonstrate that by cell-autonomously inhibiting activation of Paneth cells, Paneth cell SIRT1 unexpectedly confers sensitivity to stress-induced dysfunction of the whole gut epithelium.

Given the well-documented importance of SIRT1 in protection against a variety of stresses through regulation of metabolism, apoptosis, and inflammation (Schug and Li, 2011; Yamamoto et al, 2007), the observation that deletion of Paneth cell SIRT1 in mice increases their resistance to aging- and chemically-induced stress and inflammation in the gut is surprising. Particularly, we have previously reported that deletion of SIRT1 in the whole intestinal epithelium (SIRT1 iKO) also activated Paneth cells and altered gut microbiota (Wellman et al, 2017). However, in contrast to SIRT1 PKO mice, these changes in SIRT1 iKO mice were associated with an increased sensitivity to stress and inflammation (Wellman et al, 2017). A careful comparison of these two studies revealed that although Paneth cells were hyperactivated in both SIRT1 PKO and SIRT1 iKO mice, the age at which Paneth cell activation first occurred was different in these two mouse models. In SIRT1 iKO mice, Paneth cell activation occurred only in aged but not young mice, and this late-onset activation appeared to have minimal impact on gut microbiota or intestinal inflammation (Wellman et al, 2017). Instead, dysbiosis in aged SIRT1 iKO mice was primarily induced by defective ileal bile acid resorption and

subsequent bile acid accumulation, which *sensitized* the mice to colitis and age-associated inflammation (Wellman et al, 2017). In SIRT1 PKO mice, on the other hand, Paneth cell activation occurred at a young age (Fig. 1) in the absence of bile acid defect (data now shown). This early activation resulted in a distinct gut microbiota composition (Fig. 5), which *increased* gut epithelial resistance to chemically induced colitis (Figs. 6 and EV4). In support of the notion that an early-stage Paneth cell activation could increase intestinal epithelial stress resistance by influencing the gut microbiota, Lo Sasso et al reported that in a mouse model in which exons 5, 6, and 7 of the mouse *Sirt1* gene were deleted specifically in the intestinal epithelium, Paneth, and goblet cells were activated at a young age (Lo Sasso et al, 2014). This early activation of Paneth cells resulted in the rearrangement of distinct groups of gut microbiota and *protected* animals from colitis (Lo Sasso et al, 2014). Therefore, our findings, and previous observations (Lo Sasso et al, 2014; Wellman et al, 2017), uncover an age-dependent impact of Paneth cell activation on gut microbiota and intestinal epithelial homeostasis. Given the substantial literature supporting the importance of microbial interactions with the epithelium during the so-called "window of opportunity" when commensalism and homeostasis of the immune system is being established (Gensollen et al, 2016), we envision these findings could provide valuable insights for identifying a therapeutic window for SIRT1/microbiome-based interventions in intestinal disorders. It will be of great interest to test whether pharmacological suppression of SIRT1 in young animals might boost their gut epithelial stress resistance by activating Paneth cells in future studies.

Our observations in the present study further demonstrate that Paneth cell SIRT1 deficiency protects mice from chemically induced colitis by modulating the gut microbiota (Figs. 7 and EV4). Specifically, our results suggest that the decrease of *Muribaculaceae uc* and/or increase of gut *Lactobacillus* abundance might underlie Paneth cell SIRT1 deficiency-induced gut epithelial resistance to DSS-colitis (Figs. 7 and EV4). While the genus of *Muribaculaceae* we identified in the present study is uncharacterized (thus *uc*), a wealth of literature has shown that *Lactobacillus* are reduced in IBD patients (Ott et al, 2004), and supplementation of *Lactobacillus* can protect against intestinal inflammation in humans and mice (Borruel et al, 2002; Llopis et al, 2009). Intriguingly, microbial dysbiosis induced by antibiotics or other stresses has been reported to suppress Paneth cell differentiation, and *Lactobacillus* transplantation can rescue this defective Paneth cell differentiation (Kim et al, 2022). Therefore, Paneth cell SIRT1 may influence intestinal stress response by direct modulation of *Lactobacillus*, impacting Paneth cell differentiation and activation. How Paneth cell-induced alteration of gut microbiota influences the response of the colon to DSS-colitis in SIRT1 PKO mice is still not fully understood. The possible mechanisms include outcompeting pathogenic bacteria that contribute to inflammation and colitis, modulating immune cells and suppressing their production of inflammatory cytokines upon DSS treatment, and producing beneficial metabolites such as short-chain fatty acids to boost colonocyte function and survival. While our current findings (Figs. 6 and 7) support the first two mechanisms, futher studies are needed to fully elucidate the underlying pathways.

Our analysis showed that in both ileum and colon of CD patients, low Paneth cell SIRT1 was associated with increased Paneth cell abundance (Fig. 8; Appendix Fig. S4). Moreover, in the

     

absence of intestinal inflammation, the low expression of SIRT1 in Paneth cells was coupled with increased ATF4 activity and Paneth cell activity (Fig. 8C,D). However, whether Paneth cell SIRT1 plays a role in regulation of human intestinal inflammation is still unclear. In the future, it will be interesting to test whether the NAD-dependent protein deacetylase activity of SIRT1 is activated in the inflamed gut of IBD patients either through phosphorylation (Guo et al, 2010) or by an increase of the local cellular NAD level.

In summary, our study establishes SIRT1 as a cell-autonomous suppressor of Paneth cell function and unveils an unexpected negative impact of this suppression on the gut epithelial stress resistance. Our findings further highlight the importance of the gut microbiota in mediating the influence of Paneth cell SIRT1 on Paneth cell activity and intestinal epithelial homeostasis and suggest that age-specifically targeting Paneth cells via a combination of SIRT1 inhibition and microbiome manipulation could be a novel therapeutic strategy against intestinal diseases.

# Methods

### Reagents and tools table

| Reagent/Resource | Reference or Source | Identifier or Catalog Number |
|---|---|---|
| **Experimental models** | | |
| WT and SIRT1 KO DLD1 human colorectal cell lines | (Ren et al, 2017) | Original DLD1 cells were from ATCC (cat# CCL-221), routinely tested in the lab for mycoplasma contamination |
| SIRT1 PKO mice, Sirt1$^{f/f}$;Defa6-Cre$^+$ | This study | |
| Flox control mice, Sirt1$^{f/f}$;Defa6-Cre$^-$ | This study (Cheng et al, 2003) | |
| SIRT1 PKO mice on the Lgr5-EGFP background Sirt1$^{f/f}$;Defa6-Cre$^+$; Lgr5-EGFP-IRES-creERT2 | This study | |
| Flox control mice on the Lgr5-EGFP background Sirt1$^{f/f}$;Defa6-Cre$^-$; Lgr5-EGFP-IRES-creERT2 | This study | |
| SIRT1 PHet on the R26-TdTomato background (Sirt1$^{f/+}$;R26-TdTomato$^+$;Defa6-Cre$^+$) | This study | |
| SIRT1 PKO on the R26-TdTomato background (Sirt1$^{f/f}$;R26-TdTomato$^+$;Defa6-Cre$^+$) | This study | |
| Small intestinal organoids from Flox and PKO mice | This study | From both male and females |

| Reagent/Resource | Reference or Source | Identifier or Catalog Number |
|---|---|---|
| **Recombinant DNA** | | |
| **Antibodies** | | |
| CD45-PE | Invitrogen | 12-0451-83 |
| CD31-PE | BioLegend | 102508 |
| Ter119-PE | BioLegend | 116208 |
| CD24-Pacific Blue | BioLegend | 101820 |
| EPCAM-APC | Invitrogen | 17-5791-82 |
| Normal rabbit IgG | Millipore | NI01 |
| Lyzome (rabbit polycolonal) | Dakocytomation Corporation | A0099 |
| Ki67 (rabbit polycolonal) | Abcam | ab-16667 |
| CD11C BV510 | BD | 744178 |
| CD19 CPA | BD | 550992 |
| CD8a BV711 | BIOLEGEND | 100759 |
| CD3 FITC | BIOLEGEND | 100204 |
| CD11b PE/CY5.5 | BIOLEGEND | 101227 |
| IA/IE PE/CY7 | BIOLEGEND | 107629 |
| CD115 PE/594 | BIOLEGEND | 135527 |
| Ly6c APC/CY7 | BIOLEGEND | 128025 |
| CD45 e450 | eBioscience™ | 48-0451-82 |
| Ly6G PE | eBioscience™ | 12-9668-82 |
| F4/80 PE/CY5 | eBioscience™ | 15-4801-82 |
| CD4 AF700 | eBioscience™ | 56-0042-82 |
| GAPDH | Cell Signaling | 2118 |
| ACTIN | EMD,Millipore | MAB1501 |
| SIRT1 | Cell Signaling | 2310 |
| Ubiquitin | Santa Cruz | sc-8017 |
| ATF4 | Cell Signaling | 11815 |
| Acetylated-Lysine | Cell Signaling | 9814 |
| β-catenin | Cell Signaling | 8480 |
| β-catenin | GeneTex | GTX101435 |
| β-catenin (rabbit polycolonal) | Abcam | ab-16051 |
| **Oligonucleotides and other sequence-based reagents** | | |
| Gene name | For 5′ → 3′ | Rev 5′ → 3′ |
| Lyz1 | GAGACCGAAG CACCGACTATG | CGGTTTTGACATTGTG TTCGC |
| Defa1 | TCAAGAGGCTGC AAAGGAAGAGAAC | TGGTCTCCATGTTCAG CGACAGC |
| Ang4 | GGTTGTGATTCCT CCAACTCTG | CTGAAGTTTTCTCC ATAAGGGCT |
| Defa5 | CTCCTCTCTGC CCTTGTCCT | GATTTCTGCAGG TCCCAAAA |
| Lgr5 | CCTACTCGAAGA CTTACCCAGT | GCATTGGGGGTGAA TGATAGCA |
| Olfm4 | CAGCTGCCTGG TTGCCTCCG | GGCAGGTCCCAT GGCTGTCC |

| Reagent/Resource | Reference or Source | Identifier or Catalog Number |
|---|---|---|
| Cdk4 | ATGGCTGCCACT CGATATGAA | TCCTCCATTAGGAA CTCTCACAC |
| Cdc20 | TTCGTGTTCGAGA GCGATTTG | ACCTTGGAACTAGA TTTGCCAG |
| Ccnd1 | GCGTACCCTGAC ACCAATCTC | CTCCTCTTCGCACTTCT GCTC |
| Ccnb2 | GCCAAGAGCCATG TGACTATC | CAGAGCTGGTACTTTGG TGTTC |
| Ephb2 | CCATTGAACAGGAC TACAGACTACC | CACCGTGTTAAAG CTGGTGTAG |
| c-Myc | TCTCCACTCACCAGC ACAACTACG | ATCTGCTTCAGGACCCT |
| Cd19 | GGAGGCAATGTTGT GCTGC | ACAATCACTAGCAA GATGCCC |
| Cd83 | CGCAGCTCTCCTAT GCAGTG | GTGTTTTGGATCG TCAGGGAATA |
| Il17a | TTTAACTCCCTTG GCGCAAAA | CTTTCCCTCCGCATT GACAC |
| Foxp3 | CCCATCCCCAGGAG TCTTG | ACCATGACTAGGGG CACTGTA |
| Il10 | TTTGAATTCCCTGG GTGAGAA | GGAGAAATCGATG ACAGCGC |
| Tnfa | CCCTCACACTCAGAT CATCTTCT | GCTACGACGTGG GCTACAG |
| Il1b | ACCTGTCCTGTGTAA TGAAAGACG | TGGGTATTGCTTGGG ATCCA |
| F4/80 | CTTTGGCTATGGGC TTCCAGTC | GCAAGGAGGACAGAG TTTATCGTG |
| Cd68 | CCAATTCAGGGTGG AAGAAA | CTCGGGCTCTGA TGTAGGTC |
| Clca3b | GCTGCCTCAATACCC AAATGGC | TTGACCTCCTGAAA GCACGAGC |
| Atp12a | ATGCGCCGGAAAAC AGAAATC | CCTCCTCCTGACTC TTGTTGG |
| Slpi | GGCCTTTTACCTTT CACGGTG | TACGGCATTGTGGCTT CTCAA |
| Gfpt1 | GAAGCCAACGCCTG CAAAATC | CCAACGGGTATGAGCT ATTCC |
| Hspb1 | ATCCCCTGAG GGCACACTTA | GGAATGGTGATCTCC GCTGAC |
| Muc2 | TGTGGTCTGTG TGGGAACTTT | CATAGATGGGCCTGTCC TCAGG |
| Sst | ACCGGGAAACAG GAACTGG | TTGCTGGGTTCG AGTTGGC |
| Si | TTCAAGAAATCACAAC ATTCAATTTACCTAG | CTAAAACTTTCTTT GACATTTGAGCAA |
| Sirt1 | GTTGACCTCCTCAT TGTTATTGG | TGAGGCACTTCATGG GGTAT |
| LaminA | AGTGAGAAGCGCA CATTGG | TCAGCATCTCATCCTG AAGC |
| ATF4 | CCCTTCACCTTCTTA CAACCTC | TGCCCAGCTCTAAACT AAAGGA |
| ASNS | GCAGCTGAAAGAAG CCCAAGT | TGTCTTCCATGCCAA TTGCA |

| Reagent/Resource | Reference or Source | Identifier or Catalog Number |
|---|---|---|
| CHOP | GAACGGCTCAAGCA GGAAATC | TTCACCATTCGG TCAATCAGAG |
| WNT3 | CCTCGGCGCCTCTT CTAATGGA | CCAGAGATGTGT ACTGCTGGCCC |
| CTNNB1 | AGCTTCCAGACACGC TATCAT | CGGTACAACGAGCTG TTTCTAC |
| DEFA6 | CTGAGCCACTCCAA GCTGAG | GTTGAGCCCAAAGCTCT AAGAC |
| DEFB4A | CTCCTCTTCTCGTT CCTCTTCA | GCAGGTAACAGGATC GCCTAT |
| 18S | AGTCCCTGCCCTTT GTACACA | CGATCCGAGGGCCTCA CTA |
| V3/V4 library primers | TCGTCGGCAGCGTCA GATGTGT ATAAGAGACAGCCTA CGGGNGGCWGCAG | GTCTCGTGGGCTCGGAG ATGTGTATAAGAGACA GGACTACHVGGGTAT CTAATCC |

**Chemicals, Enzymes and other reagents**

| Reagent/Resource | Reference or Source | Identifier or Catalog Number |
|---|---|---|
| Dextran sulfate sodium | Sigma-Aldrich | cat# 42867 |
| Brefeldin A | Sigma-Aldrich | cat# B7651 |
| Matrigel | Sigma-Aldrich | cat# 129925 |
| 7-aminoactinomycin D | Sigma-Aldrich | cat# A9400 |
| SMEM | Sigma-Aldrich | cat# M8403 |
| TrypLE Express | ThermoFisher Scientific | cat# 12604013 |
| DNase I | Roche | cat# 03724778103 |
| IntestiCult™ Organoid Growth Medium (Mouse) | StemCell | cat# 06005 |
| LIVE/DEAD fixable Aqua dead cell stain kit | ThermoFisher Scientific | cat# L34957 |
| FACS Annexin V-FITC Apoptosis Detection Kit | R&D Systems, Inc. | cat# 4830-250-K |
| High-Capacity cDNA Reverse Transcription Kit | ThermoFisher Scientific | cat# 4368813 |
| iQ SYBR Green Supermix | Biorad | cat# 1708880 |
| TRIzol™ Plus RNA Purification Kit | ThermoFisher Scientific | cat# 12183555 |

**Software**

| Reagent/Resource | Reference or Source | Identifier or Catalog Number |
|---|---|---|
| Prism v10 | GraphPad Software | https://www.graphpad.com/scientific-software/prism/ |
| Microsoft Office Excel | Microsoft | |
| g:Profiler | | https://biit.cs.ut.ee/gprofiler/gost. |
| ImageJ 2.14.0 | | https://imagej.net/software/imagej/ |
| FACSDiva | BD Biosciences | |
| STAR (Version 2.6) | | For RNA-seq analysis |

| Reagent/Resource | Reference or Source | Identifier or Catalog Number |
|---|---|---|
| featureCount (subread, Version 1.4.6) | | For RNA-seq analysis |
| Bioconductor package DESeq2 | | For RNA-seq analysis |
| CellRanger 6.0.0 | | For scRNA-seq analysis |
| R version 4.0.2 | | For scRNA-seq analysis |
| Seurat_4.0.3 | | For scRNA-seq analysis |
| celda_1.4.7 | | For scRNA-seq analysis |
| uwot_0.1.8 | | For scRNA-seq analysis |
| harmony_l.0 | | For scRNA-seq analysis |
| MAST_1.16.0 | | For scRNA-seq analysis |
| pheatmap_1.0.12 | | For scRNA-seq analysis |
| reticulate_1.20 | | For scRNA-seq analysis |
| Other | | |

## Animal experiments

Paneth-cell specific SIRT1 KO mice (SIRT1 PKO, $Sirt1^{f/f};Defa6-Cre^+$) and their age- and gender-matched littermate Flox controls (Flox, $Sirt1^{f/f};Defa6-Cre^-$) on the C57BL/6 J background were generated by breeding mice containing a SIRT1 Flox allele (floxed exon 4) (Cheng et al, 2003) with a Paneth-cell-specific Defa6-Cre line (Adolph et al, 2013). To analyze Lgr5$^+$ ISCs by FACS analysis, SIRT1 Flox and PKO mice on the *Lgr5*-EGFP background were generated by breeding SIRT1 Flox and PKO mice with *Lgr5*-EGFP-IRES-creERT2 mice (Jackson Laboratory, 008875). To visualize and quantify Paneth cells, SIRT1 PHet ($Sirt1^{f/+};R26-TdTomato^+;Defa6-Cre^+$) and PKO ($Sirt1^{f/f};R26-TdTomato^+;Defa6-Cre^+$) mice on the R26-TdTomato background, in which Paneth cells were labeled by Defa6-Cre induced TdTomato expression, were generated by breeding SIRT1 PKO mice with Ai9(RCL-tdT) mice (Jackson Laboratory, 007909). Age and gender-matched mice were used for all experiments.

All mice were housed in individualized ventilated cages (Techniplast, Exton, PA) with a combination of autoclaved nesting material (Nestlet, Ancare Corp., Bellmore, NY and Crink-l'Nest, The Andersons, Inc., Maumee, OH) and housed on hardwood bedding (Sani-chips, PJ Murphy, Montville, NJ). Mice were maintained on a 12:12-h light:dark cycle at 22±0.5 °C and relative humidity of 40% to 60%. Mice were provided ad libitum autoclaved rodent diet (NIH31, Harlan Laboratories, Madison, WI) and deionized water treated by reverse osmosis. Mice were negative for mouse hepatitis virus, Sendai virus, pneumonia virus of mice, mouse parvovirus 1 and 2, epizootic diarrhea of infant mice, mouse norovirus, *Mycoplasma pulmonis, Helicobacter* spp., and endo- and ectoparasites upon receipt and no pathogens were detected in sentinel mice during this study.

Acute colitis in mice was induced by adding 2.5% DSS in the drinking water *ad libitum* for five to seven consecutive days. Chronic colitis was induced by challenged mice with two cycles of 7-day 2.5% DSS drinking water separated by a 14-day regular water break. Body weights and rectal bleeding were monitored daily during the DSS treatment and one week immediately after DSS treatment. According to the animal protocol, mice with more than 20% loss of their initial body mass were sacrificed during the experiments.

To deplete gut microbiota, mice were treated with an antibiotic cocktail in their drinking water containing 1 g/L Ampicillin, 500 mg/L Vancomycin, 1 g/L Neomycin, and 1 g/L Metronidazole for 4 weeks. Germ-free mice were generated and housed at the National Gnotobiotic Rodent Resource Center at UNC.

For fecal transplantation experiments, one cohort of 2–4-month-old female Flox and PKO mice were used as donors to provide fresh feces for fecal transplantation. 8–10-week-old germ-free (GF) Flox and PKO mice (both sexes) were then inoculated daily by oral gavage for 7 days with 100 μl/mouse of fecal microbes. Recipient mice were maintained with sterile RO/DI water during fecal transplantation and afterward. Ten days after the last dose of fecal transplantation, they were treated with 2.5% DSS in sterile drinking water for 7 days as described above.

All animal procedures were reviewed and approved by National Institute of Environmental Health Sciences Animal Care and Use Committee. All animals were housed, cared for, and used in compliance with the *Guide for the Care and Use of Laboratory Animals* and housed and used in an Association for the Assessment and Accreditation of Laboratory Animal Care, International (AAALAC) Program.

## Histopathological evaluation of DSS-induced colitis tissue samples

To evaluate the histopathology of Flox and SIRT1 PKO mice treated with DSS-containing drinking water, H&E-stained sections of the large intestine (colon) were examined, and non-regenerative lesions were graded and recorded by a board-certified comparative pathologist based on criteria listed in Appendix Table S3.

## Flow cytometry and isolation of ISCs and Paneth cells

The isolation of small intestinal single cells was performed essentially as described in (Pentinmikko et al, 2019). Small intestines were removed and the fat/mesentery was dissected away. The intestinal lumen was washed with ice-cold PBS (Mg2-Ca2-) using a 18 G feeding needle (Roboz FN-7905) until the intestines appeared white/pink. They were then opened longitudinally. The mucus was removed by gently rubbing the intestine between fingers in cold PBS. The intestines were cut into 3–5-mm fragments and placed into 50-ml conical tubes that were filled with ice-cold 30 ml of PBS (Mg2-Ca2-/)EDTA (10 mM). The samples were incubated and shook intermittently on ice for 30 min continuously discarding and replacing (at least three times) the supernatant. The fragments were then continually re-suspended with ice-cold 30 ml PBS (Mg2-Ca2-)EDTA (10 mM) and intermittently shaken on ice for 10 min, discarding the supernatant again for three times. The fragments were re-suspended again with ice-cold 30 ml PBS (Mg2-Ca2-)/EDTA (10 mM) and incubated and intermittently shaken while waiting on ice for 30 min. The samples were then triturated with a 10-ml pipette 1 to 2 times, and the contents were filtered twice through a 70-μm mesh (BD Falcon) into a 50-ml conical tube to remove villous material and tissue fragments. The tissue

suspensions were centrifuged for 5 min at 300 g (4 °C or room temperature), and the pellets were gently resuspended in 1.0 ml of undiluted TrypLE Express (Invitrogen), 120 µl of DNase I (10 U/µl, Roche) and transferred to 15-ml conical tubes. The samples were incubated in a 32 °C water bath for 3 min (placed 15 ml tubes inside water filled 50 ml tube on 32 °C heat block), were not titurated, and were then placed on ice. Twelve milliliters of cold SMEM (Sigma) were added to each sample and were gently triturated twice. The samples were then centrifuged for 5 min at $300 \times g$.

The pellets were re-suspended and incubated for 15 min on ice in 0.5–1 ml SMEM that contained an antibody cocktail consisting of 10 µl of each antibody, CD45-PE (Invitrogen, 30-F11), CD31-PE (Biolegend, Mec13.3), Ter119-PE (Biolegend, Ter119), CD24-Pacific Blue (Biolegend, M1/69) and EPCAM-APC (Invitrogen, G8.8). Do not add 7AAD to unstained samples. 15 ml tubes: No GFP unstained, PE antibodies only. CD24 Pac. Blue only, EPCAM only, 7AAD only, GFP unstained, 7AAD with all stains GFP. Twelve milliliters of SMEM were added and the samples were centrifuged for 5 min at $250 \times g$. The pellets were resuspended with 0.5–2 ml (depending on the size of the pellet) of SMEM/7-AAD (7-aminoactinomycin D) solution (1:500 dilution). 150 µl of 7-AAD in 1 ~ 2 ml of MEM medium. Stained samples were filtered through a 40-µm mesh/filter (FLOW tubes) before cell sorting using a BD FACSAriaII cell sorter (Becton Dickinson Biosciences, San Jose, CA) equipped with FACSDiVa software.

## FACS analysis of blood and intestinal immune cells

Blood samples were collected with EDTA coated tubes and red blood cells were lysed with ACK lysis buffer at room temperature for 10 min. The collected lymphocytes were then incubated with anti-mouse CD16/32 at room temperature for 10 min to block the IgG Fc receptors. Expression of surface markers was detected by simultaneously staining with indicated antibodies and LIVE/DEAD fixable Aqua dead cell stain kit (ThermoFisher, Cat. # L34957) at room temperature for 15 min followed by flow cytometry.

Intestinal single cells were prepared as described above and immune cells were stained with indicated antibodies and LIVE/ DEAD fixable Aqua dead cell stain kit (ThermoFisher, Cat. # L34957) at room temperature for 15 min followed by flow cytometry.

## AI-assisted image analysis of nuclear β-catenin staining intensity

Visiopharm machine learning and image analysis algorithms were used to quantify the nuclear β-catenin staining intensity in the epithelium of the small intestine of young and aged Flox and PKO mice. Specifically, whole slide images from four groups of mice were imported into the Visiopharm Image Analysis Software platform. Two Visiopharm algorithms were trained on a subset of the data and were used to process each image. The first algorithm was trained to segment out all epithelial tissue from any other tissue or background. The algorithm used a DeepLab convolutional neural network (CNN) at low resolution (10x). Detected epithelial tissue with an area of less than 100 µm² was removed to remove incorrect detections. An example is shown in Appendix Fig. S3D. The second algorithm used a DeepLab CNN at 40x magnification to detect stained nuclei in the previously detected epithelium.

Detected nuclei with areas of less than 5 µm² were discarded to remove incorrect detections. For each detected nuclei, Visiopharm was used to assign an intensity score (integer values 1 to 13) based on the average intensity (for a custom DAB color band) of the nuclei, where 1 represents the lightest stained nuclei and 13 is the most intensely stained. An example of segmented and classified nuclei is shown in Appendix Fig. S1E. For each image, a count of the number of stained nuclei for each score was determined. Data generated in Visiopharm was exported to Excel and subjected to statistical analysis.

## scRNA-seq analysis of total live small intestinal cells

### Isolation of live cells from the small intestine
The single cells from the small intestine of young and aged mice were prepared as described above. The resulting single cell suspension from three pairs of young (2–3 months old) and 2 pairs of aged (12–14 months old) female mice was combined according to their age and genotype, then stained with PI. Flow cytometry sorted live single cells were used for generation of scRNA-seq libraries.

### scRNA-seq library preparation and sequencing
The cells were counted and examined for viability with trypan blue staining using a TC-20 cell counter (Bio-Rad). Approximately 10,000 live cells with above 70% viability were loaded into the Single Cell Chip to generate single cell emulsion in Chromium Controller (10x Genomics, 120263) with Chromium Single Cell 3' Library & Gel Bead Kit v3.1 (10x Genomics, 1000268). Reverse transcription of mRNA and cDNA amplification were carried out following the manufacture's instruction (10x Genomics, 1000268). The amplified cDNA was further fragmented to construct NGS libraries. The libraries were then sequenced by the NIEHS Epigenomics and DNA Sequencing Core Laboratory with the parameters recommended by the manufacture.

### scRNA-seq data preprocessing
The scRNA-seq reads underwent processing using Cell Ranger Single-Cell Software Suite (version 6.0.1, 10X Genomics Inc., CA, USA) for tasks such as alignment, filtering, UMI quantification, and generation of the gene-barcode matrix. Genes were annotated using Ensembl build 93.

### Single-cell gene expression quantification and filtering
The raw gene expression matrices produced per sample through CellRanger were imported into R (version 4.3.0) and transformed into a Seurat object utilizing the R package "Seurat" (version 4.3) (Satija et al, 2015). The ambient RNA was cleaned using SoupX (R package, Version 1.5.2) (Young and Behjati, 2020). The filtration process eliminated deceased cells, doublets, and low-quality cells by considering total UMIs, total genes, and the percentage of mitochondrial UMIs. The upper and lower thresholds for filtering were computed as the mean ± 2 standard deviations (SD).

### Data integration and major cell type determination
Among the initial 66,023 cells, a total of 36,135 cells from 4 samples were integrated using Seurat and R package "harmony" was used to correct batch effects (Korsunsky et al, 2019). In Seurat, normalization and cell cycle scoring was conducted. Highly variable genes

    

(HVG) were chosen using *SCTransform*, with mitochondrial genes being excluded. Subsequently, scaled PCA was performed on the filtered HVG, and UMAP was used for dimensionality reduction. A shared nearest neighbor graph was constructed, followed by clustering at a resolution of 0.6 in Seurat. Major cell types were identified using marker gene prediction with SingleR (https://github.com/LTLA/SingleR) and CellMarker database (http://bioc.hrbmu.edu.cn/CellMarker/) (Zhang et al, 2019).

### Identification of marker genes and differential expression genes (DEG)

Marker genes for cell types were identified by comparing expression levels in the cluster of interest against other clusters using Seurat function *FindMarkers* (MAST test) (Finak et al, 2015). The criteria employed were: (1) at least a 1.2-fold higher average expression compared to other clusters, (2) over 10% detectable cells in the cluster, and (3) highest mean expression compared to other clusters. Differentially expressed genes (DEGs) between two cell groups were computed using Seurat function *FindMarkers* (MAST) with parameters set as min.pct = 0.01 and logfc.threshold = 0.01. Subsequently, GO and pathway analyses were performed on marker genes and DEGs using the R package ClusterProfile (version 3.18.1) (Wu et al, 2021).

### Sub-cell type analysis

Subsequent to the initial data cleaning, raw gene expression counts specific to sub-cell types were isolated from the processed data, establishing a new Seurat object. The gene expression matrices were normalized to total cellular read counts, and highly variable genes (HVG) were identified from the normalized data using the Seurat NormalizeData function with default settings. Following this, observed batch effects were identified and adjusted using the R package "harmony" (Version 1.0.0). The subsequent process of dimensional reduction and clustering followed the same procedure used for the entire cell population.

### Trajectory and RNA velocity analysis

Trajectory and pseudotime analysis were performed using monocle3 (version 1.0.0) (Trapnell et al, 2014). The trajectories were visualized through UMAP plots. For RNA velocity analysis, gene-relative velocity was estimated by python package.py (version 0.17)(La Manno et al, 2018), and downstream analyses, including visualization were performed by R package velocyto.R (Version 0.6). To keep the data analysis consistent, cell coordinates of UMAP generated by Seurat were passed to monocle and velocyto. R. scVelo (version 0.2.5) was utilized to explore cellular dynamics as well (Bergen et al, 2020). This approach helped identify potential driver genes, detect regulatory changes, infer a latent time for reconstructing the temporal sequence of transcriptomic events, and estimate reaction rates of transcription, splicing, and degradation. To ensure consistency, UMAP coordinates generated by scVelo were employed.

### Gene regulatory network analysis

Gene regulatory network analysis was conducted using pySCENIC (Aibar et al, 2017) to reconstruct transcriptional states and regulatory networks from the scRNA-seq data. The activity of transcription factors was inferred by calculating the AUC (Area Under the Curve) for each transcription factor-cell combination using AUCell. Differences in transcription factor activity between the control and treatment groups were analyzed utilizing limma package (version 3.50.0) (Ritchie et al, 2015). Transcription factors exhibiting an adjusted *p* value < 0.05 were considered statistically significant.

## Bacterial 16S rRNA gene amplicon sequencing

### Samples DNA isolation

Fecal samples were transferred to a 2 ml tube containing 200 mg of ≤106 μm glass beads (Sigma, St. Louis, MO) and 0.5 ml of Qiagen PM1 buffer (Valencia, CA). Bead beating was performed for 5 min in a Qiagen TissueLyser II at 30 Hz. After centrifugation for 5 min, 0.45 ml of supernatant was transferred to a new tube containing 0.15 ml of Qiagen IRS solution followed by incubation at 4 °C overnight. After a brief centrifugation, supernatants were transferred to deep well plates containing 0.45 ml of Qiagen binding buffer supplemented with Qiagen ClearMag Beads. DNA was purified using the automated KingFisher™ Flex Purification System and eluted in DNase free water.

Small intestine and colon including luminal contents, were collected under sterile conditions. Tissues with contents were immediately snap-frozen in liquid nitrogen and stored at −80 °C until DNA extraction. Frozen tissues were finely powdered in liquid nitrogen, and approximately 50 mg of the homogenized material was used for extraction of total tissue and microbial DNA using the QIAamp Fast DNA Stool Mini Kit (Qiagen, Hilden, Germany) according to the manufacturer's instructions.

### 16S rRNA gene amplicon sequencing

For fecal samples, 12.5 ng of total DNA was amplified using universal primers targeting the V3/V4 region of the bacterial 16S rRNA gene. Primer sequences contained overhang adapters appended to the 5' end of each primer for compatibility with Illumina sequencing platform. Master mixes contained 12.5 ng of total DNA, 0.5 μM of each primer and 2x KAPA HiFi HotStart ReadyMix (KAPA Biosystems, Wilmington, MA). Each 16S rRNA gene amplicon was purified using the AMPure XP reagent (Beckman Coulter, Indianapolis, IN). In the next step each sample was amplified using a limited cycle PCR program, adding Illumina sequencing adapters and dual-index barcodes (index 1(i7) and index 2(i5)) (Illumina, San Diego, CA) to the amplicon target. The final libraries were again purified using the AMPure XP reagent (Beckman Coulter), quantified and normalized prior to pooling. The DNA library pool was then denatured with NaOH, diluted with hybridization buffer and heat denatured before loading on the MiSeq reagent cartridge (Illumina) and on the MiSeq instrument (Illumina). Automated cluster generation and paired-end sequencing with dual reads were performed according to the manufacturer's instructions.

For small intestinal and colonic samples, the V3-V4 region of the bacterial 16S rRNA gene was amplified using universal primers containing Illumina overhang adapter sequences. Each 25 μL PCR reaction contained 12.5 μL of 2× KAPA HiFi HotStart ReadyMix, 0.5 μL of each 10 μM primer, and 50 ng of template DNA, with nuclease-free water added to volume. PCR cycling conditions were as follows: initial denaturation at 95 °C for 3 min; 25 cycles of 95 °C for 30 s, 55 °C for 30 s, and 72 °C for 45 s; and a final extension at 72 °C for 5 min. Amplicon size (~550 bp) was confirmed using an Agilent TapeStation system (Agilent Technologies, Santa Clara,

CA, USA) following the manufacturer's instructions. PCR products were purified using AMPure XP magnetic beads (Beckman Coulter, Brea, CA, USA). Sequencing libraries were prepared following the Illumina 16S Metagenomic Sequencing Library Preparation protocol, including a second limited-cycle PCR to incorporate dual indices and Illumina sequencing adapters (IDT for Illumina DNA/RNA UD Indexes; Illumina, San Diego, CA, USA). Indexed libraries were purified with AMPure XP beads, quantified using a Qubit fluorometer and an Agilent TapeStation. Equimolar quantities of each library were pooled and sequenced on an Illumina NextSeq platform according to the manufacturer's instructions.

### Bioinformatics analysis
Sequencing output from the Illumina MiSeq platform were converted to fastq format and demultiplexed using Illumina Bcl2Fastq 2.18.0.12. The resulting paired-end reads were processed using QIIME 2 2018.11 (Bolyen et al, 2019). Index and linker primer sequences were trimmed using the QIIME 2 invocation of cutadapt. The resulting paired-end reads were processed with DADA2 through QIIME 2 including merging paired ends, quality filtering, error correction, and chimera detection (Callahan et al, 2016). Amplicon sequencing units from DADA2 were assigned taxonomic identifiers with respect to Green Genes release 13_08 using the QIIME 2 q2-featureclassifier (Bokulich et al, 2018). Significance of differential abundance was estimated using ANCOM as implemented in QIIME 2 (Lin and Peddada, 2020).

## RNA-seq analysis of DSS-treated ileum and colon

To systematically evaluate the response of Flox and PKO mice to DSS-induced colitis, a pilot RNA-seq analysis was performed in the colon (3 pairs) and ileum (2 pairs) of representative 9-month-old females after treatment with two cycles of 7-day 2.5% DSS.

### Sample collection and RNA extraction
RNA was extracted from the ileum and colon with QIAzol. Purification of total RNA was performed with the QIAGEN RNeasy RNA isolation kit (Redwood City, CA) according to the manufacturer's protocol.

### Library preparation and sequencing
RNA-seq libraries were generated with 1 μg of RNA as input using the TruSeq RNA Sample Prep Kit (Illumina, San Diego, CA) with poly(A)-enrichment according to the TruSeq protocol. Index multiplexed samples were sequenced using the 75-bp single-end protocol via the NextSeq500 (Illumina) according to the manufacturer's protocol.

### Reads processing and genome alignment
Raw RNA-Seq reads were subjected to FASTQC (version 0.11.9) [https://www.bioinformatics.babraham.ac.uk/projects/fastqc/] for sequence quality verification. Sequences were then processed through TrimGalore (version 0.6.6) [https://www.bioinformatics.babraham.ac.uk/projects/trim_galore/] for adapter sequence trimming and low-quality reads filtering (--cores 7 --fastqc). The resulting cleaned high-quality reads for each sample/replicate were then employed for the alignment step. STAR (version 020201) (Dobin et al, 2013) (--runMode genomeGenerate) was performed to generate the index file for the mouse reference genome mm10 using the corresponding GENCODE gene models prediction

(version M10). STAR (version 020201) (Dobin et al, 2013) (--runMode AlignReads) was conducted for aligning RNA-Seq reads to the reference genome with the following parameters (--outFilterType BySJout --outSAMtype BAM SortedByCoordinate). The resulting sorted and indexed BAM files were then utilized for the downstream analysis.

### Expression quantification and DEGs analysis
Featurescount (version 2.16.1) (Liao et al, 2014) was used to quantify the gene expression level of the RNA-Seq samples considering the exons from the GENCODE mouse gene models vM10 with unique mapping. The resulted count matrix was loaded into Partek Genomic Suit (version 7.0) to generate a 3D PCA plot for any possible outlier sample detection. The raw read count matrix (after the removal of any outlier sample) was then loaded into R and analyzed for each pairwise comparison using DESeq2 package available on R Bioconductor.

### Functional enrichment analysis
Enrichment analysis was carried out using gprofiler2 (Kolberg et al, 2020), an R package for gene list functional enrichment analysis and namespace conversion toolset g: Profiler.

## Immunohistochemical staining

Immunohistochemical staining was performed using the HRP-polymer technique. Formalin-fixed, paraffin-embedded tissue sections were deparaffinized in xylene and hydrated through graded ethanol to wash buffer. Heat-induced epitope retrieval was performed using a 1x citrate buffer (Biocare Medical, Concord, CA) in the Decloaker® pressure chamber for 15 min at 110 °C; after which endogenous peroxidase blocking was done by immersing the sections in 3% $H_2O_2$ for 15 min. (1) For lysozyme staining, the sections were incubated with 10% normal donkey serum (Jackson Immunoresearch Laboratories, Inc., West Grove, PA) for 20 min, followed by the avidin-biotin blocking kit (Vector Laboratories, Burlingame, CA). The sections were incubated with plyclonal anti-lysozyme antibody (Catalog#A0099, Lot# 20055029, Dako Corp., Carpinteria, CA) and an equivalent dilution of rabbit immunoglobulin fraction (negative control; Dako Corp., Carpinteria, CA) for 30 min at 1:3000 dilution. Sections were incubated with a biotin-sp-conjugated donkey anti-rabbit secondary antibody (Jackson Immunoresearch Laboratories, Inc., West Grove, PA) for 30 min at 1:500 dilution. Label incubation was performed using Vectastain Elite ABC reagent, RTU (Vector Laboratories, Burlingame, CA) for 30 min also. (2) For Ki67 staining, sections were incubated with Rodent Block M (BioCare Medical) for 20 min to block any non-specific binding sites in the sections. Rabbit polyclonal anti-Ki67 (Abcam, Cat# ab16667, Lot# GR3185488-3) antibody was applied to the sections for staining at a dilution of 1:150 for 30 min. Afterwards, tissues were incubated with a Rabbit on Rodent HRP Polymer (BioCare Medical #RMR622) for 30 min. (3) For β-catenin staining, non-specific sites were blocked by incubating slides for 20 min with 2.5% NHS (Vector Labs). The sections were then incubated with rabbit anti-β-catenin antibody (Abcam Cat#ab-16051, Lot#GR3267054-1, 1 mg/mL) at a 1:2500 dilution for 30 min at room temperature. Negative control tissue section received normal rabbit IgG (Millipore Cat#NI01, lot#3238429, 0.1 mg/mL) diluted to match the protein concentration of anti-β-catenin

    

antibody. Vector ImmPress anti-Rabbit IgG Polymer (Vector Labs Cat#MP-7401) was applied for 15 min followed by 3,3-diamino-benzidine (Dako, Carpinteria, CA) for 6 min. All slides were counterstained with hematoxylin, dehydrated and cover slipped. All antibodies are listed in the Reagents & Tools table.

### ER stress response in cultured DLD1 cells

WT and SIRT1 KO DLD1 cell lines were established and cultured as previously described (Ren et al, 2017). To assess the ER stress response, cells were cultured in 10% FBS RPMI medium overnight in a 6-well plate, followed by three washes with PBS. Subsequently, the cells were switched to different media: serum-free DMEM, serum-free and glutamine-free DMEM, glucose-free RPMI with 10% FBS, and RPMI with 10% FBS containing 5 µg/ml of Brefeldin A, all for a duration of 24 h.

To evaluate cell death in response to glucose-depletion induced ER stress, both WT and SIRT1 KO DLD1 cells were cultured overnight in a 6-well plate. Subsequently, they were washed three times with PBS before being switched to glucose-free RPMI with 10% FBS for a period of 3 days. The cells were stained with Annexin V-FITC and propidium iodide (PI) using the TACS Annexin V-FITC Apoptosis Detection Kit (Catalog # 4830-250-K, R&D Systems, Inc.). Stained cells were analyzed on a BD Fortessa cell analyzer (Becton Dickinson Biosciences, San Jose, CA) equipped with FACSDiVa software.

### Small intestinal organoid culture

To visualize and quantify Paneth cells, small intestinal organoids were isolated from 6- to 8-week-old SIRT1 PHet ($Sirt1^{f/+};R26\text{-}TdTomato^+; Defa6\text{-}Cre^+$) and PKO mice ($Sirt1^{f/f};R26\text{-}TdTomato^+;Defa6\text{-}Cre^+$) female mice according to IntestiCult™ protocol (StemCell). Organoids were grown in 3D in 25 µl Matrigel (Sigma-Aldrich cat# 129925) domes per well of 24-well plates in IntestiCult™ Organoid Growth Medium (Mouse) (StemCell).

To test the impact of Paneth cell SIRT1 deletion on ER stress, organoids were isolated from the small intestine of 6–8-week-old Flox and SIRT1 PKO female mice. They were then cultured in regular or glucose-free IntestiCult™ Organoid Growth Medium (Mouse) overnight.

### Quantitative real-time PCR

Total RNA was purified from tissues using RNeasy Mini kit (Qiagen) followed by cDNA synthesis with the High-Capacity cDNA Reverse Transcription Kit (ThermoFisher Scientific). Real-time PCR was performed using iQ SYBR Green Supermix (Biorad). Specific primers are listed in the Reagents & Tools table.

### Statistics

Values are expressed as mean ± standard error of the mean (SEM) from at least three biological replicates, unless otherwise indicated in the Figure legend. Significant differences between the means with two comparison groups were analyzed by two-tailed, unpaired Student's t-test (de Winter, 2013). Statistical significance was set at $p < 0.05$. For all in vivo experiments, outlier samples that fall below Q1 − 3.0 IQR or above Q3 + 3.0 IQR were removed. Significant differences between two

means with more than one comparison were analyzed with multiple unpaired t-tests with correction for multiple comparisons by controlling the false discovery rate. Significant differences between the means with more than two comparison groups were analyzed by either Kruskal-Wallis test or two-way ANOVA with correction for multiple comparisons by controlling the false discovery rate. The exact $p$ values are included in the Source data file. Data analyses were performed using Prism Software 10.0 (GraphPad) or Microsoft Office Excel (Version 16.79.1).

Statistical methods for scRNA-seq analysis, RNA-seq analysis, and bacterial 16S rRNA gene amplicon sequencing are detailed in the corresponding Methods sections.

For animal studies, the sample size in each independent experiment was determined based on our prior experience and mouse availability. The combined sample size for DSS-colitis study was estimated to achieve 2-fold difference of colitis phenotypes (tissue damage scores or rectal bleeding scores) with 80% of power. Experimental animals were randomly assigned into experimental groups. Experimental participants were not blinded when performing experiments and analyzing the data. For gene expression analysis, the sample size was estimated to achieve 2-fold difference of gene expression level with 80% of power. In vitro cell culture experiments were independently performed at least three times and similar results were observed. Each independent experiment was performed with at least three biological replicates, no explicit calculations were done to determine the sample size.

### Study approval

All animal experiments were approved by the NIEHS/NIH Animal Care and Use Committee under the Animal Study Proposal # 2014-0016, STL.

## Data availability

The scRNA-seq dataset has been deposited to Gene Expression Omnibus under the accession number GSE261216: https://www.ncbi.nlm.nih.gov/geo/query/acc.cgi?acc=GSE261216. The RNA-seq dataset has been deposited to Gene Expression Omnibus under the accession number GSE262546: https://www.ncbi.nlm.nih.gov/geo/query/acc.cgi?acc=GSE262546.

The source data of this paper are collected in the following database record: biostudies:S-SCDT-10_1038-S44319-026-00726-3.

## Peer review information

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

## Acknowledgements

We thank NIEHS Comparative Medicine Branch for support with animal experiments; Pathology core laboratory for histopathological evaluation of DSS-treated colon tissues; Epigenomics Core Facility for performance of single-cell RNA-seq, RNA-seq, and 16S rRNA gene amplicon sequencing experiments; Flow Cytometry Center assistance with FACS analysis; Fluorescence Microscopy and Imaging Center for confocal imaging analysis; and UNC National Gnotobiotic Rodent Resource Center for generation and maintenance of germ-free Flox and SIRT1 PKO mice. Synopsis image was created with BioRender.com. This research was supported by the Intramural Research Program of National Institute of Environmental Health Sciences of the NIH to XL (Z01 ES102205). LMG-P was supported by a postdoctoral fellowship from the Postdoctoral Research Associate Training (PRAT) Program of the National Institute of General Medical Sciences (NIGMS, 1FI2GM143339-01). RSB was supported by an NIH grant (DK088199). The contributions of the NIH authors were made as part of their official duties as NIH federal employees, are in compliance with agency policy requirements, and are considered Works of the

United States Government. However, the findings and conclusions presented in this paper are those of the authors and do not necessarily reflect the views of the NIH or the U.S. Department of Health and Human Services.

## Author contributions

**Liz M Garcia-Peterson**: Data curation; Formal analysis; Validation; Investigation; Methodology; Writing—review and editing. **Alicia S Wellman**: Data curation; Formal analysis; Investigation; Visualization; Methodology; Writing—review and editing. **Xiaojiang Xu**: Software; Formal analysis; Visualization; Writing—review and editing. **Ming Ji**: Data curation; Validation; Investigation; Methodology. **Caroline Duval**: Data curation; Methodology. **Igor Shats**: Data curation; Methodology. **Xiaoyue Wu**: Data curation; Methodology. **Thomas A Randall**: Formal analysis; Visualization; Writing—original draft. **Hamed Bostan**: Formal analysis; Visualization. **David Cunefare**: Data curation; Visualization. **Charan K Ganta**: Data curation; Visualization. **Maria Sifre**: Data curation; Methodology. **Xin Xu**: Data curation; Methodology. **Richard S Blumberg**: Resources; Writing—review and editing. **Jian-Liang Li**: Software; Formal analysis; Supervision; Visualization. **Xiaoling Li**: Conceptualization; Resources; Formal analysis; Supervision; Funding acquisition; Investigation; Visualization; Writing—original draft; Project administration; Writing—review and editing.

Source data underlying figure panels in this paper may have individual authorship assigned. Where available, figure panel/source data authorship is listed in the following database record: biostudies:S-SCDT-10_1038-S44319-026-00726-3.

## Funding

## Disclosure and competing interests statement

David Cunefare is a government contractor from Experimental Pathology Laboratories, Inc and Charan K. Ganta is a government contractor from Inotiv, Inc. The authors declare that they have no conflict of interest.

# Expanded View Figures

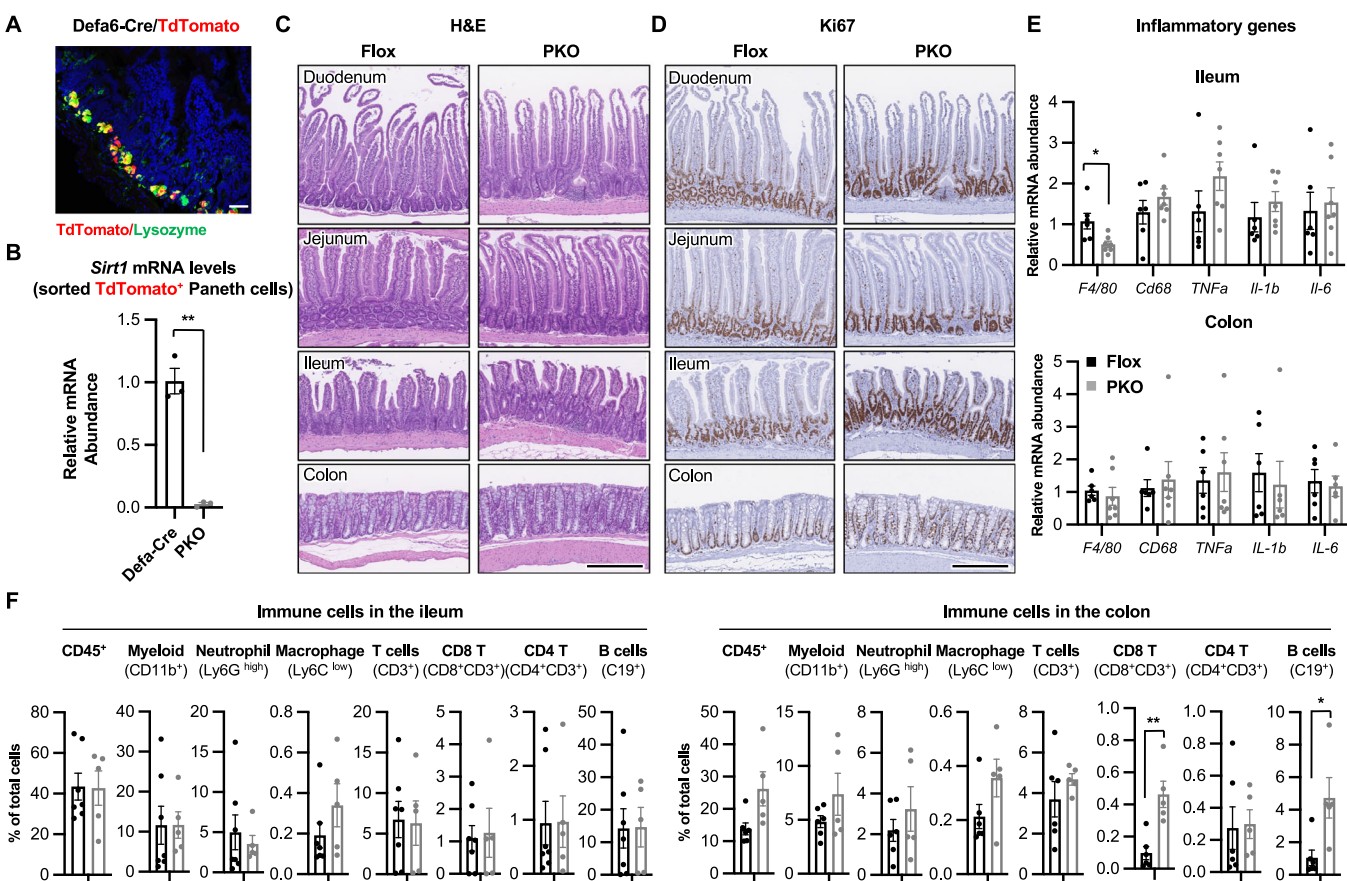

**Figure EV1.  Young SIRT1 PKO mice have normal gut biology under standard feeding conditions.**

(A) Defa6Cre drives specific expression of TdTomato in Paneth cells. The expression of TdTomato and lysozyme in the ileum of Defa6-Cre/TdTomato mice were analyzed by IF. Bar, 50 μm. (B) *Sirt1* is efficiently deleted in Paneth cells in SIRT1 PKO mice. Paneth cells in Defa6-Cre control and PKO mice on a TdTomato+ background were sorted out and the levels of *Sirt1* mRNA in sorted Paneth cells were analyzed by qPCR ($n = 3$ pairs of mice, paired t-test). (C) H&E staining of intestinal sections of young Flox and SIRT1 PKO mice. Bars, 250 μm. (D) Young Flox and SIRT1 PKO mice have comparable cell proliferation under normal feeding condition. Intestinal sections from 4-month-old Flox and SIRT1 PKO mice were stained with an anti-Ki67 antibody. (E) Young Flox and SIRT1 PKO mice have comparable expression of inflammatory genes in the gut under normal feeding condition. The expression of indicated genes was analyzed by qPCR ($n = 6$ Flox and 7 PKO, Student's t-test). (F) Immune cells from the ileum and colon of young Flox and SIRT1 PKO mice. The fraction of indicated immune cells in total live cells isolated from Flox and PKO mice was analyzed by FACS as described in Methods (Ileum: $n = 7$ Flox and 5 PKO mice; Colon: $n = 6$ Flox and 5 PKO mice, Student's t-test). Data information: in (B, E, and F), values are expressed as mean ± SEM; *$p < 0.05$, **$p < 0.01$; no marks, not significant. Source data are available online for this figure.

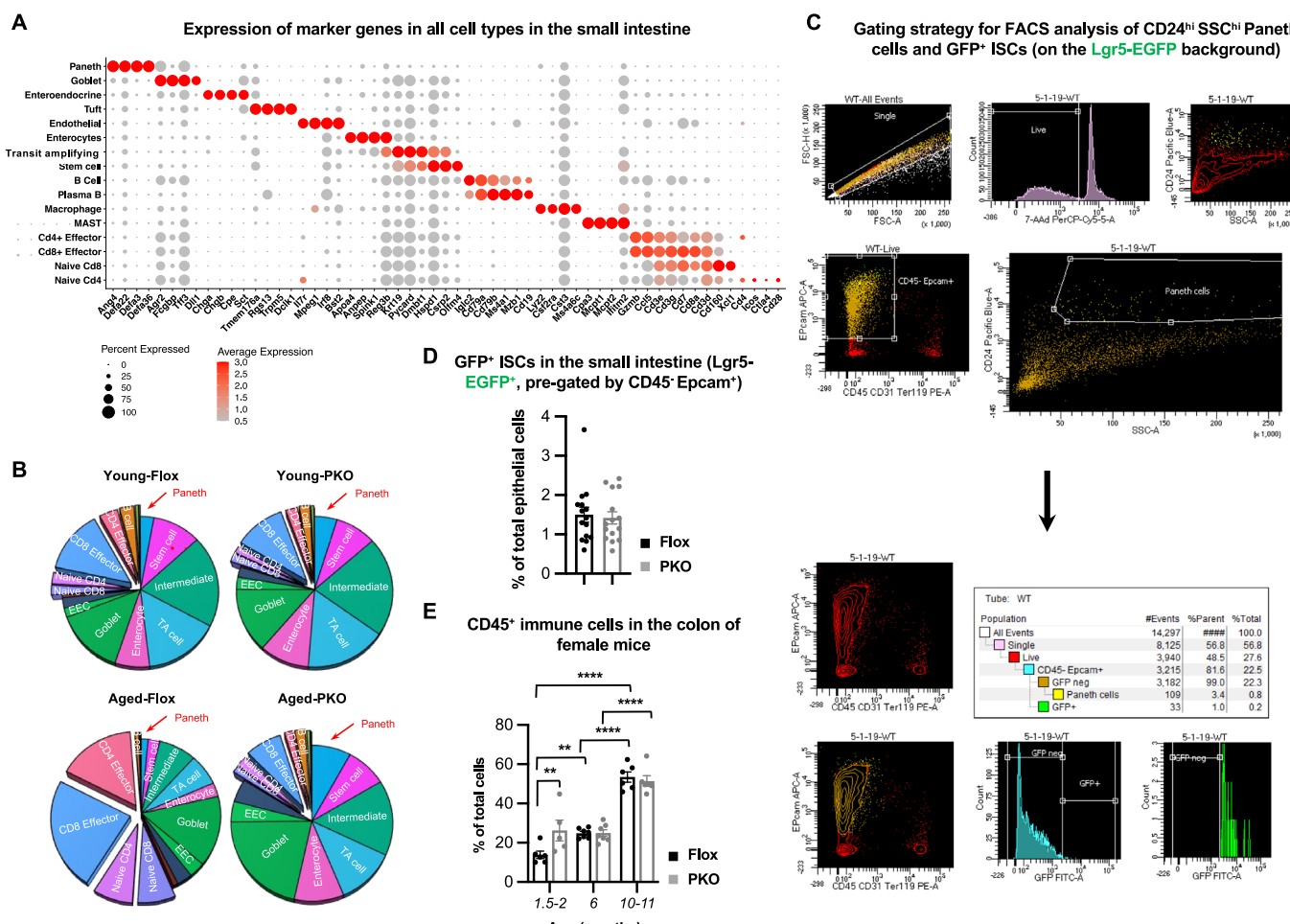

**Figure EV2. scRNA-seq analysis of small intestinal cells in young and aged Flox and SIRT1 PKO mice.**

(A) Dot plot showing the expression of marker genes in different function groups of total live cells isolated from the small intestine. (B) Relative composition of different cell types in the small intestine of young and aged mice. (C) Gating strategy for FACS analysis of CD24$^{hi}$ SSC$^{hi}$ Paneth cells and GFP$^{+}$ intestinal stem cells (ISCs). Small intestinal epithelial cells from 2- to 3-month-old Flox and SIRT1 PKO mice on the Lgr5-EGFP background were isolated and analyzed as described in Methods. The FACS plots from the same Flox mouse in Fig. 1C are used here to demonstrate the gating strategy. (D) SIRT1 PKO mice have normal abundance of ISCs in the small intestine. The abundance of ISCs in Flox and PKO mice on the Lgr5-GFP background was analyzed by FACS ($n = 15$ pairs, paired t-test). (E) Age induces immune cell expansion in the colon. The fraction of CD45$^{+}$ immune cells in total live cells isolated from Flox and PKO mice at indicated ages was analyzed by FACS as described in Methods ($n = 6, 5, 7, 6, 6, 6$ mice per indicated group, two-way ANOVA). Data information: (D and E), values are expressed as mean ± SEM; *$p < 0.05$, **$p < 0.01$, ****$p < 0.0001$; no marks, not significant. Source data are available online for this figure.

 

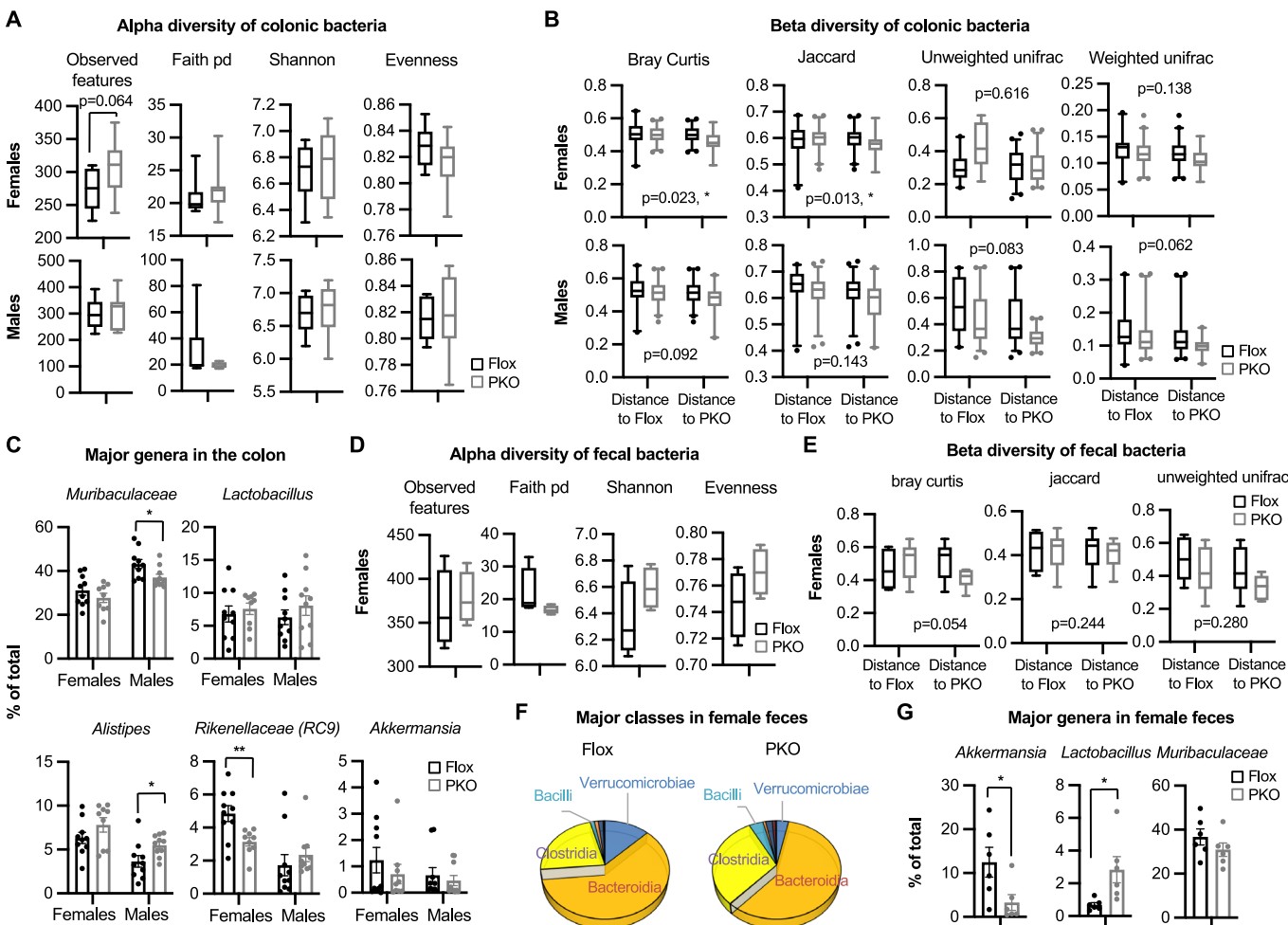

**Figure EV3. The impact of Paneth cell SIRT1 deficiency on colonic and fecal bacteria.**

(A–C) SIRT1 PKO mice have altered colonic microbiota. Total colonic DNA from Flox and SIRT1 PKO mice were analyzed for mucosal adherent microbiota using 16S rRNA gene amplicon sequencing as described in Methods (n = 10 Flox and 9 PKO females and 10 Flox and 10 PKO males). (A) Alpha diversity of colonic bacteria (Student's t-test). (B) Beta diversity of colonic bacteria (Pair-wise permanova test). (C) The abundance of major genera in the colon (n = 10 Flox and 9 PKO females and 10 Flox and 10 PKO males, Student's t-test). (D–G) SIRT1 PKO female mice have altered fecal microbiota. Total fecal DNA from Flox and SIRT1 PKO females were analyzed using 16S rRNA gene amplicon sequencing. (D) Alpha diversity of fecal bacteria (n = 6 mice/genotype, Student's t-test). (E) Beta diversity of fecal bacteria (n = 6 mice/genotype, Pair-wise permanova test). (F) Altered abundance of several major classes of fecal microbiota in SIRT1 PKO females. (G) SIRT1 PKO females have altered levels of several major genera in feces (n = 6/genotype, Student's t-test). Data information: in (A, B, D, and E), box-and-whisker plot with the box representing the interquartile range (Q1 to Q3), a line inside indicating the median, and whiskers representing the 2.5–97.5 percentile; in (C and G), values are expressed as mean ± SEM; *p < 0.05, **p < 0.01; no marks, not significant. Source data are available online for this figure.

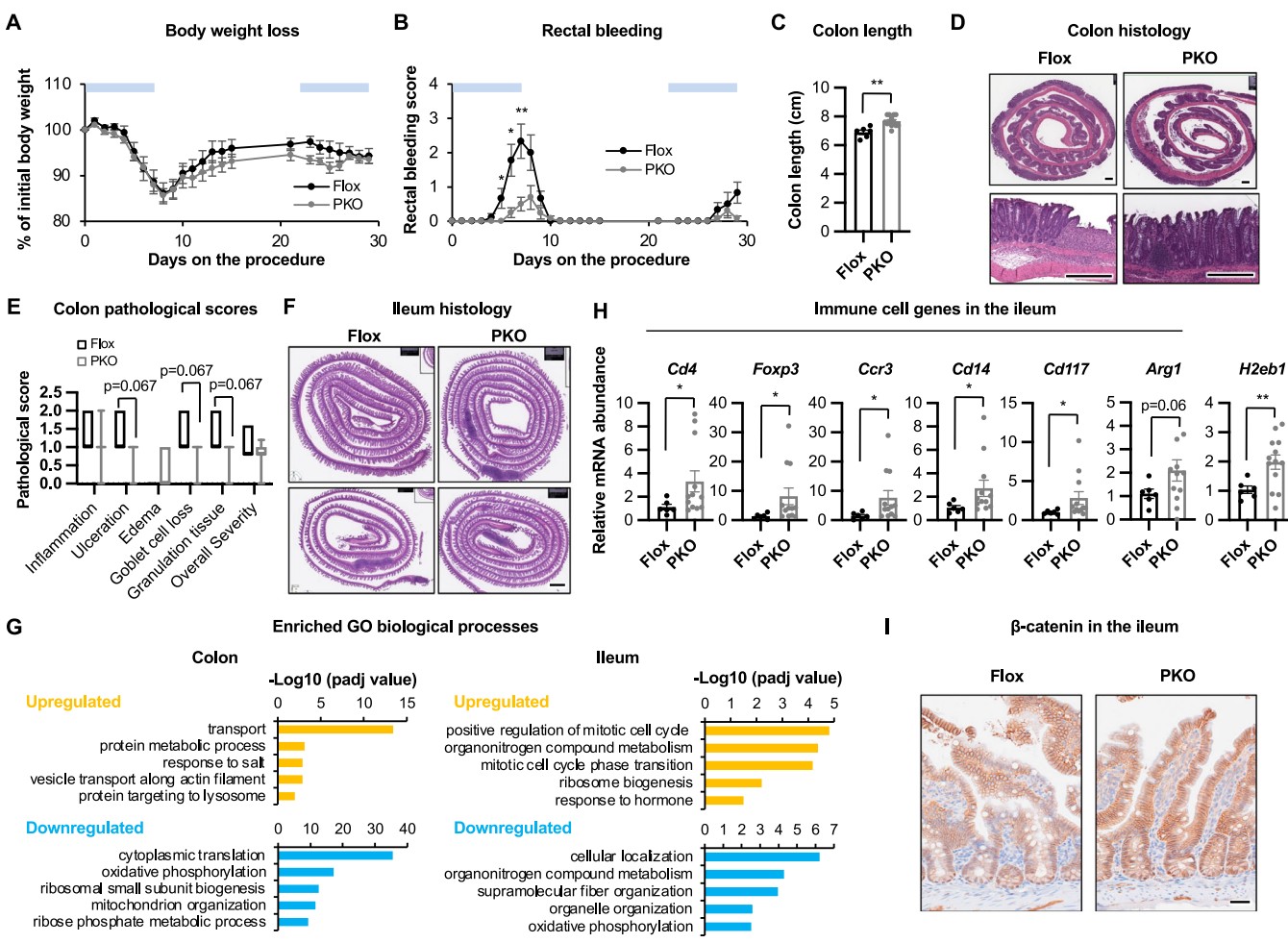

**Figure EV4. SIRT1 PKO mice are protected from DSS-colitis.**

(A–C) Nine-month-old SIRT1 PKO mice are more resistant to DSS-induced colitis than Flox mice. Nine-month-old Flox and SIRT1 PKO mice were treated with 2 cycles of 2.5% DSS for 7 days with 14 day of regular water break in between. Their body weight (A), rectal bleeding (B), and colon length (C) were analyzed ($n = 6$ Flox and 12 PKO mice, Student's t-test). (D, E) Colon histology of 9-month-old Flox and SIRT1 PKO mice after 2 cycles of DSS treatment. (D) Representative H&E staining of Swiss swirl from 2 Flox and 2 PKO mice are shown. Bars, 500 µm. (E) The histopathological severity was evaluated by a professional pathologist as described in Methods ($n = 6$ Flox and 13 PKO, Mann–Whitney test). (F) Ileum histology of 9-month-old Flox and SIRT1 PKO mice after 2 cycles of DSS treatment. Representative H&E staining of Swiss swirl from 2 Flox and 2 PKO mice are shown. Bar, 1 mm. (G) The top enriched functional pathways in the colon or ileum of 9-month-old Flox and SIRT1 PKO mice after 2 cycles of DSS treatment. Total RNA from the indicated mice were analyzed by RNA-seq. DEGs between SIRT1 KO vs Flox were analyzed by GO biological process and top enriched pathways are shown (Permutation testing with the Benjamini–Hochberg adjusted p-values). Enrichment score represents $-\log_{10}$-transformed p-values after adjusted for FDR false discovery rate. (H) The small intestine of DSS-treated SIRT1 PKO mice have increased expression of immune cell genes. The expression of indicated genes was analyzed by qPCR ($n = 6$ Flox and 12 PKO mice, Student's t-test). (I) The expression of β-catenin was analyzed by IHC in the ileum of DSS-treated Flox and SIRT1 PKO mice. Bar, 50 µm. Data information: in (A, B, C, and H), values are expressed as mean ± SEM; in (E), box-and-whisker plot with the box representing the interquartile range (Q1 to Q3), a line inside indicating the median, and whiskers representing the 2.5–97.5 percentile; *$p < 0.05$, **$p < 0.01$; no marks, not significant. Source data are available online for this figure.

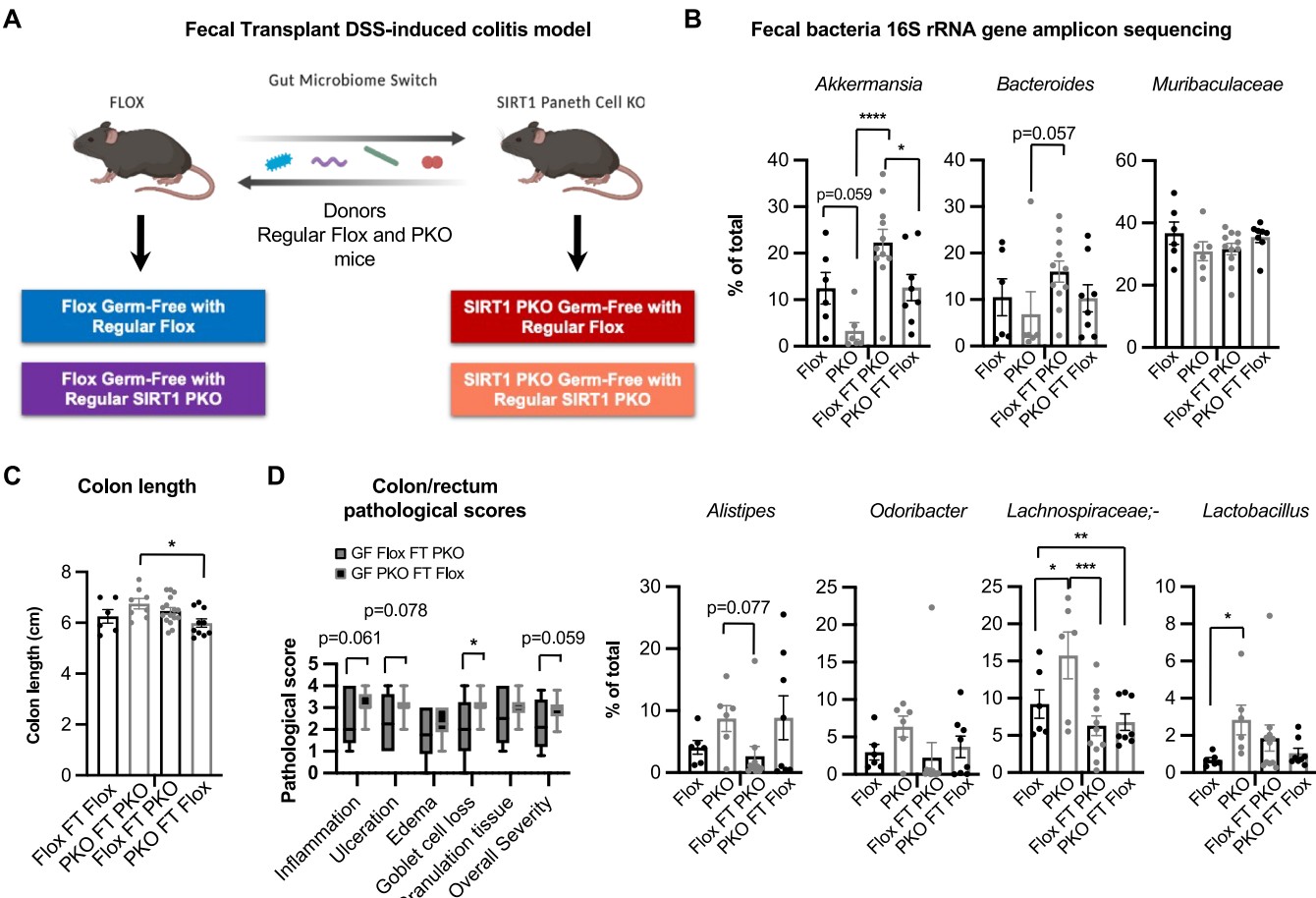

**Figure EV5. Paneth cell SIRT1 regulates the sensitivity to DSS-colitis through the gut microbiota.**

(A) Schematic representation of fecal transplantation DSS-colitis model. Germ-free Flox mice transplanted with fecal microbes from regular Flox mice (Flox FT Flox) or PKO mice (Flox FT PKO), and germ-free PKO mice transplanted with fecal microbes from regular Flox mice (PKO FT Flox) or PKO mice (PKO FT PKO) were treated with 2.5% DSS. (B) The abundance of major genera of fecal bacteria before and after fecal transplantation. Fecal transplantation was performed as described in Methods and fecal microbiota were analyzed by 16S rRNA amplicon sequencing ($n = 6$, 6, 11, and 8 mice, two-way ANOVA). (C) Colon length of DSS-treated mice ($n = 6$ GF Flox FT Flox, 7 GF PKO FT PKO, 16 GF Flox FT PKO, and 11 GF PKO FT Flox, two-way ANOVA). (D) The histopathological severity of DSS-treated mice evaluated by a professional pathologist as described in Methods ($n = 14$ GF Flox FT PKO, and 10 GF PKO FT Flox, Student's t-test). Data information: in (B and C), values are expressed as mean ± SEM; in (D), box-and-whisker plot with the box representing the interquartile range (Q1 to Q3), a line inside indicating the median, and whiskers representing the 2.5–97.5 percentile; *$p < 0.05$, **$p < 0.01$, ***$p < 0.001$, ****$p < 0.0001$; no marks, not significant. Source data are available online for this figure.

