## [Peer Review File · EMBO Reports]

Paneth cell SIRT1 deficiency increases intestinal stress resistance by modulating the gut microbiota

Liz Garcia-Peterson, Alicia Wellman, Xiaojiang Xu, Ming Ji, Caroline Duval, Igor Shats, Xiaoyue Wu, Thomas Randall, Hamed Bostan, David Cunefare, Charan Ganta, Maria Sifre, Xin Xu, Richard Blumberg, Jianliang Li, and Xiaoling Li

Corresponding author(s): Xiaoling Li (lix3@niehs.nih.gov)

Review Timeline:

Submission Date:	18th Apr 25
Editorial Decision:	5th Jun 25
Revision Received:	6th Dec 25
Editorial Decision:	21st Jan 26
Revision Received:	27th Jan 26
Accepted:	9th Feb 26

Editor: Achim Breiling

Transaction Report:

Dear Dr. Li,

Thank you for the submission of your manuscript to EMBO reports. I have now received reports from the three referees that were asked to evaluate your study, which can be found at the end of this email. As you will see, the referees think that these findings are of interest. However, they have several comments, concerns, and suggestions, indicating that a major revision of the manuscript is necessary to allow publication of the study in EMBO reports. As the reports are below, and all the referee concerns need to be addressed, I will not detail them here.

Given the constructive referee comments, I would like to invite you to revise your manuscript with the understanding that the concerns of the referees must be addressed in the revised manuscript and/or in a detailed point-by-point response. Acceptance of your manuscript will depend on a positive outcome of a second round of review. It is EMBO reports policy to allow a single round of revision only and acceptance of the manuscript will therefore depend on the completeness of your responses included in the next, final version of the manuscript.

- 1) a .docx formatted version of the final manuscript text (including legends for main figures, EV figures and tables), but without the figures included. Figure legends should be compiled at the end of the manuscript text.
- 2) individual production quality figure files as .eps, .tif, .jpg (one file per figure), of main figures and EV figures. Please upload these as separate, individual files upon re-submission.

- 4) a complete author checklist, which you can download from our author guidelines

(<https://www.embopress.org/page/journal/14693178/authorguide>). Please insert page numbers in the checklist to indicate where the requested information can be found in the manuscript. The completed author checklist will also be part of the RPF.

- 5) that primary datasets produced in this study (e.g. RNA-seq, ChIP-seq, structural and array data) are deposited in an appropriate public database. If no primary datasets have been deposited, please also state this in a dedicated section (e.g. 'No

primary datasets have been generated and deposited'), see below.

The accession numbers and database should be listed in a formal "Data Availability" section that follows the model below. This is now mandatory (like the COI statement). Please note that the Data Availability Section is restricted to new primary data that are part of this study. This section is mandatory. As indicated above, if no primary datasets have been deposited, please state this in this section

Data availability

6) We now request the publication of original source data with the aim of making primary data more accessible and transparent to the reader. You will receive a separate email with instructions for providing source data with your revised manuscript, including information how to upload and organize the files.

8) Regarding data quantification and statistics, please make sure that the number "n" for how many independent experiments were performed, their nature (biological versus technical replicates), the bars and error bars (e.g. SEM, SD) and the test used to calculate p-values is indicated in the respective figure legends (also for EV and Appendix figures). Please also check that all the p-values are explained in the legend, and that these fit to those shown in the figure. Please provide statistical testing where applicable. Please avoid the phrase 'independent experiment', but clearly state if these were biological or technical replicates. Please also indicate (e.g. with n.s.) if testing was performed, but the differences are not significant. In case n=2, please show the data as separate datapoints without error bars and statistics. See also: <http://www.embopress.org/page/journal/14693178/authorguide#statisticalanalysis>

9) Please add scale bars of similar style and thickness to all microscopic images, using clearly visible black or white bars (depending on the background). Please place these in the lower right corner of the images themselves. Please do not write on or near the bars in the image but define the size in the respective figure legend.

10) Please also note our reference format:
<http://www.embopress.org/page/journal/14693178/authorguide#referencesformat>

12) We now use CRediT to specify the contributions of each author in the journal submission system. CRediT replaces the author contribution section. Please use the free text box to provide more detailed descriptions and do NOT provide your final manuscript text file with an author contributions section. See also our guide to authors: <https://www.embopress.org/page/journal/14693178/authorguide#authorshippinguidelines>

13) All Materials and Methods need to be described in the main text using our 'Structured Methods' format, which is required for all research articles. According to this format, the Methods section should include a Reagents and Tools Table (listing key

reagents, experimental models, software, and relevant equipment and including their sources and relevant identifiers), uploaded as separate file, and a Methods section in which we encourage the authors to describe their methods using a step-by-step protocol format with bullet points, to facilitate the adoption of the methodologies across labs. More information on how to adhere to this format as well as downloadable templates (.doc) for the Reagents and Tools Table can be found in our author guidelines (section 'Structured Methods'):

14) Please reduce the keywords to five and order the sections like this, using these names:

Title page - Abstract - Keywords - Introduction - Results - Discussion - Methods - Data availability section - Acknowledgements (please include here also the funding information) - Disclosure and Competing Interests Statement - References - Figure legends - Expanded View Figure legends

15) Please make sure that all the funding information is also entered into the online submission system and that it is complete and similar to the one in the acknowledgement section of the manuscript text file.

I look forward to seeing a revised form of your manuscript when it is ready.

Yours sincerely,

Referee #1:

Previous studies by the authors and others showed that the SIRT1 deficiency in intestinal epithelial cells resulted in the increased numbers and activity of Paneth cells. This study by Garcia-Peterson et al used Paneth cell-specific SIRT1 KO (SIRT1 PKO) mice and shows that SIRT1 suppresses the numbers and activity of Paneth cells in a cell-autonomous manner. They further analyzed the mechanisms by which SIRT1 suppress the activity of Paneth cells. SIRT1 PKO mice showed enhanced Wnt signaling and ATF4/ER stress pathway in Paneth cells, leading to increased anti-microbial peptides production. SIRT1 PKO are resistant to DSS-induced colitis, and it is dependent on microbiota.

Based on the results of previous studies, the findings are predictable, but the authors provide some mechanistic insights into how SIRT1 suppresses Paneth cell activity. There are several concerns that should be addressed.

1. The authors should provide evidence showing how SIRT1 activity is enhanced by aging.
2. The authors should analyze how SIRT1 reduces ATF4 protein degradation. Is it dependent on deacetylation?
3. In Fig. 5, the authors should show alpha- and beta-diversities of microbiota/
4. In Fig. 6, data showing the increased immune cell infiltration in DSS-treated SIRT1 KO mice seem misleading. In general, the increase in immune cells correlates with the severity of DSS colitis. Thus, the data seems contradictory. The authors should carefully discuss the discrepancy.
5. In Fig. 7A-D experiments, the authors claim that the protection against DSS colitis requires gut microbiota. But, in wild-type SPF mice, Abx-mediated depletion of microbiota usually improves the severity of DSS colitis. Thus, it is possible that after Abx treatment, wild-type SPF mice became resistant to DSS colitis, and accordingly the severity of colitis became to the similar level to that of SIRT1 KO mice. Thus, the author should perform this experiment by including non-Abx group.

Referee #2:

In a prior paper this group reported on the role of intestinal SIRT1 in epithelial knockouts suggesting that activation of this deacetylase might be developed as treatment for IBD (Wellman et al 2017). By contrast, in the current Paneth cell specific knockout there was increased resistance to chemically induced colitis, elevated Wnt signalling and ATF4/endoplasmic reticulum stress pathways. Most importantly, increased Paneth cell abundance was observed as well as enhanced AMP production. Apparently, the protection was mediated by Paneth cell induced alterations in the gut microbiota.

This is clearly a very interesting paper but looking at the details there are quite a few inconsistencies that should be clarified.

Specific points:

1. In Fig. 1B the increase in Paneth cells looks impressive but statistics are lacking. N? p? On the other hand, in Fig. 1C Paneth cell percentage is up minimally from 6 to 7% with significant overlap but a $p < 0.05$ is given using Student's t-test with paired statistics? And no correction for multiple comparisons (Bonferroni)? Such questionable statistics were used throughout the data, with the exception of Fig. 7. The statistics should be redone appropriately.
2. In the violin plots of Fig. 2E there is little difference between Flox and PKO in 2 Paneth cell clusters with a p as low as < 0.001 ? Each dot represents a single cell, but how many animals were used here? In 2 F with respect to lysozyme, Defa1, Defa5 and angiogenin 4 the differences are not statistically significant, only the Regs are? In this regard measuring mucosal antibacterial activity may be helpful (Wehkamp et al. PNAS 2005), otherwise it remains unclear how expression translates into function.
3. In Fig. 3C statistics are missing, in Fig. 3 E and 3F there is no obvious difference between Flox and PKO at the respective age groups with respect to β -catenin staining? At the least this does not support "enhanced Wnt signalling".
4. The "activation" of ATF4 in Fig. 4A is not obvious to this reviewer. The DLD1 cells used to make the point may or may not reflect the in vivo situation.
5. In Figure 5 there is no evidence for "interaction" between Paneth cells and microbiota but only evidence for a change in microbial composition. Again, differences in AMPs between Flox and PKO are minimal (Fig. 5C).
6. Fig. 6 E looks convincing but was the pathologist blinded?
7. In Fig. 7 histology should be "quantitated" as in Fig. 6. No difference between Flox and PKO with respect to weight loss and colon length, only rectal bleeding?
8. In Fig. 8 no statistics are given, just stars. p? Statistical method? There is virtually no difference in the means between healthy and non-inflamed?

Referee #3:

The manuscript by Liz Garcia-Peterson et al, referred as EMBOR-2025-61780V1, reports that the loss of SIRT1 in Paneth cells results a greater expression of antimicrobial peptides that are produced by Paneth cells as what observed in mice with an epithelial deficiency of SIRT1. Such phenotype is absent when mice are treated with antibiotics and can be transferred in wild-type mice upon fecal microbiota transplantation.

This accords with a significant compositional change of the gut microbiota However it remains unclear why the deletion of SIRT1 in Paneth cells improves DSS-induced colitis severity in mice while its absence in all intestinal epithelial cells is associated with increased susceptibility to DSS-induced colitis (Gastroenterology 2017).

One of the possible explanation may be related to the compositional change in the gut microbiota since the loss of SIRT1 in Paneth cells failed to heighten the expression in antimicrobial peptides in the germ-free condition.

This said, the microbiome-mediated sequence of events leading to ATF4/ER stress response and to enhancement of Wnt/ β -catenin signaling remains unclear. This question should be answered by evaluating the ATF4/ER stress response and the susceptibility of several mutants (eg. Rag mice) that are co-housed with mice with a Paneth cell deficiency of SIRT1.

Equally of importance, it is expected that the authors will evaluate whether there is gender disparity upon loss of SIRT1 in Paneth cells and whether the composition of the mucosal-adherent microbiota may differ between the colon and ileum of mutant mice for explaining the reported spatio-temporal differences.

Referee #1:

Previous studies by the authors and others showed that the SIRT1 deficiency in intestinal epithelial cells resulted in the increased numbers and activity of Paneth cells. This study by Garcia-Peterson et al used Paneth cell-specific SIRT1 KO (SIRT1 PKO) mice and shows that SIRT1 suppresses the numbers and activity of Paneth cells in a cell-autonomous manner. They further analyzed the mechanisms by which SIRT1 suppress the activity of Paneth cells. SIRT1 PKO mice showed enhanced Wnt signaling and ATF4/ER stress pathway in Paneth cells, leading to increased anti-microbial peptides production. SIRT1 PKO are resistant to DSS-induced colitis, and it is dependent on microbiota.

Based on the results of previous studies, the findings are predictable, but the authors provide some mechanistic insights into how SIRT1 suppresses Paneth cell activity. There are several concerns that should be addressed.

1. The authors should provide evidence showing how SIRT1 activity is enhanced by aging.

Figure R1. The expression of *Sirt1* is differentially impacted by aging in different small intestinal cell types. The mRNA levels of *Sirt1* in different cell types from young and aged mouse small intestines were compared using the scRNA-seq dataset in Figure 1A and 1B.

To answer this question, we analyzed the mRNA levels of *Sirt1* in different small intestinal cell types from young and aged mice using the scRNA-seq dataset created in Figure 1A and 1B. This analysis revealed an intriguing cell type specific impact of aging on the expression of *Sirt1* (Figure R1). Specifically, in enterocytes, the most abundant small intestinal epithelial cells responsible for nutrient absorption and transport (Noah *et al*, 2011), the mRNA levels of *Sirt1* were significantly decreased by aging, which is consistent with literature reports as well as our previous observation using whole intestinal tissues (Wellman *et al*, 2017). However, unexpectedly, in many other cell types in the small intestine, including TA cells, goblet cells, and CD8 effector T cells, the expression of *Sirt1* was significantly increased in aged mice. The mRNA levels of *Sirt1* also displayed a trend of increase in aged Paneth cells. Therefore, depending on the cell type, aging could enhance SIRT1 activity by promoting its expression.

This observation reinforces the idea that SIRT1 is regulated and functions in a cell-type specific manner.

2. The authors should analyze how SIRT1 reduces ATF4 protein degradation. Is it dependent on deacetylation?

Our data in the original Figure 4F (new Figure 4E) and Figure 4G showed that SIRT1 *deficiency* reduces ATF4 protein degradation. To answer the reviewer's question, we analyzed the ubiquitination and acetylation status of ATF4 in WT and SIRT1 KO DLD1 cells after treatment with TSA (to inhibit other classes of HADCs) and MG-132 (to inhibit degradation of ubiquitinated proteins). Our results (in new Figure 4F) showed that ATF4 in SIRT1 KO cells have reduced ubiquitination, just like β -catenin. However, this reduction was not coupled with any alteration of its acetylation levels (Figure 4G, AcK), suggesting that SIRT1 induces ATF4 protein degradation independent of its deacetylation.

4F Acetylation and ubiquitination of ATF4 in DLD1 cells (treated with TSA and MG-132)

3. In Fig. 5, the authors should show alpha- and beta-diversities of microbiota/

Thanks for the suggestion. We analyzed the alpha- and beta-diversities of fecal microbiota previously shown in Figure 5, they are included in Figure EV3D and EV3E in the revised manuscript.

Per the request from the Reviewer #3, we also sequenced and compared the mucosal microbiota from the small intestine and colon of Flox and SIRT1 PKO mice. The results of alpha-diversity, beta-diversity, and taxa analyses of the small intestinal microbiota are now included in Figure 5 together with other small intestinal data. The results of colonic and fecal microbiota are included in Figure EV3. Consistent with the notion that the gut microbiota alternations in SIRT1 PKO mice originated from the increased anti-microbial peptide production in the small intestine, particularly in female mice, we observed the most significant changes of microbiota in the small intestine of female mice (Figure 5) compared to the colonic microbiota and fecal microbiota (Figure EV3).

4. In Fig. 6, data showing the increased immune cell infiltration in DSS-treated SIRT1 KO mice seem misleading. In general, the increase in immune cells correlates with the severity of DSS colitis. Thus, the data seems contradictory. The authors should carefully discuss the discrepancy.

The reviewer is right, we were also surprised to observe the increased expression of immune cell genes in SIRT1 PKO mice after two rounds of DSS treatment despite their reduced severity of DSS-colitis. Because SIRT1 PKO mice also displayed increased expression of genes involved in gut epithelial cell proliferation and recovery, we previously discussed the possible role of these immune cells in inducing the Wnt signaling and promoting tissue repair in the original submission (on Page 15). After careful analyzing different immune cell markers elevated in SIRT1 PKO mice, we believe the seemingly contradictory phenotypes observed in SIRT1 PKO mice suggest a complex but functionally protective immune remodeling in response to DSS-induced colitis. The PKO mice were not broadly immunosuppressive. Instead, they likely reprogrammed the immune response toward regulation, resolution, and repair rather than unchecked inflammation in this chronic colitis model. Particularly, the elevation of *Cd19*, *Foxp3*, and *Il10* suggests activation of regulatory B cells and Treg cells, which are known to suppress inflammation, protect against tissue injury, and promote tissue repair (Li & He, 2004; Wang *et al*, 2015; Yanaba *et al*, 2011).

We incorporated above discussion on Page 16-17 in the revised manuscript.

5. In Fig. 7A-D experiments, the authors claim that the protection against DSS colitis requires gut microbiota. But, in wild-type SPF mice, Abx-mediated depletion of microbiota usually improves the severity of DSS colitis. Thus, it is possible that after Abx treatment, wild-type SPF mice became resistant to DSS colitis, and accordingly the severity of colitis became to the similar level to that of SIRT1 KO mice. Thus, the author should perform this experiment by including non-Abx group.

This is a great suggestion. We in fact previously performed DSS-colitis in another group of aged matched non-Abx mice at approximately the same time as the current Abx-treated group. The only difference was that the DSS treatment had to be discontinued on Day 5 instead of Day 8 due to the rapid body weight loss in the Regular Flox mice. Histological data from this group were already included in Figure 7D (Regular). We have now included the data from these Regular mice into Figure 7A-7C.

Consistent with the reviewer's point, a side-by-side comparison of Regular and Abx-treated mice revealed that Abx-treatment increased resistance to DSS-induced colitis in Flox mice, but not further in PKO mice. This observation suggests that the highly expressed anti-microbial peptides in PKO mice may eliminate certain colitogenic gut microbiota species, thereby mimicking Abx-

induced protection against DSS-colitis. We have updated the main text to include these new observations and conclusions on Page 17-18.

Referee #2:

In a prior paper this group reported on the role of intestinal SIRT1 in epithelial knockouts suggesting that activation of this deacetylase might be developed as treatment for IBD (Wellman et al 2017). By contrast, in the current Paneth cell specific knockout there was increased resistance to chemically induced colitis, elevated Wnt signalling and ATF4/endoplasmic reticulum stress pathways. Most importantly, increased Paneth cell abundance was observed as well as enhanced AMP production. Apparently, the protection was mediated by Paneth cell induced alterations in the gut microbiota.

This is clearly a very interesting paper but looking at the details there are quite a few inconsistencies that should be clarified.

Specific points:

1. In Fig. 1B the increase in Paneth cells looks impressive but statistics are lacking. N? p? On the other hand, in Fig. 1C Paneth cell percentage is up minimally from 6 to 7% with significant overlap but a $p < 0.05$ is given using Student's t-test with paired statistics? And no correction for multiple comparisons (Bonferroni)? Such questionable statistics were used throughout the data, with the exception of Fig. 7. The statistics should be redone appropriately.

First of all, we apologize for the confusion caused by our statistics. As stated in the "Statistics" section, "Significant differences between the means with two comparison groups were analyzed by two-tailed, unpaired Student's t-test based on de Winter et al. (de Winter, 2013). Significant differences between two means with more than one comparison were analyzed with multiple unpaired t-tests with correction for multiple comparisons by controlling the false discovery rate, and significant differences between the means with more than two comparison groups were analyzed by either Kruskal-Wallis test or two-way ANOVA with correction for multiple comparisons by controlling the false discovery rate". So correction for multiple comparisons was only performed for data with multiple comparisons.

The data in Figure 1B (and 3C) are derived from scRNA-seq analysis. As detailed in the section of "scRNA-seq analysis of total live small intestinal cells" in the Supplemental Methods, this experiment was performed with one replicate per group, where each replicate consisted of pooled single cells from 2-3 mice. This pooled design is common in scRNA-seq analysis due to the high cost of the technique. Consequently, statistical analysis of cell composition between groups was not feasible, and further validation via FACS analysis is needed (as we did in Figure 1C). However, statistical analysis comparisons for differential gene expression analysis in different cell types between genotypes or ages remain valid, given the large number of single cells analyzed per cell type (e.g. Figure 2E and 3B). Please note that statistical methods for scRNA-seq analysis (and RNA-seq analysis) are detailed in "scRNA-seq analysis of total live small intestinal cells" and "RNA-seq analysis of DSS-treated ileum and colon" sections in the

Supplemental Methods. We added this information at the end of the “Statistics” section in the revision.

We used paired Student’s t-test for data in Figure 1C because the FACS experiment was performed using one pair of Flox and PKO mice at a time, over a period spanning more than half a year (for a total of 14 pairs of mice). Due to inevitable variations in experimental conditions across different time points of this period (e.g. equipment settings, animal age, and reagent batches, and so on), we treated each pair as an internally controlled unit. This approach was recommended after consultation with a professional statistician. We added more details of this experiment in the Methods section.

We would also like to point out that scRNA-seq and FACS analyses use fundamentally different approaches to classify cell types. In scRNA-seq analysis, single cells are clustered based on their transcriptomic profiles and subsequently annotated using the expression patterns of known marker genes, which include both cell surface and intracellular markers. In contrast, FACS analysis of live cells relies primarily on cell surface markers and cell-type-specific morphological characters. For instance, the CD24^{high} SSC^{high} FACS method identifies Paneth cells by their high surface CD24 signal and high side scatter, which reflects the abundance of antimicrobial peptide-containing vesicles in their cytoplasm. However, CD24 is not exclusive to Paneth cells, as some intestinal stem cells and enteroendocrine cells also express CD24 (von Furstenberg *et al*, 2011). Moreover, many CD45⁺ lymphocytes also exhibit high granularity. As a result, while the CD24^{high} SSC^{high} FACS method is widely used for purification of live Paneth cells, it only yields a partially purified Paneth cell population due to contamination of other cell types. Therefore, it is not surprising that scRNA-seq and FACS results may differ. In our case, we were pleased to observe a consistent trend of increased Paneth cells in SIRT1 PKO mice using both methods.

2. In the violin plots of Fig. 2E there is little difference between Flox and PKO in 2 Paneth cell clusters with a p as low as <0.001? Each dot represents a single cell, but how many animals were used here? In 2 F with respect to lysozyme, Defa1, Defa5 and angiogenin 4 the differences are not statistically significant, only the Regs are? In this regard measuring mucosal antibacterial activity may be helpful (Wehkamp *et al*. PNAS 2005), otherwise it remains unclear how expression translates into function.

Yes, in the violin plots in Figure 2E, each dot represents a single Paneth cell, and the Y-axis is displayed on a log scale. As noted in the response to Point #1, this experiment was performed with one replicate per group, where each replicate consisted of pooled single cells from 2-3 mice. The comparison is between two groups of single cells, and the detailed statistical methods are in “scRNA-seq analysis of total live small intestinal cells” section in the Supplemental Methods.

In Figure 2F, yes, due to substantial variability between individual animals (each dot here represents one mouse), we observed only a trend, rather than a statistically significant increase, in the expression of *Lyz1*, *Defa1*, *Defa5*, and *Ang4* by qPCR analysis of total RNA from the whole ileal tissue (not isolated Paneth cells). We also attempted IHC staining for Lysozyme, but the results remained variable between individual animals. We measured the small intestinal mucosal antibacterial activity *in vitro* using *E. coli* and *lactobacillus* per the suggestion of the

reviewer. Again, due to the substantial variability between individual animals, we failed to observe any significant difference between Flox and PKO mice (data not shown).

On the other hand, per the request of Reviewer #3, we sequenced mucosal adherent microbiota in both the ileum and colon of male and female mice using 16S rRNA gene amplicon sequencing for this revision (new Figure 5 and Figure EV3). Our data showed that in female mice, deletion of Paneth cell SIRT1 resulted in the most significant changes of microbiota in the small intestine (Figure 5) compared to the colonic microbiota and fecal microbiota (Figure EV3). Particularly, the abundance of *Lactobacillus* was significantly increased in the small intestine of SIRT1 PKO females (Figure 5D), supporting the notion that the gut microbiota alternations in SIRT1 PKO mice were originated from the increased anti-microbial peptide production in the small intestine.

3. In Fig. 3C statistics are missing, in Fig. 3E and 3F there is no obvious difference between Flox and PKO at the respective age groups with respect to β -catenin staining? At the least this does not support "enhanced Wnt signalling".

As mentioned in our response to Point #1, the Data in Figure 3C are derived from scRNA-seq analysis. Because this experiment was performed with one replicate containing pooled single cells from 2-3 mice, we were not able to perform statistical analysis of cell composition between groups.

For data in Figure 3E and 3F, the nuclear β -catenin staining intensity in the crypts was quantified by AI-enhanced imaging analysis. Based on our experience, this method is highly specific (distinguishing between nuclear and cytoplasmic signals) and sensitive, capable of detecting subtle differences that would otherwise be invisible to the naked eye. Even through the effect size is small, the difference of β -catenin staining intensity between groups are statistically significant differences, and this difference was further enhanced after DSS treatment (Figure 6H and 6I).

4. The "activation" of ATF4 in Fig. 4A is not obvious to this reviewer. The DLD1 cells used to make the point may or may not reflect the in vivo situation.

Again, this result is from pySCENIC analysis of the transcriptomes of individual Paneth cells, with each dot representing the ATF4 activity in a single Paneth cell. It is evident that a greater proportion of Paneth cells from PKO mice exhibit high ATF4 activity (indicated in yellow). Figures S4A and S4B (new Appendix Figure S2A and S2B) also present these findings as heatmaps, in which the average transcription factor activity in each cell type is presented using a color scale. We hope these heatmaps provide clear evidence that Paneth cells from both young and aged PKO mice display elevated ATF4 activity.

We completely agree that the DLD1 cells are not the best to model the in vivo situation. To address this concern, we cultured small intestinal organoids from Flox and PKO mice, then cultured them in glucose free medium overnight to induce ER stress. Our results showed that in small intestinal organoids from Flox mice, ER stress led to increased expression of *Atf4* and its target gene *Chop*, while suppressing anti-microbial gene expression (Figure 4H, Flox, Glu free

vs regular medium). In small intestinal organoids from PKO mice, ER stress induced a higher induction of ATF4 and its targets, and this increase was coupled with a reduced suppression of anti-microbial genes (Figure 4H, PKO, Glu free vs regular medium), indicating enhanced resistance to ER stress. These findings are consistent with our observations in vivo from PKO mice (Figure 2).

5. In Figure 5 there is no evidence for "interaction" between Paneth cells and microbiota but only evidence for a change in microbial composition. Again, differences in AMPs between Flox and PKO are minimal (Fig. 5C).

Regarding our conclusion that "Paneth cell SIRT1 modulates the interaction between Paneth cells and the gut microbiota", we believe the results that Paneth cell SIRT1 deficiency increases the expression of AMPs in regular mice but not germ-free mice in Figure 5C indicate that Paneth cell SIRT1-mediated regulation of Paneth cells requires the gut microbiota. However, we agree with the reviewer that the differences of AMPs between Flox and PKO are small (please note the log scale of the Y-axis in Figure 5C). We toned down the conclusion from this figure into "Together, our data indicate that Paneth cell SIRT1 modulates the gut microbiota.". We hope this addressed the reviewer's concern.

6. Fig. 6 E looks convincing but was the pathologist blinded?

No, the pathologist was not blinded.

7. In Fig. 7 histology should be "quantitated" as in Fig. 6. No difference between Flox and PKO with respect to weight loss and colon length, only rectal bleeding?

Thanks for the suggestion. The histology in Figure 7I is now evaluated by a board-certified comparative pathologist. The results from two fecal switched groups, GF-Flox FT PKO and GF-PKO FT Flox are included in new Figure EV5D. The other two non-switched control groups were not included due to the small sample sizes.

Yes, no difference in the body weight loss between groups. In our hands, body weight loss did not consistently correlate with the severity of DSS-colitis (please also see new Figure EV4A-EV4E).

Colon length data are presented in Figure S6C (new Figure EV5C), and the only difference is PKO transplanted with fecal microbes from Flox mice (PKO FT Flox) had significantly shorter colons compared to PKO transplanted with fecal microbes from PKO mice (PKO FT PKO).

Both findings support the notion that fecal microbes from regular Flox mice contain colitogenic microbiota species which are absent or inhibited in regular PKO mice.

8. In Fig. 8 no statistics are given, just stars. p? Statistical method? There is virtually no difference in the means between healthy and non-inflamed?

Again, the Data in Figure 8 are derived from a published scRNA-seq dataset, and the experiment was performed with one replicate per group, where each replicate consisted of pooled single cells from multiple patients. The statistical methods used in this analysis are essentially the same as described in “scRNA-seq analysis of total live small intestinal cells” in the Supplemental Methods.

The difference of *SIRT1* mRNA levels between healthy and non-inflamed samples is statistically significant.

Referee #3:

The manuscript by Liz Garcia-Peterson et al, referred as EMBOR-2025-61780V1, reports that the loss of SIRT1 in Paneth cells results a greater expression of antimicrobial peptides that are produced by Paneth cells as what observed in mice with an epithelial deficiency of SIRT1. Such phenotype is absent when mice are treated with antibiotics and can be transferred in wild-type mice upon fecal microbiota transplantation.

This accords with a significant compositional change of the gut microbiota However it remains unclear why the deletion of SIRT1 in Paneth cells improves DSS-induced colitis severity in mice while its absence in all intestinal epithelial cells is associated with increased susceptibility to DSS-induced colitis (Gastroenterology 2017).

One of the possible explanation may be related to the compositional change in the gut microbiota since the loss of SIRT1 in Paneth cells failed to heighten the expression in antimicrobial peptides in the germ-free condition.

The reviewer is right that loss of SIRT1 in Paneth cells vs loss of SIRT1 in whole intestinal epithelium differentially impacts the susceptibility of mice to DSS-induced colitis, and this difference is associated with distinct alterations in gut microbiota composition. As noted in the

Discussion section of our original submission, we speculated that the timing of Paneth cell activation in these two mouse models might underlie this discrepancy. In our previous study (Wellman *et al.*, 2017), we showed that in SIRT1 iKO mice, Paneth cell activation occurred only in aged but not young mice, and this late-onset activation appeared to have minimal impact on gut microbiota or intestinal inflammation. Instead, dysbiosis in aged SIRT1 iKO mice was primarily induced by defective ileal bile acid resorption and subsequent bile acid accumulation, which *sensitized* the mice to colitis and age-associated inflammation. In SIRT1 PKO mice, on the other hand, Paneth cell activation occurred at a young age in the absence of bile acid defect. This early activation resulted in a distinct gut microbiota composition (notably, *increase* of *Lactobacillus*), which *increased* gut epithelial resistance to chemically induced colitis.

This said, the microbiome-mediated sequence of events leading to ATF4/ER stress response and to enhancement of Wnt/beta-catenin signaling remains unclear. This question should be answered by evaluating the ATF4/ER stress response and the susceptibility of several mutants (eg. Rag mice) that are co-housed with mice with a Paneth cell deficiency of SIRT1.

Thanks for this insightful suggestion. In order to finish the revision within a reasonable timeframe, we utilized an alternative experimental model, small intestinal organoids derived from Flox and PKO mice, to address this question. We cultured these organoids in the sterile condition, then analyzed whether small intestinal organoids from PKO mice still exhibit an increased response to ER stress and have elevated levels of anti-microbial peptide genes.

As shown in new Figure 4H, when cultured in a regular medium, small intestinal organoids from PKO mice displayed either reduced (females) or comparable (males) expression of ATF4/targets and anti-microbial peptide genes compared to those from Flox mice (Figure 4H, regular medium). This observation suggests that activation of both pathways observed in PKO mice requires gut microbiota. On the other hand, in response to ER stress induced by overnight culture in a glucose free medium, small intestinal organoids from PKO mice displayed an increased induction of ATF4 and its target genes, and this increase was coupled with a reduced suppression of anti-microbial genes (Figure 4H, PKO, Glu free vs regular medium), indicating an enhanced resistance to ER stress. These findings are consistent with our observations from the small intestine of PKO mice (Figure 2) and suggest that the elevated levels of anti-microbial peptide genes require both microbiome and ER stress.

Equally of importance, it is expected that the authors will evaluate whether there is gender disparity upon loss of SIRT1 in Paneth cells and whether the composition of the mucosal-

adherent microbiota may differ between the colon and ileum of mutant mice for explaining the reported spatio-temporal differences.

We performed DSS-colitis experiments in both males and females, and data in original Figure 6 were combined data from both males and females. As shown in Figure R2A-R2C, when we analyzed males and females separately, we found that SIRT1 PKO mice of both sexes were similarly protected from DSS-induced body weight loss and rectal bleeding compared with Flox controls. However, we did notice that male mice were overall more sensitive to DSS-colitis than females, with treatment terminated in more than half of male mice by day 5 due to severe body weight loss and/or rectal bleeding. For this reason, we replotted Figure 6A and 6B using only the data from the first 5 days of DSS treatment.

Figure R2. Young SIRT1 PKO mice are protected from DSS-colitis compared to Flox controls. Three-month old Flox and SIRT1 PKO mice were treated with 2.5% DSS in drinking water for 5-7 days. Their body weight (A), rectal bleeding (B), and colon length (C) were analyzed (Females, n=25/group; Males, n=18/group; Student's t-test, *p<0.05, **p<0.01). Treatment in more than half of male mice was terminated by day 5 due to severe body weight loss and/or rectal bleeding. Data in Figure 6A-6C were combined data from both females and male.

As suggested by the reviewer, we performed 16S rRNA gene amplicon sequencing of mucosal adherent microbiota in both the ileum and colon of male and female mice. The results of alpha-diversity, beta-diversity, and taxa analyses of the small intestinal microbiota are now included in Figure 5 together with other small intestinal data. The results of colonic and fecal microbiota are included in Figure EV3. Consistent with the notion that the gut microbiota alternations in SIRT1 PKO mice originate from the increased anti-microbial peptide production in the small intestine,

particularly in female mice, we observed the most pronounced changes of microbiota in the small intestine (Figure 5) rather than in the colon or fecal samples (Figure EV3).

References:

- de Winter JCF (2013) Using the Student's t-test with extremely small sample sizes. *Practical Assessment, Research, and Evaluation* 18: Article 10
- Li MC, He SH (2004) IL-10 and its related cytokines for treatment of inflammatory bowel disease. *World J Gastroenterol* 10: 620-625
- Noah TK, Donahue B, Shroyer NF (2011) Intestinal development and differentiation. *Exp Cell Res* 317: 2702-2710
- von Furstenberg RJ, Gulati AS, Baxi A, Doherty JM, Stappenbeck TS, Gracz AD, Magness ST, Henning SJ (2011) Sorting mouse jejunal epithelial cells with CD24 yields a population with characteristics of intestinal stem cells. *Am J Physiol Gastrointest Liver Physiol* 300: G409-417
- Wang L, Ray A, Jiang X, Wang JY, Basu S, Liu X, Qian T, He R, Dittel BN, Chu Y (2015) T regulatory cells and B cells cooperate to form a regulatory loop that maintains gut homeostasis and suppresses dextran sulfate sodium-induced colitis. *Mucosal Immunol* 8: 1297-1312
- Wellman AS, Metukuri MR, Kazgan N, Xu X, Xu Q, Ren NSX, Czopik A, Shanahan MT, Kang A, Chen W *et al* (2017) Intestinal Epithelial Sirtuin 1 Regulates Intestinal Inflammation During Aging in Mice by Altering the Intestinal Microbiota. *Gastroenterology* 153: 772-786
- Yanaba K, Yoshizaki A, Asano Y, Kadono T, Tedder TF, Sato S (2011) IL-10-producing regulatory B10 cells inhibit intestinal injury in a mouse model. *Am J Pathol* 178: 735-743

Dear Dr. Li,

Thank you for the submission of your revised manuscript to our editorial offices. I have now received the reports from the three referees that I asked to re-evaluate the study, you will find below. As you will see, the referees now fully support publication of your study in EMBO reports.

Before we can proceed with formal acceptance, I have the editorial requests below I ask you to address in a final revised manuscript. Please also provide a final p-b-p-response to the editorial requests.

Editorial requests:

- It seems some authors have affiliations to Biotech Companies (Charles River Laboratories and Inotiv). Please mention this in the 'Disclosure and Competing Interests Statement' and indicate potential conflicts there.
- Please order the manuscript sections like this, using only these names:
Title page - Abstract - Keywords - Introduction - Results - Discussion - Methods - Data availability section - Acknowledgements (please include here all the funding information) - Disclosure and Competing Interests Statement - References - Figure legends - Expanded View Figure legends
- Thus, please remove the conflict of interests statement from the title page (but make sure the second statement is renamed to 'Disclosure and Competing Interests Statement') and the running title below the abstract.
- We now use CRediT to specify the contributions of each author in the journal submission system. CRediT replaces the author contribution section. Please use the free text box to provide more detailed descriptions and do NOT provide your final manuscript text file with an author contributions section. See also our guide to authors (section 'Author contributions'):
<https://link.springer.com/journal/44319/submission-guidelines#cms-Revised-submissions>
- Please remove now the referee tokens from the Data Availability section (DAS) and make sure that the datasets are public latest upon online publication of the manuscript. Please also add specific URLs for GSE261216, GSE262546 to the DAS. Moreover, please remove any information not related to the externally deposited datasets from the DAS.
- Please move all methods information and references presently in the Appendix to the main manuscript text file. We do not allow supplementary methods in the Appendix file! Then please remove any callouts to supplemental or supplementary methods.
- Please include all antibody and primer information into the Reagents & Tools table and update all related callouts. Then, please delete Appendix Tables S14 and S15 (and their callouts).
- Please add a title page to the Appendix file ('Appendix for ...' followed by the title a of the paper - do not add author names and affiliations) and a table of contents (TOC) with page numbers.
- Please add Appendix tables S1, S4, S5, S9, S10, S11, S12 and S13 directly to the Appendix file, including a title and a legend (and listed in the TOC). We need one final Appendix file (as pdf) including all Appendix items. Appendix items should not be uploaded separately.
- Tables S2, S3, S5, S7, S8 and S9 are datasets. Please upload these as dataset files using the nomenclature 'Dataset EVx' and updated their callouts in the manuscript. Please add a legend for each these on the first TAB of the respective excel file.
- Please check again that the number "n" for how many independent experiments were performed, their nature (biological versus technical replicates), the bars and error bars (e.g. SEM, SD) and the test used to calculate p-values is indicated in the respective figure legends (main, EV and Appendix figures). Please also check that all the p-values are explained in the legend, and that these fit to those shown in the figure. Please provide statistical testing where applicable. Please avoid the phrase 'independent experiment' but clearly state if these were biological or technical replicates. Please also indicate (e.g. with n.s.) if testing was performed, but the differences are not significant. In case n=2, please show the data as separate datapoints without error bars and statistics. See also:

<https://link.springer.com/journal/44319/submission-guidelines#cms-Figure-and-data-presentation>

If $n < 5$, please show single datapoints for diagrams. Moreover:

- Please note that the exact p values are not provided in the legends of figures 1C-E; 2E, F; 3B, E, F, G, H; 4C, D, G, H; 5A, B, D, E, F; 6A, E, F, G, I; 7A, B, C, E, F, H; 8A, B; EV1 B, E, F; EV2 E, EV3 A, B, C, G; EV4 B, H; EV5 B, C, D; S4 A, B.
- Please indicate the statistical test used for data analysis in the legends of figures 2B, E; 3B, 4B

- Please add scale bars of similar style and thickness to all microscopic images, using clearly visible black or white bars (depending on the background). Please place these in the lower right corner of the images themselves. Please do not write on or near the bars in the image but define the size in the respective figure legend. Presently, many scale bars have text nearby. Moreover, panel 4D, EV1A, S1D and S1E are missing scale bars.

- Thanks for uploading the sourced data. Please upload this as one zip folder per figure so that each folder has separate files/subfolders, one per panel. The file 'Figure 1 numerical data' is identical to Appendix Table S1. Please make sure that the source data are complete. It seems numerical data for 2B and 4B are missing. Please check.

- During our regular image integrity checks we noted a reuse between images in panels 1C (upper panel) and EV2c (big panel). Please check. If this is intentional, please clearly explain and state this in the respective figure legends.

Moreover, we observed that the blot images within the figure set appear pixelated under analysis. This is often a result of converting original 16-bit TIFF files to RGB format for publication. While this is not inherently problematic, it can give the impression of image alteration to critical readers.

To address this, please upload the blot figures and the Appendix blot figure file at a higher resolution. Please upload the original blot source data (at 16-bit TIFF) with your online submission or deposit the raw files on BioStudies

<https://www.ebi.ac.uk/biostudies/sourcedata/studies>

and including the archive accession number in your Data Availability section.

This will enable us to confirm the integrity of the complete figure set and enhance transparency for readers.

In addition, I would need from you uploaded separately:

- a short, two-sentence summary of the manuscript (not more than 35 words).

- two to four short (!) bullet points highlighting the key findings of your study (two lines each).

- a schematic summary figure as separate file that provides a sketch of the major findings (not a data image) in jpeg or tiff format (with the exact width of 550 pixels and a height of not more than 400 pixels) that can be used as a visual synopsis on our website.

I look forward to seeing the further revised version of your manuscript when it is ready. Please let me know if you have questions regarding the revision.

I look forward to seeing a new revised version of your manuscript as soon as possible.

Best,

Referee #1:

The authors well responded to the reviewer's comments, and the manuscript is improved.

Referee #2:

The manuscript suitable for publication in EMBO reports without further revision.

Referee #3:

This revised version of the manuscript has been greatly improved. The authors have done a nice job by addressing most of the comments and adding supporting additional data.

Editorial requests:

- It seems some authors have affiliations to Biotech Companies (Charles River Laboratories and Inotiv). Please mention this in the 'Disclosure and Competing Interests Statement' and indicate potential conflicts there.

Information is provided as instructed. Please note that Dr. David Cunefare's affiliation has changed since the submission of this manuscript last April.

- Please order the manuscript sections like this, using only these names:
Title page - Abstract - Keywords - Introduction - Results - Discussion - Methods - Data availability section - Acknowledgements (please include here all the funding information) - Disclosure and Competing Interests Statement - References - Figure legends - Expanded View Figure legends

Modified as instructed.

Thus, please remove the conflict of interests statement from the title page (but make sure the second statement is renamed to 'Disclosure and Competing Interests Statement') and the running title below the abstract.

Removed as instructed.

- We now use CRediT to specify the contributions of each author in the journal submission system. CRediT replaces the author contribution section. Please use the free text box to provide more detailed descriptions and do NOT provide your final manuscript text file with an author contributions section. See also our guide to authors (section 'Author contributions'):
<https://link.springer.com/journal/44319/submission-guidelines#cms-Revised-submissions>

Deleted the Author contributions section in the manuscript. Will do in the submission system.

- Please remove now the referee tokens from the Data Availability section (DAS) and make sure that the datasets are public latest upon online publication of the manuscript. Please also add specific URLs for GSE261216, GSE262546 to the DAS. Moreover, please remove any information not related to the externally deposited datasets from the DAS.

Modified as instructed.

- Please move all methods information and references presently in the Appendix to the main manuscript text file. We do not allow supplementary methods in the Appendix file! Then please remove any callouts to supplemental or supplementary methods.

All supplementary methods are now in the main text.

- Please include all antibody and primer information into the Reagents & Tools table and update

all related callouts. Then, please delete Appendix Tables S14 and S15 (and their callouts).

Modified as instructed.

- Please add a title page to the Appendix file ('Appendix for ...' followed by the title of the paper - do not add author names and affiliations) and a table of contents (TOC) with page numbers.

The title page is added to the Appendix file as instructed.

- Please add Appendix tables S1, S4, S5, S9, S10, S11, S12 and S13 directly to the Appendix file, including a title and a legend (and listed in the TOC). We need one final Appendix file (as pdf) including all Appendix items. Appendix items should not be uploaded separately.

We added Appendix tables S4, S6 (the first tab), and S13 directly into the Appendix file as instructed (renumbered as S1, S2, and S3).

Table S5 and S10 are too big to add, we uploaded them as datasets instead.

Appendix Table S1 is the source data of Figure 1B, Appendix Table S11 is the source data of Figure EV4G. We removed them from the Appendix tables and included them in the source data files instead.

- Tables S2, S3, S5, S7, S8 and S9 are datasets. Please upload these as dataset files using the nomenclature 'Dataset EVx' and updated their callouts in the manuscript. Please add a legend for each these on the first TAB of the respective excel file.

Appendix tables S2, S5, S7, S8, S9, S10, and S12 are modified and uploaded as instructed.

Table S3 is the source data for Figure 2B. It is moved to the source data file.

- Please check again that the number "n" for how many independent experiments were performed, their nature (biological versus technical replicates), the bars and error bars (e.g. SEM, SD) and the test used to calculate p-values is indicated in the respective figure legends (main, EV and Appendix figures). Please also check that all the p-values are explained in the legend, and that these fit to those shown in the figure. Please provide statistical testing where applicable. Please avoid the phrase 'independent experiment' but clearly state if these were biological or technical replicates. Please also indicate (e.g. with n.s.) if testing was performed, but the differences are not significant. In case n=2, please show the data as separate datapoints without error bars and statistics. See also:

<https://link.springer.com/journal/44319/submission-guidelines#cms-Figure-and-data-presentation>

The figure legends have been carefully checked and edited to contain all required information.

If $n < 5$, please show single datapoints for diagrams. Moreover:

- Please note that the exact p values are not provided in the legends of figures 1C-E; 2E, F; 3B, E, F, G, H; 4C, D, G, H; 5A, B, D, E, F; 6A, E, F, G, I; 7A, B, C, E, F, H; 8A, B; EV1 B, E, F; EV2 E, EV3 A, B, C, G; EV4 B, H; EV5 B, C, D; S4 A, B.

All graphs with $n < 5$ show single datapoints.

For the exact p values, because there are not enough spaces to include all exact p values in the legends or figures, we provided them in the corresponding source data files for most figures (except the violin plots in Figure 2E, 3B, 8, and Appendix Figure 4, see the next paragraph). Please note that the fecal microbiota data in Figure 7F and EV5B were originally analyzed using the whole 16S rRNA gene amplicon dataset. Since each microbe is presented separately in these figures and the source data are provided, we reanalyzed them separately using two-way ANOVA and included the exact p values in the source data files. Additionally, we corrected/modified p values for some graphs because the original values were calculated using tests other than the Student's t-test (e.g. non-parametric Mann Whitney test, etc).

Violin plots in Figure 2E and 3B are from our mouse small intestinal scRNA-seq dataset (GSE261216). Violin plots in Figure 8 and Appendix Figure S4 are from public scRNA-seq datasets of human CD patients (Accession DUOS-000146 CD_Atlas_2021_GIDER; DUOS-000145 CD_Atlas_2021_PRISM). We provided the exact p values directly in the figures.

- Please indicate the statistical test used for data analysis in the legends of figures 2B, E; 3B, 4B

The statistical methods are now provided.

- Please add scale bars of similar style and thickness to all microscopic images, using clearly visible black or white bars (depending on the background). Please place these in the lower right corner of the images themselves. Please do not write on or near the bars in the image but define the size in the respective figure legend. Presently, many scale bars have text nearby. Moreover, panel 4D, EV1A, S1D and S1E are missing scale bars.

Modified/added as instructed.

- Thanks for uploading the sourced data. Please upload this as one zip folder per figure so that each folder has separate files/subfolders, one per panel. The file 'Figure 1 numerical data' is identical to Appendix Table S1. Please make sure that the source data are complete. It seems numerical data for 2B and 4B are missing. Please check.

As mentioned above, Appendix Table S1 is the source data for Figure 1B, we removed it from the Appendix tables in this revision.

The numerical data of Figure 2B was in the original Appendix Table S3 and the numerical data of Figure 4B was in the second tab of original Appendix Table S6. We moved both into the respective source data files.

- During our regular image integrity checks we noted a reuse between images in panels 1C (upper panel) and EV2c (big panel). Please check. If this is intentional, please clearly explain and state this in the respective figure legends.

Yes, they are the same images. In Figure 1C, it is the representative FACS plot for CD24^{hi} SSC^{hi} analysis of Paneth cells in Flox mice (we just noticed that the labels of Flox and PKO are missing in this panel. We added them in the revision). In Figure EV2C, the same mouse was used to show the overall FACS gating strategy. We added the following sentence into the legend of Figure EV2C “*The FACS plots from the same Flox mouse in Figure 1C are used here to demonstrate the gating strategy.*”. Please let us know if you would like us to use plots from a different mouse.

Moreover, we observed that the blot images within the figure set appear pixelated under analysis. This is often a result of converting original 16-bit TIFF files to RGB format for publication. While this is not inherently problematic, it can give the impression of image alteration to critical readers.

To address this, please upload the blot figures and the Appendix blot figure file at a higher resolution. Please upload the original blot source data (at 16-bit TIFF) with your online submission or deposit the raw files on BioStudies

<https://www.ebi.ac.uk/biostudies/sourcedata/studies>

and including the archive accession number in your Data Availability section.

This will enable us to confirm the integrity of the complete figure set and enhance transparency for readers.

All immunoblots were scanned using a LI-COR Odyssey imager. However, during this revision, we realized that the original blots were not scanned with a high resolution. We sincerely apologize for this resolution issue.

As instructed, we converted the blot figures and the source blot figure files with a high resolution. We also included the original unprocessed immunoblots in 16-bit TIFF format as the source data. Hope this will enable you to check the integrity of these immunoblots. Please let us know if you have any additional questions or suggestions.

In addition, I would need from you uploaded separately:

- a short, two-sentence summary of the manuscript (not more than 35 words).

Provided in the “Summary” file.

- two to four short (!) bullet points highlighting the key findings of your study (two lines each).

Provided in the “Bullet points” file.

- a schematic summary figure as separate file that provides a sketch of the major findings (not a data image) in jpeg or tiff format (with the exact width of 550 pixels and a height of not more than 400 pixels) that can be used as a visual synopsis on our website.

A schematic summary figure was created in BioRender as instructed. The figure file and a publication license from BioRender were uploaded. Please let us know if you have any questions or suggestions on the content of the figure.

Dr. Xiaoling Li
NIH/NIEHS
Molecular and Cellular Biology Laboratory
111 TW Alexander Drive
Research Triangle Park, NC 27709
United States

Dear Dr. Li,

I am very pleased to accept your manuscript for publication in the next available issue of EMBO reports. Thank you for your contribution to our journal.

You may qualify for financial assistance for your publication charges - either via a Springer Nature fully open access agreement or an EMBO initiative. Check your eligibility: <https://link.springer.com/journal/44319/how-to-publish-with-us>

Yours sincerely,

>>> Please note that it is EMBO Reports policy for the transcript of the editorial process (containing referee reports and your response letter) to be published as an online supplement to each paper. If you do NOT want this, you will need to inform the Editorial Office via email immediately. More information is available here: <https://link.springer.com/partners/embo-press/editorial-policies#Peer%20review>